# Somatotopic organization among parallel sensory pathways that promote a grooming sequence in *Drosophila*

Katharina Eichler[1][†], Stefanie Hampel[1]*[†], Adrián Alejandro-García[1][†], Steven A Calle-Schuler[1], Alexis Santana-Cruz[1], Lucia Kmecova[1], Jonathan M Blagburn[1], Eric D Hoopfer[2], Andrew M Seeds[1]*

[1]Institute of Neurobiology, University of Puerto Rico-Medical Sciences Campus, San Juan, Puerto Rico; [2]Neuroscience Program, Carleton College, Northfield, United States

*For correspondence:
stef.hampel@gmail.com (SH);
seeds.andrew@gmail.com (AMS)

[†]These authors contributed equally to this work

Competing interest: The authors declare that no competing interests exist.

**Abstract** Mechanosensory neurons located across the body surface respond to tactile stimuli and elicit diverse behavioral responses, from relatively simple stimulus location-aimed movements to complex movement sequences. How mechanosensory neurons and their postsynaptic circuits influence such diverse behaviors remains unclear. We previously discovered that *Drosophila* perform a body location-prioritized grooming sequence when mechanosensory neurons at different locations on the head and body are simultaneously stimulated by dust (Hampel et al., 2017; Seeds et al., 2014). Here, we identify nearly all mechanosensory neurons on the *Drosophila* head that individually elicit aimed grooming of specific head locations, while collectively eliciting a whole head grooming sequence. Different tracing methods were used to reconstruct the projections of these neurons from different locations on the head to their distinct arborizations in the brain. This provides the first synaptic resolution somatotopic map of a head, and defines the parallel-projecting mechanosensory pathways that elicit head grooming.

## eLife assessment

This **valuable** work provides a near-complete description of the mechanosensory bristles on the *Drosophila melanogaster* head and the anatomy and projection patterns of the bristle mechanosensory neurons that innervate them. The data presented are **solid**. The study has generated numerous resources for the community that will be of interest to neuroscientists in the field of circuits and behaviour, particularly those interested in mechanosensation and behavioural sequence generation.

## Introduction

The ability to produce complex behaviors by assembling sequences of different movements is essential for purposeful behavior and survival. One prominent model that describes how the brain produces movement sequences is called a 'parallel model'. This model proposes that the premotor elements of different movements to be executed in sequence are activated (or readied) in parallel and then selected sequentially through a mechanism where movements occurring earlier in the sequence suppress later ones (*Bohland et al., 2010*; *Bullock, 2004*; *Houghton and Hartley, 1995*; *Lashley, 1951*). A hallmark feature of this model is a parallel circuit architecture that ensures all mutually exclusive actions to be performed in sequence are simultaneously readied and competing for output. Performance order is established by hierarchical suppression among the parallel circuits, where earlier actions suppress later actions. This architecture is supported by physiological and behavioral evidence

from the movement sequences of different animals (*Averbeck et al., 2002*; *Mushiake et al., 2006*; *Seeds et al., 2014*). Yet, despite some movement sequences exhibiting features consistent with the parallel model, we lack an organizational and mechanistic understanding of the underlying neural circuits.

The grooming behavior of fruit flies (*Drosophila melanogaster*) can be studied to define the circuit mechanisms that produce movement sequences. Making flies dirty by coating them in dust elicits a grooming sequence that starts with the cleaning of different locations on the head, such as the eyes, antennae, and proboscis, and proceeds to body locations, such as the abdomen, wings, and thorax (*Mueller et al., 2019*; *Phillis et al., 1993*; *Seeds et al., 2014*). We previously determined that the sequence is produced by a mechanism that is consistent with a parallel model (*Seeds et al., 2014*). The sequence begins when different aimed grooming movements that clean specific locations of the head or body become activated in parallel by dust. The resulting competition among mutually exclusive grooming movements is resolved through hierarchical suppression. For example, grooming of the eyes occurs first because eye grooming suppresses grooming of other locations on the head and body. This parallel model of hierarchical suppression provides a conceptual framework for dissecting the neural circuit architecture that produces *Drosophila* grooming (*Figure 1—figure supplement 1A and B*). Here, we focus on the organization of the sensory inputs in the hypothesized architecture (*Seeds et al., 2014*), the parallel mechanosensory neurons that detect dust at different locations and elicit aimed grooming movements.

Different mechanosensory structures are distributed across the head and body surface that respond to mechanical stimuli and elicit grooming. The most abundant of these structures are mechanosensory bristles (aka hairs or setae). Tactile displacement of individual bristles elicits grooming movements in which the legs are precisely aimed at the stimulus location (*Corfas and Dudai, 1989*; *Page and Matheson, 2004*; *Vandervorst and Ghysen, 1980*). Each bristle is innervated by a single *bristle mechanosensory neuron* (BMN) that is excited by displacement of that bristle (*Corfas and Dudai, 1990*; *Tuthill and Wilson, 2016a*; *Walker et al., 2000*). Thus, bristles and their corresponding BMNs can be ascribed to specific, aimed leg grooming movements. Other mechanosensory structures, including chordotonal organs and stretch receptors, also elicit stimulus location-aimed grooming (*Hampel et al., 2015*; *Zhang et al., 2020*). Simultaneous (parallel) optogenetic activation of mechanosensory neurons across the body elicits a grooming sequence that proceeds in the same order as the 'natural' dust-induced sequence (*Hampel et al., 2017*; *Zhang et al., 2020*). Thus, the sequence is elicited by parallel mechanosensory pathways that each produce a movement that grooms a specific location on the head or body (*Figure 1—figure supplement 1A and B*).

BMNs project their axons from different locations on the head or body, through different nerves, and into the *central nervous system* (CNS). Previous studies of BMNs from different body locations demonstrated that they show somatotopic organization in their CNS projections (*Johnson and Murphey, 1985*; *Murphey et al., 1989b*; *Newland, 1991*; *Newland et al., 2000*; *Tsubouchi et al., 2017*). That is, particular projection zones in the CNS correspond to specific body locations. Somatotopic organization among mechanosensory neurons and their postsynaptic circuits is consistent with the parallel model that underlies the body grooming sequence (*Seeds et al., 2014*). In this model, parallel-projecting mechanosensory neurons that respond to stimuli at specific locations on the head or body could connect with somatotopically organized parallel circuits that elicit grooming of those locations (*Figure 1—figure supplement 1A–C*). The previous discovery of a mechanosensory-connected circuit that elicits aimed grooming of the antennae provides evidence of this organization (*Hampel et al., 2015*). However, the extent to which distinct circuits elicit grooming of other locations is unknown, in part, because the somatotopic projections of the mechanosensory neurons have not been comprehensively defined for the head or body.

Here, we comprehensively map the somatotopic organization among BMNs that elicit grooming of different locations on the head. *Drosophila* use their front legs to groom their heads in a sequence that starts with the eyes and proceeds to other locations, such as the antennae and proboscis (*Seeds et al., 2014*). Two mechanosensory structures on the head (i.e. chordotonal organs and bristles) are implicated in grooming. The antennal *Johnston's organ* (JO) is a chordotonal organ containing mechanosensory neurons called *JO neurons* (JONs) that detect stimulations of the antennae and elicit aimed grooming (*Hampel et al., 2015*). There are over 1000 bristles located on the head whose stimulation we postulated could also elicit aimed grooming of different head locations. In support of this,

BMNs innervating bristles on the eyes were previously shown to elicit grooming of the eyes (*Hampel et al., 2017*; *Zhang et al., 2020*). Here, we use optogenetic tools to show that activation of subsets of BMNs at other head locations also elicits aimed grooming. We use transgenic expression, dye fills, and electron microscopy (EM) reconstructions to trace the projections of nearly all BMNs on the head, from their bristles, through their respective nerves, and into the CNS. This reveals somatotopic organization, where BMNs innervating neighboring head bristles project to overlapping zones in the CNS while those innervating distant bristles project to distinct zones. Analysis of head BMN post-synaptic connectivity reveals that neighboring BMNs show higher connectivity similarity than distant BMNs, providing evidence of somatotopically organized postsynaptic circuit pathways. This provides a comprehensive synaptic resolution projection map of head mechanosensory neurons, and further defines the organization of parallel mechanosensory pathways that elicit sequential grooming.

## Results

### Classification and quantification of the head bristles

A prerequisite for determining the somatotopy of head BMNs was to define the locations of their respective bristles on the head. Different populations of bristles are located on the eyes, antennae, proboscis, and other areas on the head. While the identities of most of these populations were known (*Bodenstein et al., 1994*), some were poorly described and their bristle numbers were not reported. Therefore, we imaged the bristles on the head and then classified and quantified each population. We developed a unified nomenclature for the different bristle populations that was based partially on published nomenclature. Most of the bristles were easily observed by imaging white light-illuminated heads (*Figure 1A–D*), and color-coded depth maps further helped to distinguish between bristles while they were being counted (*Figure 1—figure supplement 2A–H*). Some bristles could not be counted from these images because of their small size, position on the head, or because they could not be distinguished from one another (*Figure 1E*, asterisk with bristle number range). Therefore, we used confocal microscopy images, or referred to published work to estimate or obtain the numbers of bristles in these populations (see Materials and methods).

We next produced a map of the different bristles at their stereotyped locations on the head, and determined how the numbers of bristles in each population varied across individual flies (*Figure 1A–E*). By counting the bristles on both male and female heads, we found no significant sex-based differences in their numbers (*Figure 1—figure supplement 3A–E*, see *Supplementary file 1* for bristle counts for each head, see *Supplementary file 2* for head image downloads). Given that the bristles are singly innervated (*Tuthill and Wilson, 2016b*), we could use the bristle counts to estimate the number of BMNs for each bristle population. This provided a framework for us to define the somatotopic projections of BMNs that innervate particular bristles.

### Light microscopy-based reconstruction of BMNs innervating the head bristles

BMNs project from bristles at specific head locations and then through their respective nerves to enter the brain. While the nerve projections of BMNs innervating bristles on the eyes, proboscis, and antennae were previously reported (*Hampel et al., 2017*; *Homberg et al., 1989*; *Melzig et al., 1996*; *Naresh Singh and Nayak, 1985*; *Stocker, 1994*), the projections of BMNs innervating other head bristles were unknown. We determined these projections using a transgenic driver line (R52A06-GAL4) that labels BMNs on the head (*Hampel et al., 2017*). R52A06-GAL4 was used to express membrane-targeted *green fluorescent protein* (mCD8::GFP), and the anterior and posterior head was imaged with a confocal microscope (see *Supplementary file 2* for confocal Z-stack downloads). The GFP-labeled neurons had all the characteristic morphological features of BMNs (*Tuthill and Wilson, 2016b*), including a dendrite innervating a bristle, a cell body, and an axon (*Figure 2A and B*). R52A06-GAL4 labeled almost all BMNs on the head, but did not label any associated with the *postocellar* (PoOc) or *supracervical* (Su) bristles (*Figure 2—figure supplement 1A–H*). We used the software neuTube (*Feng et al., 2015*) to reconstruct the GFP-labeled projections of head BMNs from confocal Z-stacks (*Figure 2C–H*). The reconstructions enabled us to classify the BMNs into 'nerve groups', based on the nerves they project through to enter the brain (*Figure 2I and J*, groups listed in *Figure 1E*). This revealed that BMNs innervating bristles at different locations on the head project

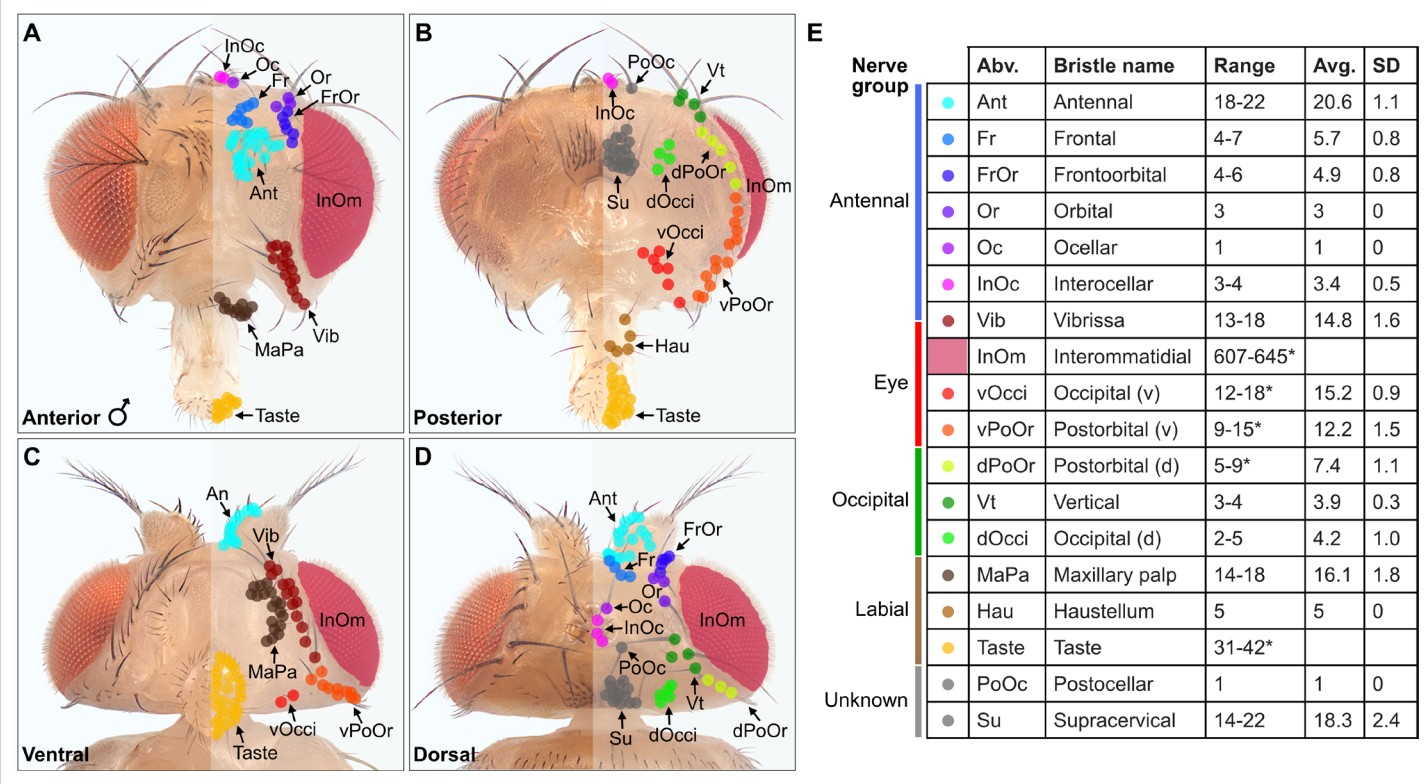

**Figure 1.** Classification and quantification of *D. melanogaster* head bristles. (**A–D**) Bristles on the anterior (**A**), posterior (**B**), ventral (**C**), and dorsal (**D**) male head. The bristles on the right half are marked with color-coded dots to indicate their classification. Bristle names are abbreviated (Abv.), and full names and color codes are listed in (**E**). (**E**) Quantification of bristle populations on the male head (per half). Range indicates the lowest and highest number of bristles counted across individuals for each population (N=8). Bristle number average (Avg.) and standard deviation (SD) across individuals for each population are shown. Bristle counting was facilitated using color-coded depth maps (examples shown in *Figure 1—figure supplement 2*). Quantification of bristles on female heads and male/female comparisons are shown in *Figure 1—figure supplement 3*. See *Supplementary file 1* for bristle counts for each head and *Supplementary file 2* for image stack download links for each head. *InOm and Taste bristle number ranges are based on published data while dPoOr, PoOr, and Occi bristles were counted using confocal microscopy (see Materials and methods). Bristles are organized into nerve groups based on the nerve each bristle's corresponding bristle mechanosensory neuron (BMN) projects through to enter the brain (evidence shown in *Figure 2*). Dorsal (d) and ventral (v).

The online version of this article includes the following figure supplement(s) for figure 1:

**Figure supplement 1.** Hypothesized grooming circuit architecture features somatotopically organized parallel mechanosensory pathways.

**Figure supplement 2.** Color-coded depth maps of the head.

**Figure supplement 3.** Comparison of bristle numbers on male and female heads.

through specific nerves, including the antennal, eye, occipital, and labial nerves. Below we introduce the BMNs in each nerve group and the bristles that they innervate.

## Head BMNs project to the brain through specific nerves

BMNs innervating the 18–22 *antennal* (Ant) bristles were previously reported to project through the *antennal nerve* (AntNv) that also carries the axons of JONs and olfactory neurons (*Homberg et al., 1989*; *Melzig et al., 1996*). We identified additional BMNs projecting through the AntNv that innervate bristles located on the anterior and dorsal head (*Figure 2E, I*, blue). These include four to seven *frontal* (Fr) bristles located medially, three *orbital* (Or), and four to six *frontoorbital* (FrOr) bristles located laterally, and one *ocellar* (Oc) and three to four *interocellar* (InOc) bristles located on the dorsal head. BMNs projecting from these bristles form a bundle below the cuticle that projects ventrally to join the AntNv. We also identified BMNs that innervate one to three of the small anterior *vibrissae* (Vib) on the ventral head whose axons project dorsally to join the AntNv.

BMNs innervating bristles on the dorsal half of the posterior head project through a previously undescribed nerve that we named the *occipital nerve* (OcciNv) (*Figure 2F and J*, green). This includes

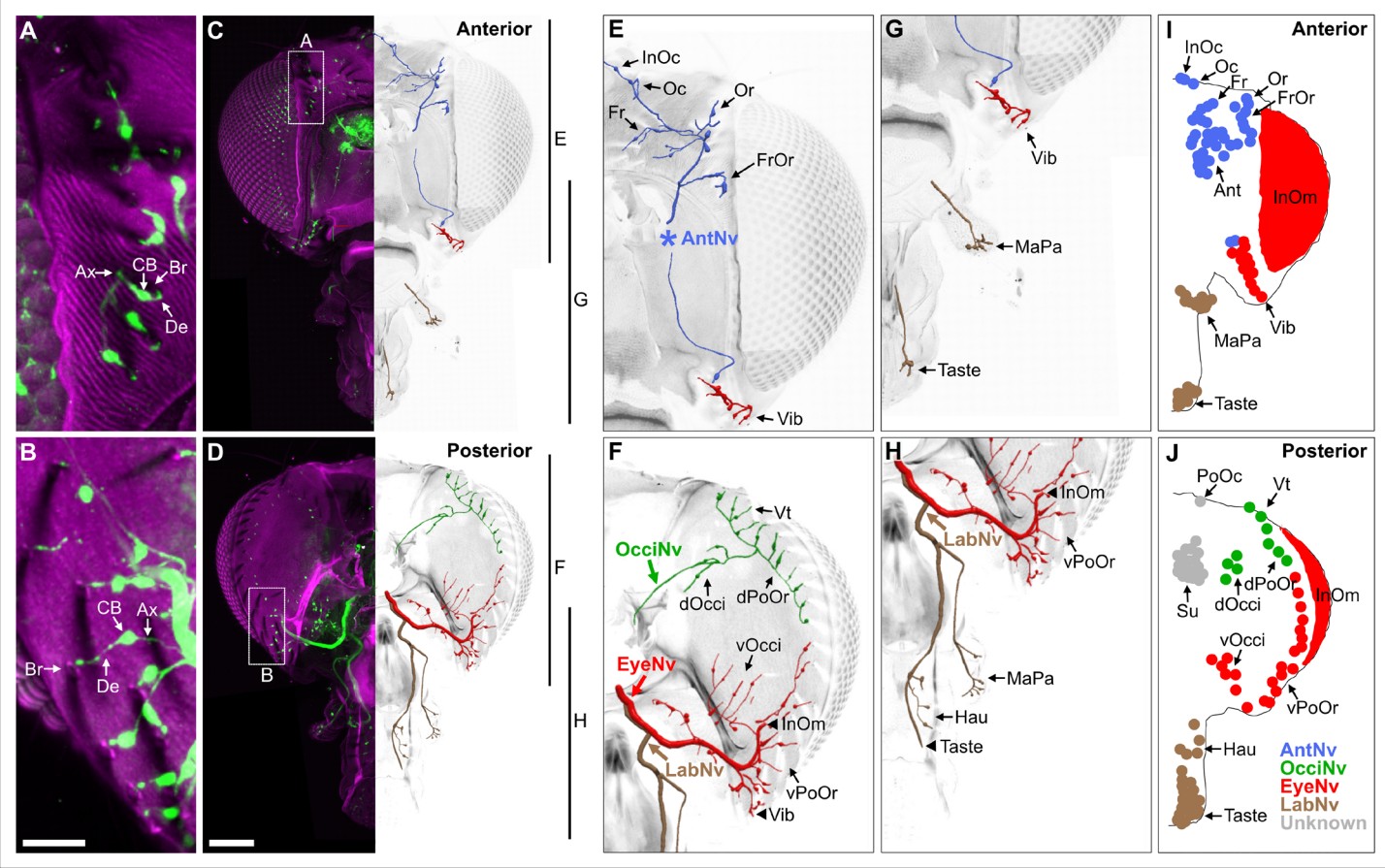

**Figure 2.** Bristle mechanosensory neurons (BMNs) on the head project through specific nerves. (**A–D**) Confocal Z-stack maximum intensity projections of the anterior (**A, C**) and posterior (**B, D**) head in which the driver line R52A06-GAL4 drives expression of GFP in BMNs (green). Cuticle is magenta. (**A, B**) Magnified views of the boxed areas indicated in (**C**) and (**D**). The *dendrite* (De), *axon* (Ax), *cell body* (CB), and innervated *bristle* (Br) of a BMN are indicated in each panel. (**C, D**) The left half of the head is shown as a maximum projection, while Z-stack-reconstructed BMNs are shown for the right half. Maximum projections of the right half of the head is shown in *Figure 2—figure supplement 1A–F*. (**E–H**) Magnified images of the reconstructions. The magnified areas are indicated by vertical lines on the right in (**C**) and (**D**). Reconstructed BMNs are color-coded and labeled according to the nerve that they project through: AntNv (blue); OcciNv (green); EyeNv (red); LabNv (brown). Unreconstructed portion of the antennal nerve is indicated by an asterisk. Innervated bristles are indicated with black arrows. Black arrowheads in (**F**) and (**H**) indicate partially reconstructed axons of BMNs innervating the InOm, Vib, and Taste bristles. Scale bars: 25 μm (**B**), 100 μm (**D**). (**I, J**) Summary of bristles innervated by BMNs that belong to particular nerve groups on the anterior (**I**) and posterior (**J**) head. Nerve groups also listed in *Figure 1E*, and *Supplementary file 2* provides confocal Z-stack download links.

The online version of this article includes the following figure supplement(s) for figure 2:

**Figure supplement 1.** R52A06-GAL4 expression in head bristle mechanosensory neurons (BMNs).

the three to four *vertical* (Vt), two to five *dorsal occipital* (dOcci), and five to nine *dorsal postorbital* (dPoOr) bristles. BMNs that innervate these different bristles form the OcciNv that projects under the cuticle ventromedially toward the brain.

Each eye contains between 645 and 828 regularly spaced ommatidia, many of which have an associated *interommatidial* (InOm) bristle (*Ready et al., 1976*). We estimated that there are between 607 and 645 InOm bristles on each eye based on published data (see Materials and methods). BMNs that innervate the InOm bristles were previously found to form a nerve that projects to the brain from the posterior head (*Hampel et al., 2017*). Because this nerve was not previously named, it is referred to here as the *eye nerve* (EyeNv). We found that the EyeNv also carries the projections of BMNs innervating bristles on the posterior and ventral head (*Figure 2E–J*, red). Those on the posterior head innervate the 12–18 *ventral occipital* (vOcci) and 9–15 *ventral postorbital* (vPoOr) bristles. Those on the ventral head innervate most of the 13–18 Vib bristles.

The proboscis has bristles on the labellum, haustellum, and maxillary palps. Each half of the labellum has 31–42 *Taste* bristles whose associated BMNs project through the *labial nerve* (LabNv) (*Falk et al.,*

*1976*; *Jeong et al., 2016*; *Nayak and Singh, 1983*; *Shanbhag et al., 2001*; *Stocker, 1994*). The LabNv also carries mechanosensory neurons innervating the labellar taste pegs, along with gustatory neurons innervating either the taste pegs or taste bristles (*Stocker and Schorderet, 1981*). We found that BMNs innervating the five *haustellum* (Hau) bristles also project through the LabNv (*Figure 2H and J*, brown). BMNs that innervate the 14–18 *maxillary palp* (MaPa) bristles project through the *maxillary nerve* (MaxNv) that also carries the axons of olfactory neurons (*Naresh Singh and Nayak, 1985*). The Lab- and MaxNvs merge as they approach the head, and in this work we refer to the merged nerve as the LabNv. The LabNv then merges with the EyeNv in the ventral head, suggesting that these nerves project into the brain at the same location (*Figure 2H*).

## Head BMNs project into discrete zones in the ventral brain

BMNs in the Ant-, Eye-, and LabNvs were previously reported to project into a region of the ventral brain called the *subesophageal zone* (SEZ) (*Figure 3A and B*; *Hampel et al., 2017*; *Jeong et al., 2016*; *Kamikouchi et al., 2006*; *Mitchell et al., 1999*; *Naresh Singh and Nayak, 1985*; *Stocker, 1994*). To determine if all head BMNs project into the SEZ, we used R52A06-GAL4 to label their projections in a dissected brain (*Figure 3C*, see *Supplementary file 2* for confocal Z-stack download). The AntNv was identified in the R52A06-GAL4 pattern based on its reported dorsal-arriving projection into the SEZ (*Kamikouchi et al., 2006*; *Stocker, 1994*), while the Eye- and LabNvs were identified based on their reported ventral-arriving projections (*Hampel et al., 2017*; *Stocker, 1994*). We found that the Eye- and LabNvs project into the ventral SEZ at the same location (*Figure 3C*), consistent with the observation that they merge as they approach the brain (*Figure 2F and H*). We tentatively identified the OcciNv projecting into the SEZ from a lateral direction, revealing that all head BMN nerves project into the SEZ (*Figure 3C*). R52A06-GAL4 also labels the antennal chordotonal JONs that are known to project through the AntNv into a dorsal region of the SEZ (*Hampel et al., 2017*; *Kamikouchi et al., 2006*; *Kim et al., 2020*). Visualization of JONs and BMNs in the same expression pattern revealed that most of the BMNs project into more ventral regions of the SEZ than the JONs (*Figure 3C*).

We next used different transgenic driver lines that express in specific populations of head BMNs to independently label and visualize the different nerves (*Figure 3D–I*, *Figure 3—figure supplement 1A–D*, see *Supplementary file 2* for confocal Z-stack downloads). The EyeNv was labeled using a previously identified driver line (VT017251-LexA) that expresses in BMNs innervating the InOm bristles (*Figure 3D*; *Hampel et al., 2017*). Here, we refer to this line as InOmBMN-LexA. We also used a screening approach to produce three new Split GAL4 (spGAL4) combinations that express in BMNs innervating bristles at other locations on the head (see Materials and methods). One line named dBMN-spGAL4 labels BMNs innervating some dorsally located bristles (InOc, Vt, and dPoOr) that project through the Ant- and OcciNvs (*Figure 3E*). Another line named pBMN-spGAL4 labels BMNs innervating bristles on the posterior head (Vt, dOcci, dPoOr, and vOcci) that project through the Occi- and EyeNvs (*Figure 3F*). The third line named TasteBMN-spGAL4 labels BMNs innervating Taste bristles on the labellum that project through the LabNv (*Figure 3G*). These driver lines each provided independent labeling of one or two different nerves (*Figure 3H*).

Consistent with what we observed using R52A06-GAL4, each driver line labeled BMNs that projected into the SEZ and no other regions of the brain or *ventral nerve cord* (VNC) (*Figure 3D'–G'*, *Figure 3—figure supplement 1A'–D'*, see *Supplementary file 2* for confocal Z-stack downloads). dBMN-spGAL4 and pBMN-spGAL4 both labeled the OcciNv that was found to project into the SEZ from a lateral direction (*Figure 3E' and F'*), in agreement with what we observed in the R52A06-GAL4 pattern (*Figure 3C*). A comparison of the nerves labeled by the different driver lines revealed that each nerve has morphologically distinct projections. To further visualize the spatial relationships between these projections, we computationally aligned the expression patterns of the different driver lines into the same brain space (*Figure 3J*, upper right corner). Indeed, BMNs from different nerves were found to project into distinct zones of the ventral SEZ. However, we also observed potential zones where overlap could occur between the projections of BMNs from different nerves (discussed more below).

## Brain projections of BMNs that innervate specific head bristles

Our results suggested that different BMN 'types' innervate specific populations of bristles on the head and project into distinct zones in the SEZ. However, it was unclear to what extent BMNs of the

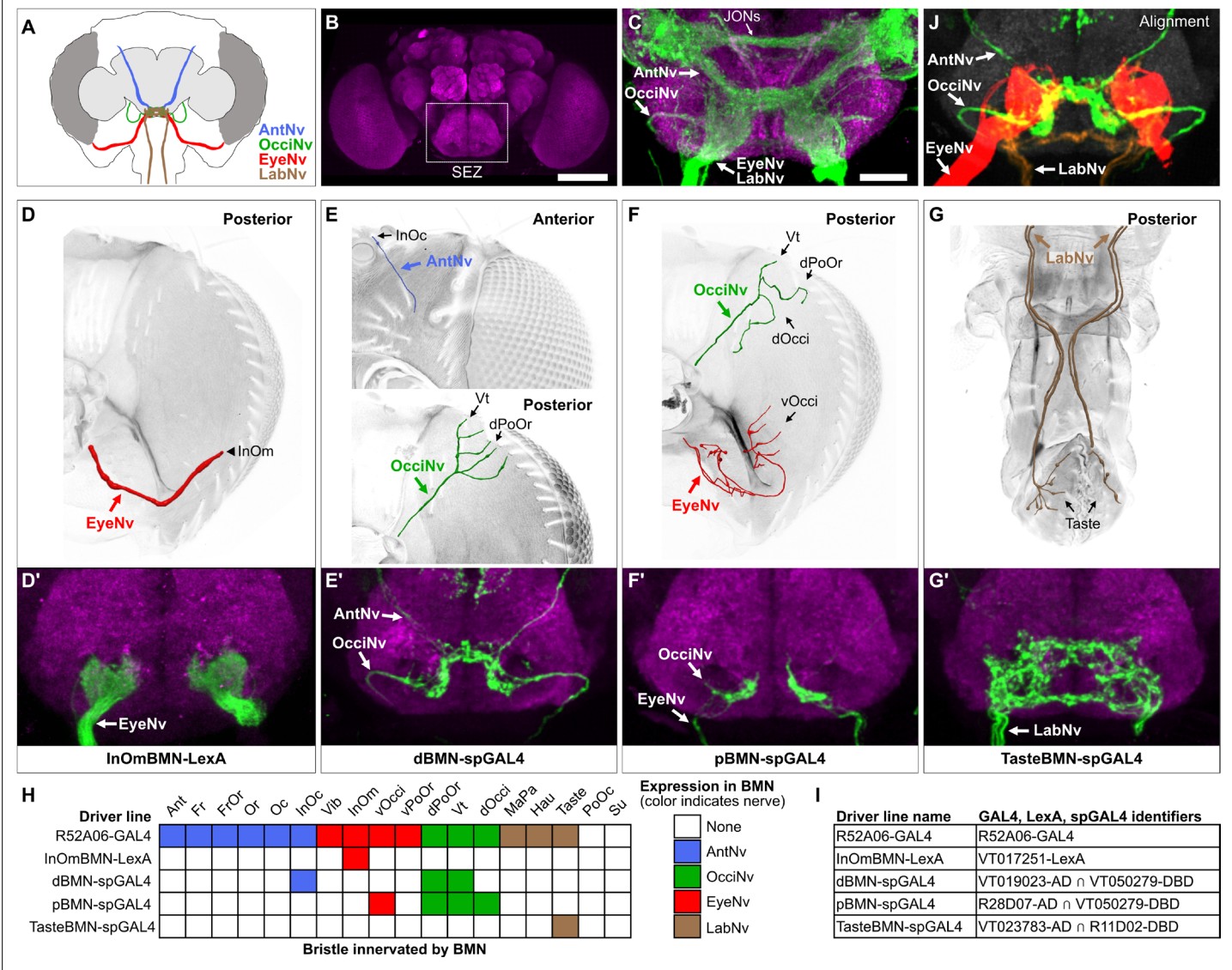

**Figure 3.** Head bristle mechanosensory neurons (BMNs) project into the ventral brain region called the subesophageal zone (SEZ). (**A**) Schematic of BMNs projecting from different nerves into the SEZ. (**B**) Anterior view of the brain immunostained for Bruchpilot (magenta) to visualize the neuropile. White box indicates the SEZ. Scale bar, 100 μm. (**C**) Image of the SEZ in which R52A06-GAL4 expressed GFP in BMNs and Johnston's organ neurons (JONs). Brains were immunostained for GFP (green) and Bruchpilot (magenta). BMN nerves and JONs are labeled. Scale bar, 25 μm. (**D–G**) Driver lines that label BMNs from different nerves. Reconstructed BMNs on half of the head that are labeled by the following driver lines: InOmBMN-LexA (**D**), dBMN-spGAL4 (**E**), pBMN-spGAL4 (**F**), and TasteBMN-spGAL4 (whole proboscis shown) (**G**). Images of the heads used for each reconstruction are shown in *Figure 3—figure supplement 1A–D*. Reconstructed neurons are color-coded and labeled as described in *Figure 2*. (**D'–G'**) SEZ projections of BMNs from both halves of the head that are labeled by InOmBMN-LexA (**D'**), dBMN-spGAL4 (**E'**), pBMN-spGAL4 (**F'**), and TasteBMN-spGAL4 (**G'**). (**H**) Table of BMNs innervating specific bristles that are labeled by each driver line, indicated by box shading (numbers of labeled BMNs innervating different bristles shown in *Figure 3—figure supplement 1E*). Shaded color indicates the nerve that each BMN projects through. (**I**) Driver line names and identifiers. (**J**) Shown in the upper right corner of the figure are the aligned expression patterns of InOmBMN-LexA (red), dBMN-spGAL4 (green), and TasteBMN-spGAL4 (brown). *Supplementary file 2* provides confocal Z-stack download links.

The online version of this article includes the following figure supplement(s) for figure 3:

**Figure supplement 1.** Driver line expression in head bristle mechanosensory neuron (BMNs).

same type projected to the same zones, and if other BMN types had distinct or overlapping projections. Therefore, we next compared the projections of individual BMNs from different populations of bristles.

The head contains different sized bristles, ranging from large Vt bristles on the dorsal head, to small vOcci bristles on the posterior head. We performed dye fills to label individual BMNs that innervate the largest bristles. This was done by modifying a previously published method for filling BMNs innervating bristles on the thorax (*Kays et al., 2014*). In the modified method, a particular bristle was plucked from the head and a small volume of dye (DiD) pipetted into the exposed socket containing the dendrite of the associated BMN. The dye then diffused into the neuron, and its projection morphology in the brain was imaged using a confocal microscope (experiment schematic and example fills shown in *Figure 4—figure supplement 1A–E*). This method was particularly amenable to large bristles that were relatively easy to pluck. We successfully filled individual BMNs that innervate the Oc, Or, Ant, Vib, and Vt bristles (*Figure 4A–Q*, see *Supplementary file 2* for confocal Z-stack downloads). The BMNs were named based on the bristle populations that they innervate. For example, BMNs that innervate the Ant bristles were named *bristle mechanosensory Ant neurons* (BM-Ant neurons).

The large bristles are invariant in number and location across individuals (*Figure 1A–E*). For example, all flies have one Oc bristle on each half of the head that is always in the same location. We therefore performed dye fills on the same bristles from multiple different heads. This revealed that BMNs innervating the same bristle have the same general projection morphology across individual flies (*Figure 4—figure supplements 2–5*). We also performed dye fills on different bristles from the same population, such as the Ant 1, Ant 2, Ant 3, and Ant 4 bristles (*Figure 4—figure supplement 3A–M*). BMNs innervating the same populations were found to have similar projections. For example, BM-Ant neurons all showed similar ipsilateral and midline projecting branches (*Figure 4G–J*). Morphological similarity among BMNs innervating the same bristle populations was also observed for the BM-Or (*Figure 4D–F*), -Vib (*Figure 4K–N*), and -Vt (*Figure 4O–Q*) neurons.

While BMNs innervating the largest bristles could be labeled using dye fills, we could not label BMNs innervating small bristles using this method. Therefore, we used the *multicolor flipout* (MCFO) method (*Nern et al., 2015*) to stochastically label individual BMNs innervating bristles within the expression patterns of the driver lines shown in *Figure 3E'–G'*. This enabled us to determine the morphologies of BMNs that innervate the InOc, dOcci, dPoOr, vOcci, and Taste bristles (*Figure 4R–V*, see *Supplementary file 2* for confocal Z-stack downloads). Unlike the dye-filled BMNs, the MCFO-labeled BMNs could not be matched to specific bristles within a population (e.g. Ant 1 or Ant 2), but only to a specific population (e.g. Ant). In agreement with what we observed with dye-filled BMNs innervating the same populations of large bristles, the MCFO-labeled BMNs innervating the same populations of small bristles also showed similar projection morphologies (*Figure 4—figure supplements 6–8*).

We next compared the projections of the dye-filled and MCFO-labeled BMNs (*Figure 4C–V*). This revealed that some BMNs innervating neighboring bristle populations have similar morphologies. For example, BM-InOc and -Oc neurons have similar morphology, including ipsilateral and midline-crossing projections (*Figure 4C and R*), while BM-dPoOr, -dOcci, and -vOcci neurons show similar ipsilateral projections. This suggested that BMNs innervating neighboring head bristle populations show similar morphology and project into overlapping zones in the SEZ.

## EM-based reconstruction of the head BMN projections in a full adult brain

We next used a previously reported serial-section EM volume of a *full adult fly brain* (FAFB) to reconstruct the SEZ projections of all head BMNs and produce a comprehensive map of their organization (*Zheng et al., 2018*). FAFB consists of a brain that was dissected from the head capsule, making it impossible to reconstruct BMNs all the way from their bristles. Instead, the severed Ant-, Occi-, Eye-, and LabNvs were identified in FAFB at the same anatomical locations that we had observed using light microscopy (*Figure 3C and J*, *Figure 5—figure supplement 1A*). We used the FlyWire.ai platform (*Dorkenwald et al., 2023*; *Dorkenwald et al., 2022*) to seed all automatically segmented neurons within the different nerve bundles as they entered the neuropil (left brain hemisphere nerves, *Figure 5—figure supplement 1B–D*), and the neurons were then fully proofread and edited by human

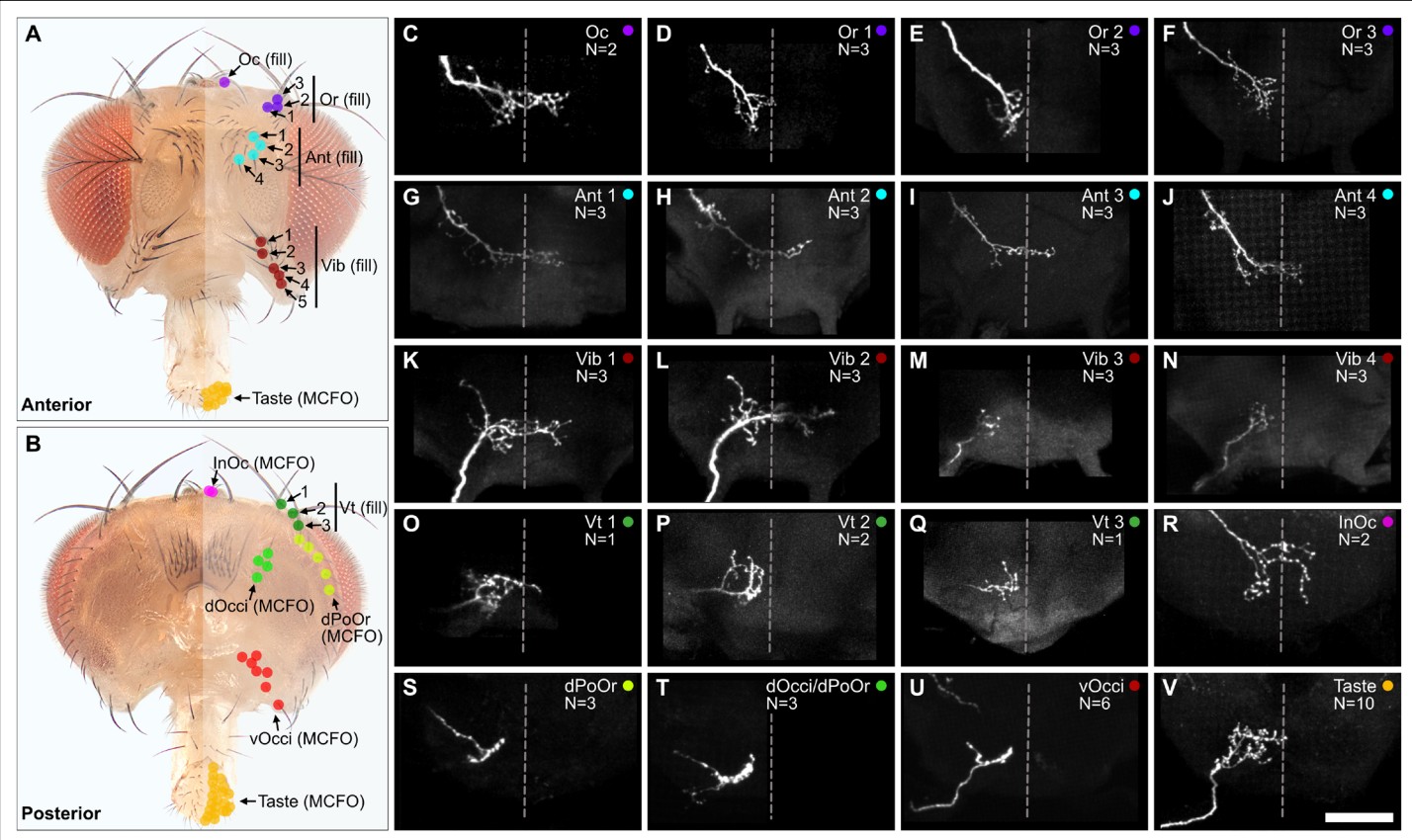

**Figure 4.** Projections of bristle mechanosensory neurons (BMNs) that innervate specific head bristles. (**A, B**) Bristles on the anterior (**A**) and posterior (**B**) head whose associated BMNs were labeled using dye fill (**C–Q**, fill) or multicolor flipout (**R–V**, MCFO) techniques. (**C–V**) Subesophageal zone (SEZ) projections of individual BMNs that innervate the bristle indicated in the upper right corner (anterior view). BMNs are oriented as if they are projecting from the right side of the head. Dotted line indicates approximate SEZ midline. Scale bar, 50 μm. (**C–Q**) BMNs labeled by dye filling. Schematic of the filling technique and whole brain examples shown in *Figure 4—figure supplement 1*. Filled BMNs innervate the Oc (**C**), Or (**D–F**), Ant (**G–J**), Vib (**K–N**), and Vt (**O–Q**) bristles. All fill trials for the different bristles are shown in *Figure 4—figure supplement 2*, *Figure 4—figure supplement 3*, *Figure 4—figure supplement 4*, and *Figure 4—figure supplement 5*. (**R–V**) MCFO-labeled BMNs innervate the InOc (**R**), dPoOr (**S**), dOcci/dPoOr (**T**), vOcci (**U**), and Taste (**V**) bristles. BMNs were MCFO labeled using the following driver lines: dBMN-spGAL4 (**R, S**), pBMN-spGAL4 (**T, U**), and TasteBMN-spGAL4 (**V**). All MCFO trials for the different bristles are shown in *Figure 4—figure supplement 6*, *Figure 4—figure supplement 7*, and *Figure 4—figure supplement 8*. The number (N) of fill or MCFO trials obtained for each BMN is indicated in the upper right corner. *Supplementary file 2* provides confocal Z-stack download links.

The online version of this article includes the following figure supplement(s) for figure 4:

**Figure supplement 1.** Overview of the dye filling technique and whole brain examples.

**Figure supplement 2.** Different fill trials for Oc and Or bristles.

**Figure supplement 3.** Different fill trials for Ant bristles.

**Figure supplement 4.** Different fill trials for Vib bristles.

**Figure supplement 5.** Different fill trials for Vt bristles.

**Figure supplement 6.** Multicolor flipout (MCFO) trials for bristle mechanosensory neurons (BMNs) innervating the InOc, dPoOr, and Vt bristles.

**Figure supplement 7.** Multicolor flipout (MCFO) labeled trials for bristle mechanosensory neurons (BMNs) innervating the dOcci/dPoOr and vOcci bristles.

**Figure supplement 8.** Multicolor flipout (MCFO)-labeled trials for Taste bristles.

experts to identify their individual morphologies. The morphologies of the majority of the reconstructed neurons matched those of mechanosensory neurons, including BMNs (discussed below), JONs (*Hampel et al., 2020a*; *Kamikouchi et al., 2006*; *Kim et al., 2020*), and labellar taste peg mechanosensory neurons (TPMNs) (*Jeong et al., 2016*; *Miyazaki and Ito, 2010*; *Zhou et al., 2019*; *Figure 5—figure supplement 1E*). The remaining neurons included gustatory neurons (*Engert et al.,*

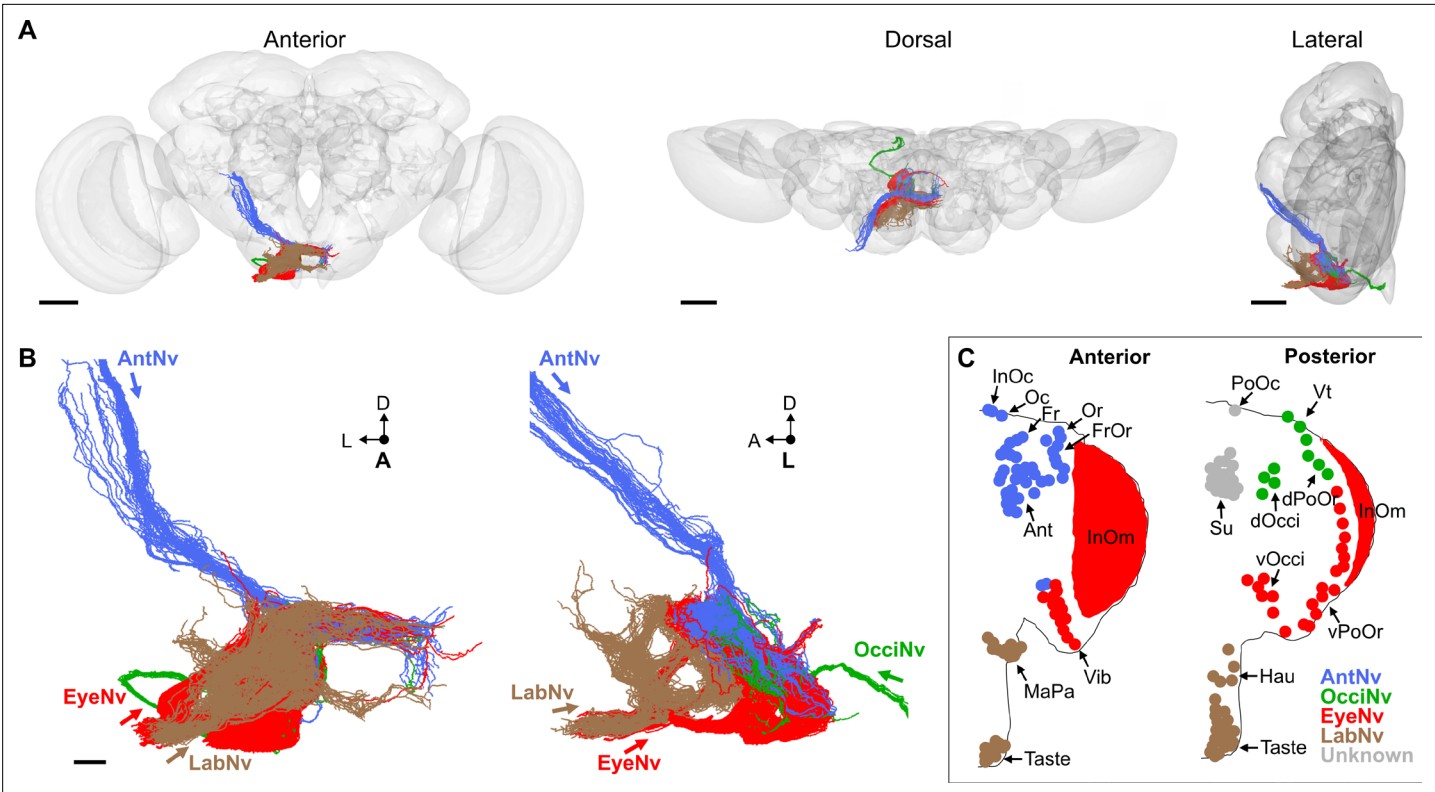

**Figure 5.** Electron microscopy (EM)-based reconstruction of head bristle mechanosensory neurons (BMNs). (**A**) All reconstructed BMNs projecting into the brain from the left side of the head (anterior, dorsal, and lateral views shown). BMN colors correspond to the nerves that they project through, including the AntNv (blue), EyeNv (red), OcciNv (green), and LabNv (brown). Scale bars, 50 μm. (**B**) Zoomed anterior (left) and lateral (right) views of the BMNs in the subesophageal zone (SEZ). Labeled arrows for each incoming nerve indicate BMN projection direction. Scale bar, 10 μm. (**C**) Bristles on the anterior (left) and posterior (right) head that are innervated by BMNs in the nerve groups indicated by their color. *Figure 5—figure supplement 1* summarizes the EM reconstruction strategy. Sensory neurons that could not be assigned an identity are shown in *Figure 5—figure supplement 2*.

The online version of this article includes the following figure supplement(s) for figure 5:

**Figure supplement 1.** Reconstruction of mechanosensory neurons in different head nerves.

**Figure supplement 2.** Reconstructed sensory neurons that could not be assigned an identity (unknown sensory neurons).

*2022*), unidentified sensory neurons (*Figure 5—figure supplement 2A–Y*), and interneurons (not shown).

We identified 705 BMNs among the EM-reconstructed neurons by comparing their SEZ projection morphologies with light microscopy imaged BMNs (*Figure 3C and J*, *Figure 4C–V*). In agreement with the light microscopy data, the reconstructed BMNs project through different nerves into distinct zones in the SEZ (*Figure 5A and B*). For example, BMNs from the Eye- and LabNv have distinct ventral and anterior projections, respectively. This shows how the BMNs are somatotopically organized, as their distinct projections correspond to different bristle locations on the head (*Figure 5B and C*, see FlyWire.ai link 1 to view the BMN projections in three dimensions).

## Matching the reconstructed head BMNs with their bristles

The reconstructed BMN projections were next matched with their specific bristle populations. The projections were clustered based on morphological similarity using the NBLAST algorithm (example clustering at cut height 5 shown in *Figure 6—figure supplement 1A and B*, *Supplementary file 3*, FlyWire.ai link 2) (*Costa et al., 2016*). Clusters could be assigned as BMN types based on their similarity to light microscopy images of BMNs known to innervate specific bristles. 10 types were matched with dye-filled or MCFO-labeled BMNs (BM-InOc, -Oc, -Ant, -Or, -Vib, -Vt, -dPoOr, -dOcci, vOcci, and -Taste neurons, BM-InOc example shown in *Figure 6A*, all shown in *Figure 6—figure supplement 2A–M*). BM-MaPa neurons were matched using published images of labeled MaxNv

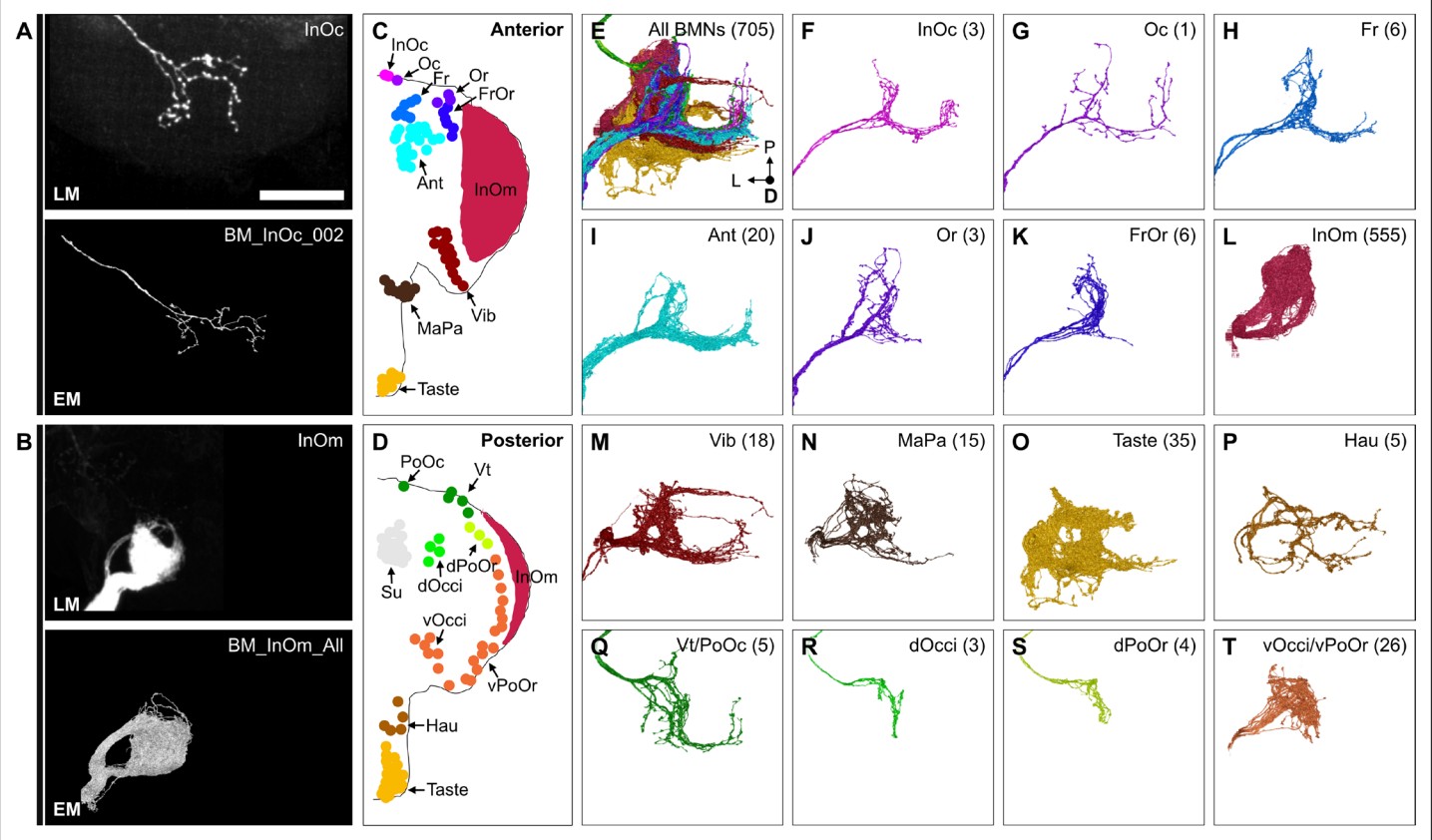

**Figure 6.** Bristle mechanosensory neuron (BMN) types that innervate specific head bristles. (**A–B**) Examples of matching light microscopy (LM) imaged BMN projections with their corresponding electron microscopy (EM)-reconstructed BMNs, including BM-InOc neurons (**A**) and BM-InOm neurons (**B**). Top panels show representative LM images of labeled BMNs that innervate the bristle indicated in the top right corner (anterior subesophageal zone [SEZ] views as shown in *Figure 4*). The individual BM-InOc neuron was labeled by dye filling using DiD while the collective projections of the BM-InOm neurons were labeled using the driver line InOmBMN-LexA expressing GFP. Bottom panels show the EM-reconstructed BMN types indicated in the top right corner. Shown is a representative example of a BM-InOc neuron (**A**) and all reconstructed BM-InOm neurons (**B**). Scale bar, 50 μm. Examples for all LM and EM matched BMNs are shown in *Figure 6—figure supplement 2*. Additional evidence used for assigning the different BMN types is shown in *Figure 6—figure supplement 1*, *Figure 6—figure supplement 3*, and *Figure 6—figure supplement 4*. (**C–D**) Different bristle populations indicated by labeled and colored dots are innervated by BMNs shown in **E–T**. The anterior (**C**) and posterior (**D**) head are shown. (**E–T**) Reconstructed SEZ projections of BMN types that are labeled and plotted in colors indicating the bristles that they innervate. Shown are the dorsal views of all BMNs (**E**), BM-InOc (**F**), BM-Oc (**G**), BM-Fr (**H**), BM-Ant (**I**), BM-Or (**J**), BM-FrOr (**K**), BM-InOm (**L**), BM-Vib (**M**), BM-MaPa (**N**), BM-Taste (**O**), BM-Hau (**P**), BM-Vt/PoOc (**Q**), BM-dOcci (**R**), BM-dPoOr (**S**), and BM-Occi/vPoOr (**T**) neurons. The number of reconstructed BMNs for each type is indicated.

The online version of this article includes the following figure supplement(s) for figure 6:

**Figure supplement 1.** NBLAST clustering of bristle mechanosensory neurons (BMNs).

**Figure supplement 2.** Matching electron microscopy (EM)-reconstructed bristle mechanosensory neuron (BMN) projections with light microscopy (LM) imaged BMNs that innervate specific bristles.

**Figure supplement 3.** Evidence used to match the electron microscopy (EM)-reconstructed bristle mechanosensory neurons (BMNs) with their bristles.

**Figure supplement 4.** Electron microscopy (EM) reconstruction of OcciNv bristle mechanosensory neurons (BMNs) from both brain hemispheres.

projections (*Naresh Singh and Nayak, 1985*). Four types were matched by comparison with BMNs innervating neighboring bristles that showed similar morphology (BM-Fr, -FrOr, -vPoOr, and -Hau neurons). Among these, the BM-vPoOr neurons were so morphologically similar to the MCFO matched BM-vOcci neurons that they could not be distinguished from each other, and were therefore treated as a single group (BM-vOcci/vPoOr neurons). The collective projections of the 555 reconstructed BM-InOm neurons were matched with BMNs labeled using the InOmBMN-LexA driver line (*Figure 6B*, *Figure 6—figure supplement 2N*). This matching involved combining 11 different NBLAST clusters

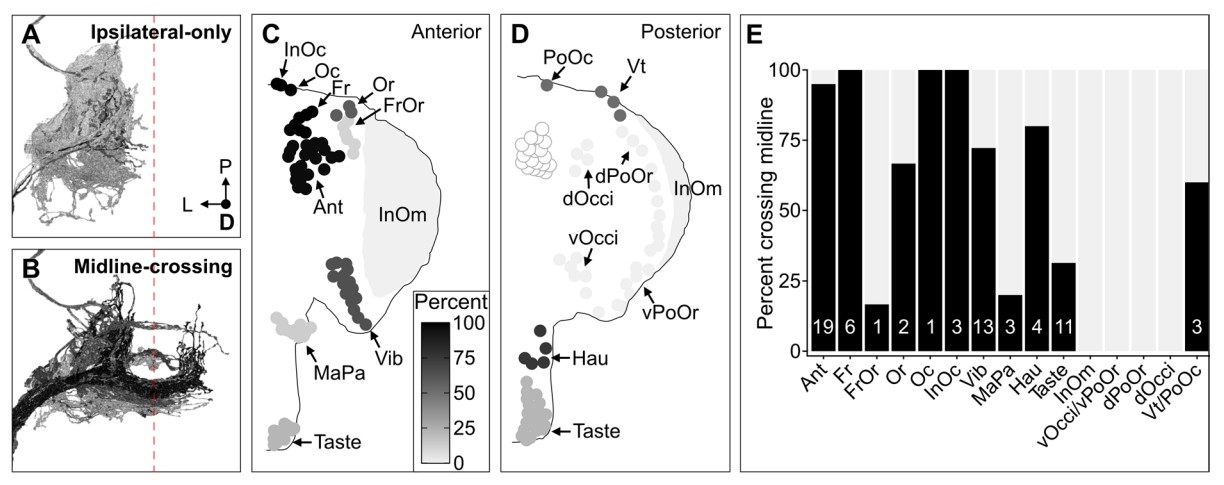

**Figure 7.** Some head bristle mechanosensory neurons (BMNs) have projections that cross the midline to the contralateral brain hemisphere. (**A–B**) BMNs that remain in the ipsilateral brain hemisphere (**A**) versus those with midline-crossing projections (**B**), shaded by percent midline-crossing for each type (scale in **C**). Red dashed line indicates the brain midline. (**C, D**) Shaded dots on the anterior (**C**) and posterior (**D**) head indicate the percent of BMNs innervating each bristle population that are midline-crossing. (**E**) Bar plots of midline-crossing percentages (numbers of midline-crossing BMNs indicated).

(*Figure 6—figure supplement 1A and B*) and revealed morphological diversity among the BM-InOm neurons.

Additional evidence was used to support our BMN-type assignments (*Figure 6—figure supplement 3A*), including a comparison of the morphology and numbers of reconstructed BMNs on both sides of the brain (for the small OcciNv, *Figure 6—figure supplement 4A–D*), and determining that BMNs of the same type show common postsynaptic connectivity (described below). Finally, we verified that the numbers of BMNs for each type were consistent with their corresponding bristle numbers (*Figure 6—figure supplement 3B–F*). This consistency of the BMN/bristle numbers, and completeness of sensory neuron proofreading in each nerve suggested that nearly all BMNs were reconstructed. Thus, we produced a near-complete brain projection map of 15 BMN types that innervate the different bristle populations on the head (*Figure 6C–T*, listed in *Supplementary file 3*).

## BMN somatotopic map

The projection map defined above revealed three features of somatotopic organization among the BMN types (*Figure 6E–T*, see FlyWire.ai link 3 to better view the BMN projections in three dimensions). First, each type has a unique branch morphology that defines its projections into distinct zones in the SEZ. Second, types that innervate neighboring bristle populations have branches that project into partially overlapping zones. For example, BMNs that innervate bristles on the dorsal head all have a common ipsilateral projection (*Figure 6F–K and Q*, lateral branch in each panel). In contrast, BMNs that innervate bristles at distant locations (e.g. dorsal and ventral head) show little or no projection overlap. Third, the projections of BMNs either remain in the ipsilateral brain hemisphere or cross the midline to the contralateral side, depending on the locations of their corresponding bristle populations (*Figure 7A–E*). That is, BMNs innervating populations located medially on the anterior head have midline-crossing projections, whereas BMNs innervating lateral, eye, and posterior head populations have ipsilateral-only projections. BMNs innervating bristles on the proboscis showed mixtures of ipsilateral-only and midline-crossing projections. These somatotopic features reveal how BMNs have distinct and overlapping SEZ projections that reflect their relative locations and proximities on the head.

The BMN somatotopic organization was further defined using NBLAST and connectomic data. NBLAST calculates similarity scores based on neuron morphology and spatial location (*Costa et al., 2016*). BMNs innervating neighboring bristle populations showed high similarity, indicating that their projections are morphologically similar and in close proximity (*Figure 6—figure supplement 1A and B*). We confirmed this close proximity through analysis of BMN/BMN interconnectivity. All neurons

in FlyWire.ai were previously linked to their corresponding automatically detected synapses in FAFB (**Buhmann et al., 2021**; **Dorkenwald et al., 2022**), which revealed that the BMN axons have both pre- and postsynaptic sites (**Figure 8—figure supplement 1A**). Analysis of all-to-all connectivity among the BMNs revealed that some of these sites corresponded to BMN/BMN synaptic connections (**Figure 8—figure supplement 1B**). The highest connectivity was among BMNs of the same type, but types innervating neighboring bristles were also connected. In contrast, BMNs innervating bristles at distant locations showed low NBLAST similarity and were not connected, consistent with these BMNs projecting into distinct zones. Interestingly, the different BMN projection zones defined by the NBLAST and connectivity data correspond roughly to the eye, ventral, dorsal, and posterior head.

## Somatotopically organized parallel BMN pathways

The map of somatotopically organized BMN projection zones provided evidence of the parallel sensory pathways predicted by the model of hierarchical suppression underlying grooming (**Hampel et al., 2017**; **Seeds et al., 2014**). In the model, mechanosensory neurons detect dust at different head locations and elicit aimed grooming through distinct postsynaptic circuits that function in parallel (**Figure 1—figure supplement 1A and C**). The projection zones could be where BMNs synapse with these circuits. Therefore, we examined the postsynaptic connectivity of the different BMN types to test if they form parallel connections with distinct partners. Nearly all neurons postsynaptic to the BMNs were first proofread in FlyWire.ai by our group and the wider proofreading community (**Dorkenwald et al., 2023**; **Dorkenwald et al., 2022**). We then compared the connectivity of the BMNs with their postsynaptic partners using cosine similarity-based clustering (**Figure 8—figure supplement 2**). The 555 BM-InOm neurons were excluded from this analysis because they were present in higher numbers and with fewer presynaptic sites than the 150 BMNs of other types, and clustering all BMNs together resulted in obscured clustering (presynaptic site counts in **Figure 8—figure supplement 1A**). The BMN/BMN connections shown in **Figure 8—figure supplement 1B** were also excluded from the cluster analysis.

Cosine similarity clustering revealed that BMNs formed parallel postsynaptic connections that reflected their head somatotopy. The lowest level clusters at the lowest cut heights shown in the **Figure 8A** dendrogram contained BMNs of the same type (colored bars next to dendrogram), demonstrating that BMNs innervating the same bristle populations had the highest connectivity similarity. Higher dendrogram cut heights (larger cluster sizes) uncovered connectivity similarity among BMN types innervating neighboring bristle populations. For example, a cut height of 4.5 identified five clusters that captured connectivity similarity among both same and neighboring BMN types (**Figure 8A and B**, colored circles 1–5, FlyWire.ai link 4). Clusters 1 and 4 contained exclusively BMNs of the same type, including BM-Vib (Cluster 1) and a subset of BM-Taste neurons (Cluster 4). The other subset of BM-Taste neurons is represented in Cluster 3, showing connectivity similarity with neighboring BMNs on the ventral head. This intratype differential clustering observed with the BM-Taste neurons was also found with other BMN types, including the morphologically diverse BM-InOm neurons (**Figure 8—figure supplement 3A and B**, FlyWire.ai link 5). Thus, while BMNs of the same type tend to show high connectivity similarity, we also find evidence that there are BMN subtypes with distinct postsynaptic partners. Clusters 2, 3, and 5 contained BMNs innervating neighboring bristle populations that were located roughly on the dorsal, ventral, and posterior head areas (**Figure 8A–C**). Clusters 2 and 3 contained exclusively dorsal or ventral BMNs, respectively, while Cluster 5 contained 77% posterior head BMNs and 23% anterior. The posterior and anterior BMNs in Cluster 5 showed relatively low postsynaptic connectivity similarity with each other (**Figure 8—figure supplement 2**), thus maintaining a mostly anterior versus posterior spatial distinction in their postsynaptic connectivity. However, the presence of both anterior and posterior head BMNs in Cluster 5 also indicates that some postsynaptic partners receive BMN inputs that are not head location specific.

Our results reveal head bristle proximity-based organization among the BMN projections and their postsynaptic partners to form parallel mechanosensory pathways. BMNs innervating neighboring bristles project into overlapping zones in the SEZ, whereas those innervating distant bristles project to distinct zones (example of BM-Fr, -Ant, and -MaPa neurons shown in **Figure 8D and E**). Cosine similarity analysis of BMN postsynaptic connectivity revealed that BMNs innervating the same bristle populations (same types) have the highest connectivity similarity. **Figure 8F** shows example parallel connections for BM-Fr, -Ant, and -MaPa neurons (vertical arrows), where the edge width indicates the

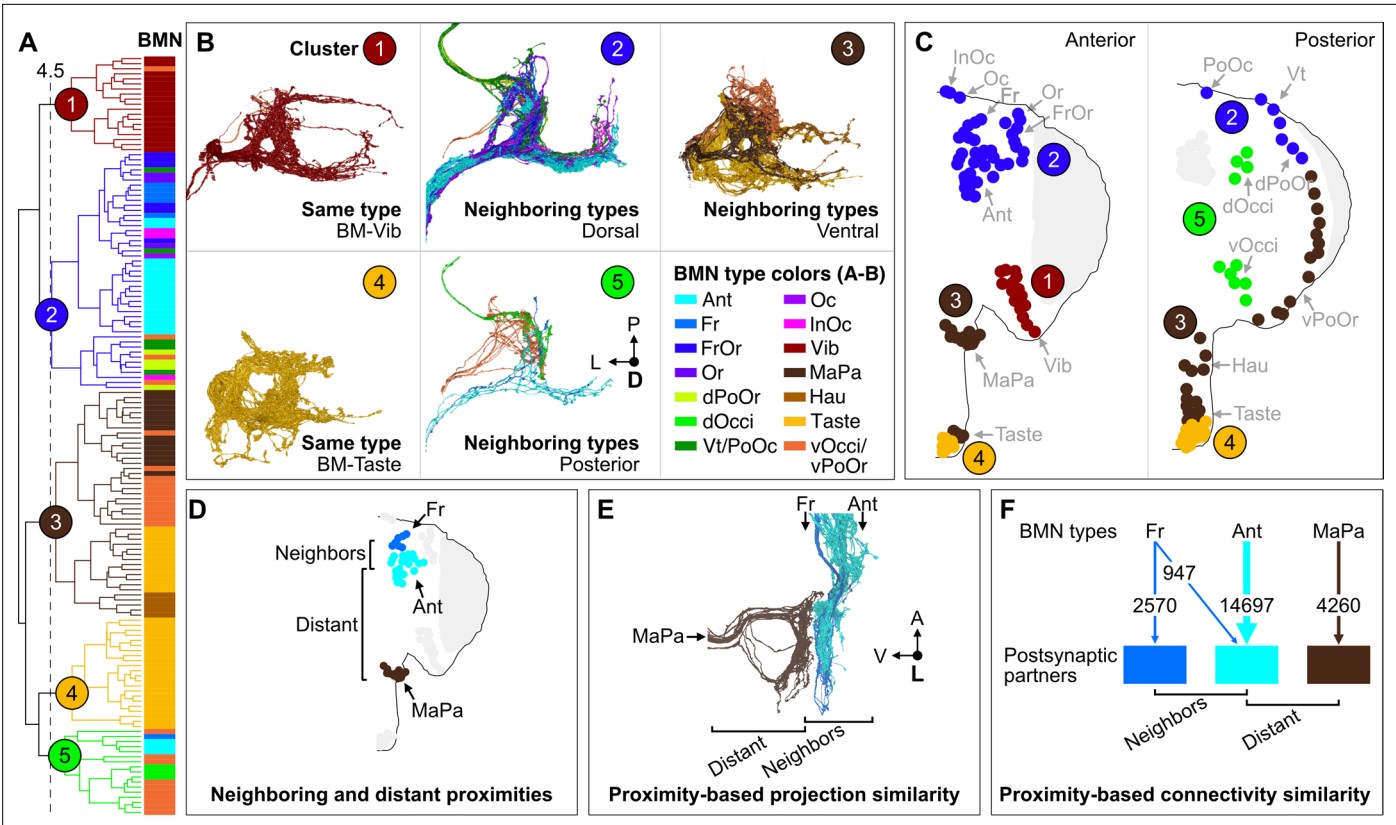

**Figure 8.** Somatotopy-based postsynaptic connectivity similarity among bristle mechanosensory neuron (BMN) types. (**A**) Dendrogram of cosine similarity clustering of BMNs by postsynaptic connectivity similarity. Analysis excludes postsynaptic partners with fewer than six synapses, and the BMN/ BMN connections shown in *Figure 8—figure supplement 1*. Individual BMNs are shown as bars and their types correspond to the colors indicated in **B** (bottom right). The five clusters are from cut height 4.5 on the dendrogram (dotted line) derived from the comparisons shown in *Figure 8— figure supplement 2*. (**B**) Morphologies of BMNs in the indicated clusters (upper right) whose types correspond to the colors shown in the bottom right. (**C**) Spatial relationships among the clustered BMNs are shown by coloring their bristles (dots) by cluster number on the anterior and posterior head. BMN types in more than one cluster are colored accordingly if at least 20% of that type was in a given cluster (e.g. BM-Taste neurons are in Clusters 3 [37%, brown] and 4 [63%, orange]). Note: the positioning of the colored dots indicating different clusters for Taste and Occi/PoOr bristles is hypothesized based on their proximity to other BMNs in the same cluster. The clusters exemplify different levels of connectivity similarity shown by the dendrogram (**A**). BMNs showing the highest connectivity similarity innervate the same bristle populations, as exemplified by BM-Vib (Cluster 1) and BM-Taste (Cluster 4) neurons. BMNs that innervate neighboring bristle populations also show high connectivity similarity, including BMNs on the dorsal (Cluster 2), ventral (Cluster 3), and posterior head (Cluster 5). Note: Cluster 5 consists mostly of posterior head BMNs, but also BM-Ant and -Fr neurons on the anterior head, although these BMNs show relatively low cosine similarity with the posterior head BMNs. BM-InOm neurons were analyzed separately (*Figure 8—figure supplement 3*). (**D–F**) Summary of BMN somatotopic features. (**D**) Different BMN types innervate bristles at neighboring and distant proximities. (**E, F**) BMNs that innervate neighboring bristles project into overlapping zones (**E**, example of electron microscopy (EM)-reconstructed BM-Fr and -Ant neuron subesophageal zone (SEZ) projections with non-overlapping -MaPa neuron projections) and can show postsynaptic connectivity similarity (**F**, edge widths based on number of total synapses from a given BMN type to its major postsynaptic partners, edges under 5% of BMN output omitted). Labeled arrows for each BMN type shown in **E** indicate projection direction.

The online version of this article includes the following figure supplement(s) for figure 8:

**Figure supplement 1.** Bristle mechanosensory neuron (BMN)-type synaptic counts and BMN/BMN connectivity.

**Figure supplement 2.** Cosine similarity clustering of bristle mechanosensory neuron (BMN) to non-BMN postsynaptic connectivity.

**Figure supplement 3.** Cosine similarity clustering of BM-InOm neurons in their connectivity with non-bristle mechanosensory neuron (BMN) postsynaptic partners.

number of synapses from each BMN type to their major postsynaptic partners. Additionally, BMNs innervating neighboring bristle populations showed postsynaptic connectivity similarity, while BMNs innervating distant bristles show little or none. For example, BM-Fr and -Ant neurons have connections to common postsynaptic partners, whereas BM-MaPa neurons show only weak connections with the main postsynaptic partners of BM-Fr or -Ant neurons (*Figure 8F*, connections under 5% of total

BMN output omitted). These results suggest that BMN somatotopy could have different possible levels of head spatial resolution, from specific bristle populations (e.g. Ant bristles) to general head areas (e.g. dorsal head bristles).

## Activation of subsets of head BMNs elicits aimed grooming of specific locations

We next tested the extent to which the parallel-projecting BMNs elicited aimed grooming of specific head locations. The driver lines described above (*Figure 3D–I*) were used to express the light-gated neural activator CsChrimson (*Klapoetke et al., 2014*) in different subsets of BMNs (*Figure 9A*). Flies were placed in chambers where they could move freely and then exposed to red light to activate the CsChrimson-expressing BMNs. We manually annotated the movements elicited by optogenetic activation of BMNs from recorded video (*Figure 9—figure supplement 1*, *Videos 1–4*).

Optogenetic activation of BMN types labeled by each driver line elicited grooming by the front legs that was aimed at specific head locations (*Figure 9B and C*). For example, a line that expressed in different BMN types on the dorsal head elicited aimed dorsal head grooming (dBMN-spGAL4; BM-InOc, -Vt, and -dPoOr neurons, blue trace, *Video 1*). Two lines expressed exclusively in specific BMN types, which enabled us to test the extent to which grooming was aimed specifically at those BMNs (i.e. BM-Taste and -InOm neurons). Indeed, BM-Taste neurons on the labellum elicited labellar grooming, but also grooming of neighboring locations on the proboscis and ventral head (TasteBMN-spGAL4, yellow trace includes proboscis and ventral head grooming, *Video 2*). Activation of BM-InOm neurons (InOmBMN-LexA, *Video 3*) elicited eye grooming (red trace), but also grooming of the neighboring dorsal head (blue trace). This suggested that head BMNs elicit aimed grooming of their corresponding bristle locations, but also neighboring locations. This result is consistent with our anatomical and connectomic data indicating that BMNs innervating neighboring bristles show overlapping projections and postsynaptic connectivity similarity (see Discussion).

Activation of BMNs on the posterior head elicited low levels of dorsal head grooming (blue trace), but mostly a forward head nodding movement (*Figure 9A–C*, pBMN-spGAL4; BM-Vt, -dOcci, -dPoOr, and -vOcci neurons, *Video 4*). Nodding was an apparent avoidance response to posterior touches of the head, and occurred while the flies either stood in place or walked around. However, nodding was also observed during dorsal head grooming. Such nodding movements during head grooming were previously shown to help the legs reach particular locations (*Honegger et al., 1979*). Nodding also occurred with the dorsal head grooming elicited using the dBMN-spGAL4 and InOmBMN-LexA driver lines, but these lines did not elicit nodding in the absence of grooming as we observed with pBMN-spGAL4. This suggested that BMN-activated nodding occurs in two different behavioral contexts: during dorsal head grooming and as an avoidance response. Different evidence led us to hypothesize that nodding in these contexts was elicited by distinct BMN types. First, pBMN- and dBMN-spGAL4 driver lines show overlapping expression in BM-Vt and -dPoOr neurons, and both elicit dorsal head grooming accompanied by nodding. Second, pBMN-spGAL4 is the only tested line that expressed in BM-dOcci and -vOcci neurons and also the only line that elicited nodding in the absence of grooming. When taken together, our experiments suggest that nodding-only movements are elicited by BM-dOcci and -vOcci neurons and dorsal head grooming is elicited by BM-InOc, -Vt, -dPoOr, and -InOm neurons.

In addition to grooming, BMNs on the dorsal head and eyes elicited backward motions that appeared as if flies were avoiding something that touched the head (dBMN-spGAL4 and InOmBMN-LexA). The backward motion and grooming were mutually exclusive and sequential, as the backward motion occurred transiently at the stimulus onset and was followed by grooming. As we reported previously (*Hampel et al., 2020a*), the red-light stimulus also elicited backward motions with control flies (*Figure 9B and C*, control, black trace, *Video 5*). However, control flies only responded in 33% of trials, whereas BMN activation flies responded with backward motions in most trials (73% for dorsal head BMNs, 100% BM-InOm neurons). Taken together, this study reveals that the somatotopically organized head BMNs elicit both aimed grooming and avoidance-like responses.

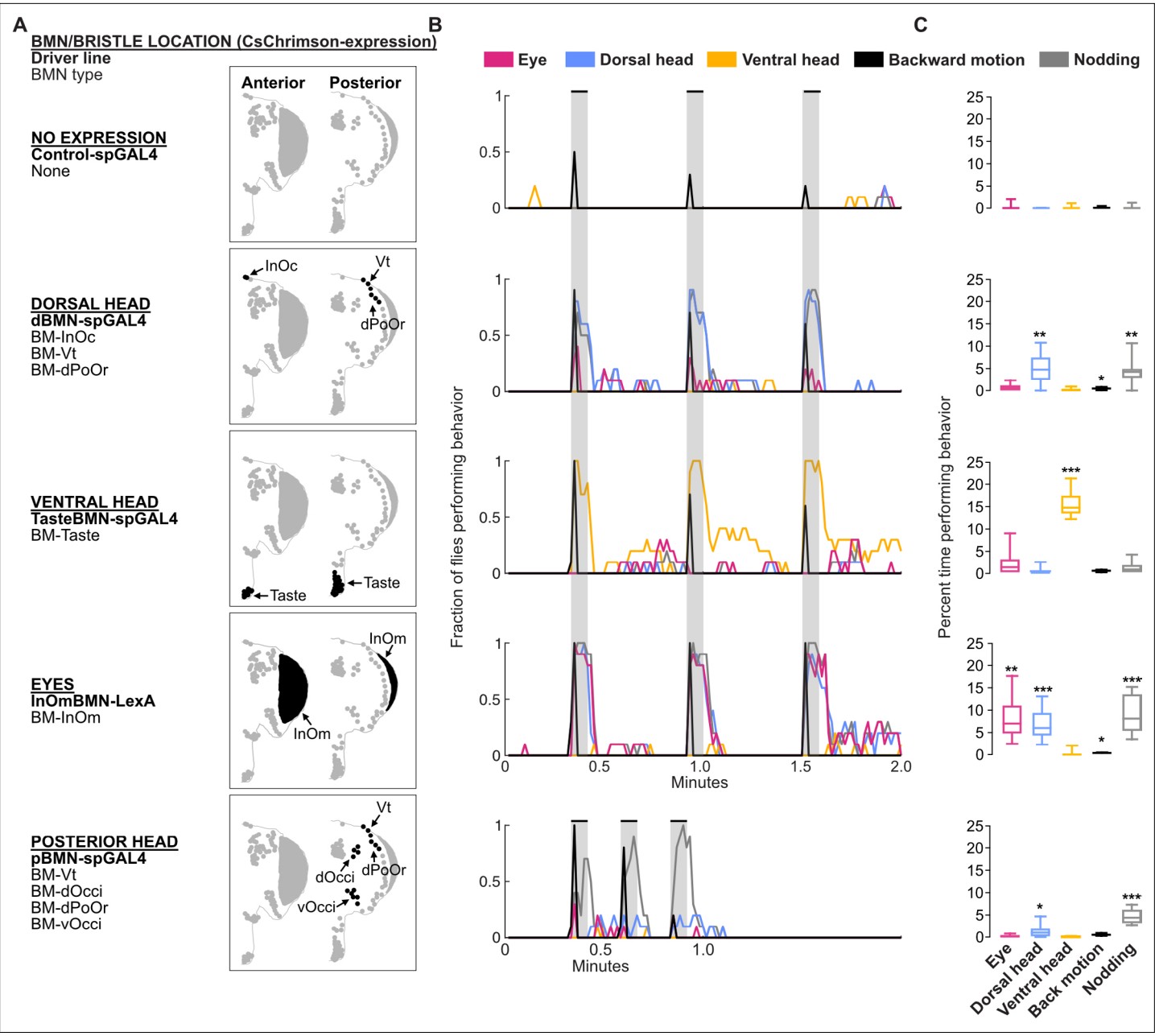

**Figure 9.** Optogenetic activation of bristle mechanosensory neurons (BMNs) at specific head locations elicits aimed grooming. (**A**) Bristles shaded black on the anterior (left) and posterior (right) head are innervated by BMNs that express CsChrimson under control of the indicated driver lines. Control-spGAL4 shows no expression. (**B**) Histograms of manually annotated video for each line show movements elicited with red-light-induced optogenetic activation. The fraction of flies performing each movement are plotted in 1 s bins (N=10 flies per line). Grooming movements are indicated by different colors, including eye (magenta), dorsal head (blue), and ventral head (orange) grooming. Other elicited movements include backward motion (black) and head nodding (gray). Gray bars indicate a 5 s red-light stimulus. Most driver lines were tested using 30 s interstimulus intervals, while pBMN-spGAL4 elicited more reliable behavior using 10 s intervals. Movements are mutually exclusive except head nodding. Representative experimental trials shown in *Video 1*, *Video 2*, *Video 3*, *Video 4*, and *Video 5*. *Figure 9—figure supplement 1* shows additional controls and ethograms for individual flies tested. (**C**) Box plots show the percent time that flies spent performing each movement during the experiment shown in **B**. Bottom and top of the boxes indicate the first and third quartiles, respectively; median is shown in each box; whiskers show the minimum and maximum values. Asterisks indicate *p, 0.05, **p, 0.001, ***p, 0.0001 from Mann-Whitney U pairwise tests between each experimental line and its corresponding control after application of Bonferroni correction. *Figure 9—source data 1* contains numerical data used for producing each box plot.

The online version of this article includes the following source data and figure supplement(s) for figure 9:

**Source data 1.** Numerical data used for producing each box plot.

**Figure supplement 1.** Ethograms of movements performed with activation of different bristle mechanosensory neurons (BMNs).

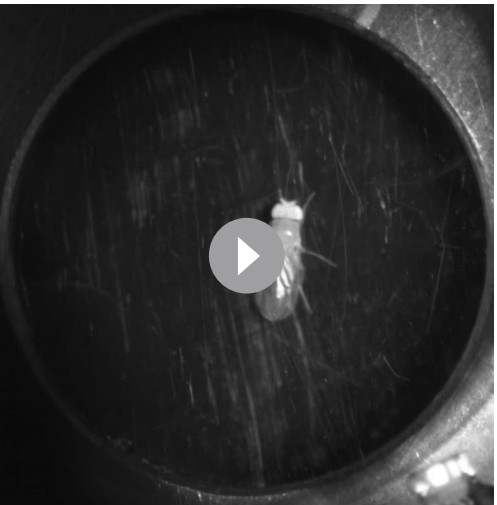

**Video 1.** Optogenetic activation of dorsal head bristle mechanosensory neurons (BMNs) elicits aimed dorsal head grooming. CsChrimson was expressed in BMNs targeted by the dBMN-spGAL4 driver line. Infrared light in the bottom right corner indicates when the red light was on to activate the targeted BMNs. Note that head nodding movements and backward motions are also elicited.

https://elifesciences.org/articles/87602/figures#video1

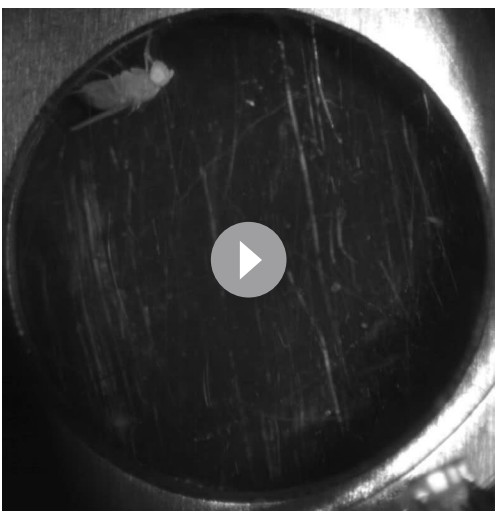

**Video 2.** Optogenetic activation of BM-Taste neurons elicits aimed proboscis and ventral head grooming. CsChrimson was expressed in bristle mechanosensory neurons (BMNs) targeted by the TasteBMN-spGAL4 driver line. Infrared light in the bottom right corner indicates when the red light was on to activate the targeted BMNs.

https://elifesciences.org/articles/87602/figures#video2

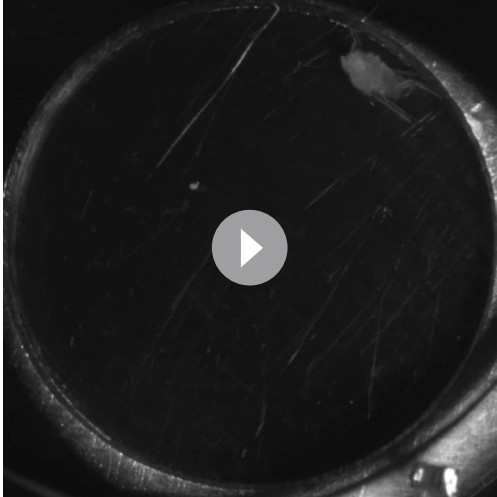

**Video 3.** Optogenetic activation of BM-InOm neurons elicits eye and dorsal head grooming. CsChrimson was expressed in bristle mechanosensory neurons (BMNs) targeted by the InOmBMN-LexA driver line. Infrared light in the bottom right corner indicates when the red light was on to activate the targeted BMNs. Note that head nodding movements and backward motions are also elicited.

https://elifesciences.org/articles/87602/figures#video3

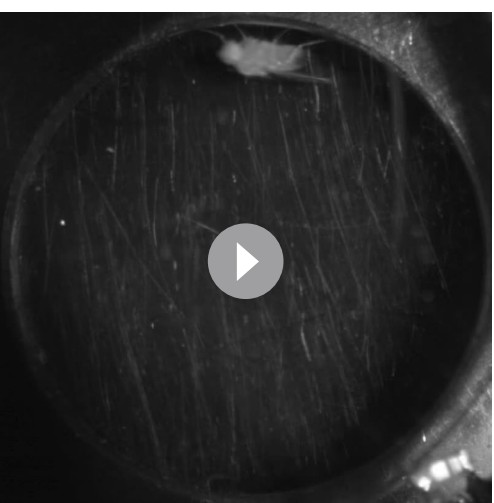

**Video 4.** Optogenetic activation of posterior head bristle mechanosensory neurons (BMNs) elicits head nodding. CsChrimson was expressed in BMNs targeted by the pBMN-spGAL4 driver line. Infrared light in the bottom right corner indicates when the red light was on to activate the targeted BMNs. Note that dorsal head grooming movements are also elicited (not shown in video).

https://elifesciences.org/articles/87602/figures#video4

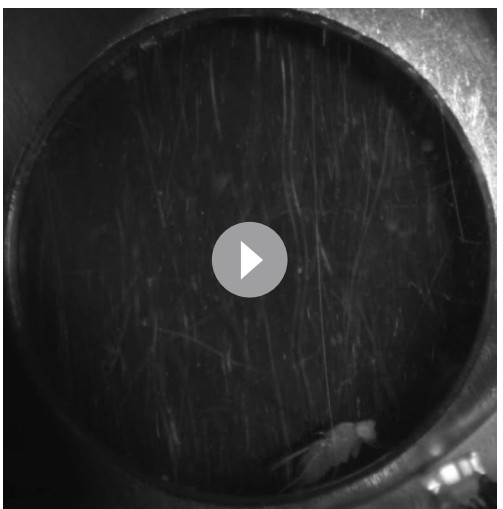

**Video 5.** Optogenetic stimulus in control flies. Control fly was exposed to the same red-light stimulus shown in *Videos 1–3*. The infrared light in the bottom right corner indicates when the red light was on.
https://elifesciences.org/articles/87602/figures#video5

## Discussion

### Comprehensive definition of head BMNs

A major outcome of this work was the definition of nearly all BMNs on the *Drosophila* head. Although there were previous descriptions of the BMNs from different body parts, there were no comprehensive descriptions of all BMNs for any part. Furthermore, the head BMNs were among the least well described. Here, we modified a previously reported BMN dye fill method and produced new transgenic driver lines to define the projection morphologies of the different head BMN types that innervate specific bristle populations on the head. We then identified and reconstructed these types in the FAFB EM dataset. This provides the most comprehensive definition of the BMNs for any body part of *Drosophila* (or any other insect), and an essential resource for future studies. The annotated neurons can be linked to the ongoing neural circuit reconstructions in FAFB (*Dorkenwald et al., 2023*; *Dorkenwald et al., 2022*), or identified in anticipated new EM reconstructions of the brains of other individuals using

available and emerging tools (*Galili et al., 2022*).

While nearly all head BMNs were reconstructed in this work, different knowledge gaps remain. First, it is unclear if the PoOc and Su bristles are innervated by BMNs because they could not be observed using transgenic driver lines or dye filling methods. We proposed that one of the BM-Vt/ PoOc neurons innervates the PoOc bristle, based on proximity and presumed morphological similarity to the neighboring BM-Vt, -InOc, and -Oc neurons. For the Su bristles, one possibility is that they are innervated by some of the 25 unknown sensory neurons reconstructed in this work (*Figure 5— figure supplement 2A–Y*). Second, it remains unclear what neurotransmitter(s) are used by the BMNs. A machine learning approach was recently developed that can predict whether a neuron in FAFB uses any of six major neurotransmitters with high accuracy (*Eckstein et al., 2020*). Given that the neurotransmitter predictions for the BMNs were overwhelmingly cholinergic (not shown), and a previous study indicated that leg BMNs are sensitive to a nicotinic acetylcholine receptor antagonist (*Tuthill and Wilson, 2016a*), the parsimonious explanation is that the BMNs are cholinergic. However, other studies suggest that BMNs could use histamine as a neurotransmitter (*Melzig et al., 1996*; *Salvaterra and Kitamoto, 2001*; *Yasuyama and Salvaterra, 1999*). Thus, the extent to which head BMNs use acetylcholine, histamine, or other neurotransmitters remains unresolved.

### Resource: nearly all head mechanosensory neurons reconstructed and annotated in FAFB

In conjunction with two previous studies, work presented here contributes to the FAFB reconstruction and annotation of neurons associated with the major head mechanosensory structures, including the bristles (BMNs), JO (JONs), and taste pegs (TPMNs). The BMNs and TPMNs were reconstructed using the FlyWire.ai platform in the present work, and the JONs were previously reconstructed using the CATMAID platform (*Hampel et al., 2020a*; *Kim et al., 2020*). The TPMNs (38 reconstructed) respond to tactile displacements of the taste pegs and are implicated in feeding behavior (*Jeong et al., 2016*; *Sánchez-Alcañiz et al., 2017*; *Zhou et al., 2019*). Subpopulations of ~480 JONs have been previously defined that respond to diverse mechanical forces that move the antennae (JO-A, -B, -C, -D, -E, -F, and -mz neurons), including sound, gravity, wind, and tactile displacements (*Hampel et al., 2015*; *Ishikawa et al., 2017*; *Kamikouchi et al., 2009*; *Mamiya and Dickinson, 2015*; *Matsuo et al., 2014*; *Patella and Wilson, 2018*). The JONs are implicated in different behaviors including courtship, flight,

locomotion, gravitaxis, wind-guided orientation, escape, and head grooming (*Hampel et al., 2015*; *Kamikouchi et al., 2009*; *Lehnert et al., 2013*; *Mamiya et al., 2011*; *Mamiya and Dickinson, 2015*; *Suver et al., 2019*; *Tootoonian et al., 2012*; *Vaughan et al., 2014*; *Yorozu et al., 2009*). The reconstruction and annotation of head mechanosensory neurons in FAFB provides an important resource for connectomics-based studies of mechanosensory processing (*Supplementary file 3*). While the majority of mechanosensory neurons on the head are now identified in FAFB, some remain unknown, such as multidendritic and pharyngeal mechanosensory neurons on the proboscis (*Yang et al., 2021*; *Zhang et al., 2016*).

The reconstructed JONs, BMNs, and TPMNs project into distinct regions in the SEZ (*Figure 5—figure supplement 1A and E*), and therefore show modality-specific projections. For example, the JONs (chordotonal neurons) define a region of the SEZ called the antennal mechanosensory and motor center while the BMNs project more ventrally. While the BMN projections are based on head location (somatotopic), the JON projections are based on mechanical stimulus modality, such as their responses to vibrational or tonic antennal movements (tonotopic) (*Hampel et al., 2020a*; *Kamikouchi et al., 2006*; *Kim et al., 2020*; *Patella and Wilson, 2018*). However, there are potential overlapping projections between the most ventral projecting JONs (JO-F neurons) and some BMNs projecting through the AntNv (*Figure 5—figure supplement 1*). Among the BMNs that appear to overlap with the JO-F neurons are the BM-Ant neurons that are located on the same antennal segment as the JONs (pedicel). This overlap suggests that the JO-F neuron projections are somatotopic like the BMNs. Modality-specific mechanosensory projections are also reported in the VNC of *Drosophila* and other insects (e.g. BMNs, hair plates, campaniform sensilla, and chordotonal neurons), revealing this organization to be fundamental in insects (*Merritt and Murphey, 1992*; *Murphey et al., 1989a*; *Phelps et al., 2021*; *Smith and Shepherd, 1996*; *Tsubouchi et al., 2017*; *Tuthill and Wilson, 2016a*).

## A synaptic resolution somatotopic map of the head BMNs

This work defines the somatotopic organization of the head BMNs. Somatotopy was previously reported for BMNs innervating bristles on the bodies of *Drosophila* and other insects (*Johnson and Murphey, 1985*; *Murphey et al., 1989b*; *Newland, 1991*; *Newland et al., 2000*; *Tsubouchi et al., 2017*). However, these studies only produced partial somatotopic maps using dye fills or transgenic driver lines. Furthermore, there were no previous descriptions of somatotopy among the head BMNs. Here, we use EM reconstructions to produce a comprehensive synaptic resolution somatotopic map of head BMNs in the same brain.

All reconstructed *Drosophila* head BMN types terminate their projections in the SEZ. This indicates that the first layers of BMN processing for the head occur in the SEZ. In contrast, head BMNs reported in other insects project into both the SEZ and thoracic ganglia, including BMNs innervating the InOm bristles of the praying mantis and cricket (*Honegger, 1977*; *Zack and Bacon, 1981*) and wind-sensitive head bristles of the locust (*Tyrer et al., 1979*).

Head BMNs that innervate the same bristle populations (same types) project into the same zones in the SEZ, show the highest morphological similarity, and their morphology is stereotyped across individual flies. These characteristics likely apply to most BMNs, as numerous studies have identified the stereotyped projections of BMNs innervating specific bristles on the bodies of *Drosophila* and other insects (*Burg and Wu, 1986*; *Burg and Wu, 1989*; *Burg et al., 1993*; *Chen et al., 2006*; *Ghysen, 1980*; *Honegger, 1977*; *Kays et al., 2014*; *Murphey et al., 1989b*; *Zack and Bacon, 1981*). Head BMNs of the same type also show the highest postsynaptic connectivity similarity. However, some BMN types fall into multiple different NBLAST and cosine similarity clusters, revealing that there are BMN subtypes with differing morphology and postsynaptic connectivity. One notable example of such intratype diversity are the BM-InOm neurons that show differential clustering (*Figure 8—figure supplement 3*). This could reflect the large surface area of the eyes that spans from the dorsal to ventral head, and the differentially clustered BM-InOm neurons may innervate bristles at different locations on the eyes. Future studies will address the organizational and functional logic of such intratype diversity.

We also find that BMN types innervating neighboring bristle populations have overlapping projections (example shown in *Figure 8E*) into zones that correspond roughly to the dorsal, ventral, and posterior head. The overlap is likely functionally significant, as cosine similarity analysis revealed that neighboring head BMN types can have common postsynaptic partners (example shown in *Figure 8F*).

However, overlap between neighboring BMN types is only partial, as they show differing projections and postsynaptic connectivity. The extent of overlap likely reflects the proximity between bristles and enables postsynaptic partners to respond to mechanosensory stimulations of neighboring bristles whose corresponding BMNs are likely to show correlated activity (*Tuthill and Wilson, 2016b*). BMN projection overlap has also been observed with other parts of the body in *Drosophila*. For example, BMNs innervating bristles on the anterior and posterior leg compartments show overlapping projections in the VNC leg neuromere anterior and posterior zones, respectively (*Murphey et al., 1989b*). Similarly, BMNs innervating neighboring bristles on the thorax show overlap in their projections into the accessory mesothoracic neuropil (*Ghysen, 1980*; *Kays et al., 2014*). This overlap may have implications for aimed grooming behavior. For example, neighboring BMNs could connect with common neural circuits to elicit grooming of overlapping locations (discussed more below).

The somatotopic map reveals that some head BMNs have projections that remain in the ipsilateral brain hemisphere, while others have midline-crossing projections to the contralateral hemisphere (*Figure 7A–E*). Interestingly, BMNs innervating bristles located medially on the anterior head show midline-crossing projections, whereas those innervating more lateral populations have ipsilateral-only projections. Previous studies found that BMNs innervating medial bristles on the thorax have midline-crossing projections, while those innervating more lateral bristles have ipsilateral-only projections (*Ghysen, 1980*; *Kays et al., 2014*). Similarly, BMNs that innervate bristles located on the leg segment most medial to the body (coxa) have midline-crossing projections (*Murphey et al., 1989b*; *Phelps et al., 2021*). This is also the case for BMNs on the legs of other insects, such as the cricket (*Johnson and Murphey, 1985*) and hawkmoth (*Kent and Levine, 1988*). Why do some BMNs have ipsilateral and midline-crossing projections? One possibility is that these BMNs can excite postsynaptic circuitry in both brain hemispheres to elicit bilateral leg grooming responses, which could be appropriate for medial stimuli. In contrast, BMNs on the proboscis have mixtures of ipsilateral-only and midline-crossing projections, while those on the posterior head show ipsilateral-only projections. Thus, the organizational logic of midline-crossing BMNs described above may not be universal.

## First synaptic resolution somatotopic map of the head

This work provides the first synaptic resolution somatotopic map of a head (or body) for any species. Previous studies identified somatotopic maps across species, such as the vertebrate maps of head and body (*Abraira and Ginty, 2013*; *Adibi, 2019*; *Brown et al., 1977*). Somatotopic organization has been found to be preserved at different layers of the nervous system and is thought to be of fundamental importance, although the full functional significance of this organization is unclear (*Kaas, 1997*; *Thivierge and Marcus, 2007*). Therefore, it remains important to produce anatomical and functional maps and define how this somatotopy interfaces with postsynaptic circuits. It has previously not been possible to obtain a comprehensive description of a somatotopic map, as most studies were limited to sparse labeling experiments and extrapolation across different animals. Thus, the spatial relationships among mechanosensory neurons that make up particular maps could not be definitively determined. We overcame this through the first complete EM reconstruction of a somatotopic map of a head in the same brain. This enables future work that will define the postsynaptic connectome of this complete map. Thus, the synaptic resolution map provided here has important implications for expanding our understanding of somatotopic neural circuit organization and function.

## Circuits that elicit aimed grooming of specific head locations

We report here that activation of the head BMNs elicits aimed grooming. Flies groom specific head locations, including the eyes, antennae, dorsal head, ventral head, and proboscis (*Dawkins, 1976*; *Hampel et al., 2015*; *Hampel et al., 2017*; *Seeds et al., 2014*; *Szebenyi, 1969*; *Zhang et al., 2020*). With the exception of the BM-InOm neurons, little was known about the roles of the other BMNs in eliciting head grooming. The BM-InOm neurons were originally identified as necessary for grooming in response to mechanical stimulation of the eyes in the praying mantis (*Zack and Bacon, 1981*). Mechanical stimulation of the *Drosophila* InOm bristles (*Melzig et al., 1996*) and optogenetic activation of the BM-InOm neurons (*Hampel et al., 2017*; *Zhang et al., 2020*) were later reported to elicit eye grooming. Here, we used optogenetic activation to further define the movements elicited by BM-InOm neurons, and show that other BMN types elicit grooming of the dorsal and ventral head. Previous studies in *Drosophila* and other insects showed that stimulations of bristles on the legs,

wings, and thorax also elicit aimed grooming (*Corfas and Dudai, 1989*; *Li et al., 2016*; *Matheson, 1997*; *Page and Matheson, 2004*; *Usui-Ishihara et al., 1995*; *Vandervorst and Ghysen, 1980*). Thus, the BMNs are important for eliciting aimed grooming of specific locations on the head and body.

While we show that the parallel-projecting head BMNs elicit grooming of specific locations (i.e. eyes, dorsal, and ventral head), the full range of aimed grooming movements that can be elicited was not explored. For example, antennal grooming was previously shown to be elicited by JON activation (*Hampel et al., 2015*; *Zhang et al., 2020*), and we hypothesize here that BM-Ant neuron activation also elicits antennal grooming. However, we did not identify a transgenic driver line that labels BM-Ant neurons that would enable us to test this hypothesis. Previous studies of the legs, wings, and thorax used mechanical stimulation of specific bristles, rather than BMN optogenetic activation to test the ranges of grooming movements that could be elicited. This was done by delivering mechanical stimuli directly to the bristles of decapitated flies that do not move unless stimulated. In contrast, stimulating the head bristles is relatively challenging, as it requires delivering precise mechanical stimulations to specific bristles in intact and tethered flies. We have used optogenetic analysis in this study, as it was previously demonstrated that BMN optogenetic activation elicits grooming that is comparable to mechanically stimulating their corresponding bristles (*Hampel et al., 2017*; *Zhang et al., 2020*). However, our ability to test the full range of grooming movements elicited with BMN activation was limited by the driver lines produced in this study (*Figure 3D–G*).

How do the parallel-projecting head BMNs interface with postsynaptic neural circuits to elicit aimed grooming of specific head locations? Different evidence supports the hypothesis that the BMNs connect with parallel circuits that each elicit a different aimed grooming movement (*Seeds et al., 2014*). First, cosine similarity analysis revealed parallel connectivity at the first postsynaptic layer. However, this analysis revealed partial convergence of neighboring BMN types onto common postsynaptic partners. Thus, the resolution of the hypothesized parallel circuits and the specificity of the aimed grooming that they elicit remains to be determined. Second, previous studies showed that optogenetic activation of different sensory and interneuron types elicits grooming of specific head locations, suggesting that they are components of putative parallel circuits (*Cande et al., 2018*; *Guo et al., 2022*; *Hampel et al., 2015*; *Seeds et al., 2014*; *Zhang et al., 2020*). Third, we identified different neuron types whose activation elicit grooming of the antennae and showed that they are connected to form a neural circuit (*Hampel et al., 2020b*; *Hampel et al., 2015*). The inputs to this circuit are JONs that detect tactile stimulations of the antennae and project to the SEZ where they excite two interneuron types (aBN1 and aBN2) and a descending neuron type (aDN) to elicit grooming. The aDNs project to a zone in the VNC where circuitry for generating antennal grooming leg movement patterns is thought to reside (*Berkowitz and Laurent, 1996*). While this circuit is postsynaptic to the JONs (*Hampel et al., 2020b*; *Hampel et al., 2015*), preliminary connectomic analysis reveals that it is also postsynaptic to BMNs (not shown). Future studies will define the BMN connectivity with the antennal grooming circuit, and other neurons (and circuits) whose activation elicit aimed grooming of different head locations.

We find that activation of specific BMN types elicits both aimed grooming of their corresponding bristle locations and neighboring locations. This suggests overlap in the locations that are groomed with the activation of different BMN types. Such overlap provides a means of cleaning the area surrounding the stimulus location. Interestingly, our NBLAST and cosine similarity analysis indicates that neighboring BMNs project into overlapping zones in the SEZ and show common postsynaptic connectivity. Thus, we hypothesize that neighboring BMNs connect with common neural circuits (e.g. antennal grooming circuit) to elicit overlapping aimed grooming of common head locations.

## BMN involvement in multiple distinct behaviors

In addition to grooming, this work identifies other movements that are elicited by the head BMNs and their corresponding bristles. Previous studies implicated the InOm bristles in an avoidance response (*Melzig et al., 1996*), although this response was not described in detail. Here, we demonstrate that activation of the BM-InOm neurons elicits an avoidance-like response in the form of backward motions. This response was also elicited by activating BMN types on the dorsal head. Another putative avoidance-like behavior, head nodding, was found to be elicited by posterior head BMNs. Avoidance responses to bristle stimulation have been previously reported in *Drosophila* and other insects, such as limb withdrawal and postural changes (*Melzig et al., 1996*; *Burrows and Newland, 1997*; *Pflüger,*

*1980*; *Vandervorst and Ghysen, 1980*). Thus, BMNs across the head and body elicit grooming and possibly avoidance responses.

## Parallel circuit architecture underlying the grooming sequence

This study examines the mechanosensory layer of the parallel model of hierarchical suppression that produces the head to body grooming sequence (*Hampel et al., 2017*; *Mueller et al., 2019*; *Seeds et al., 2014*). This layer consists of mechanosensory neurons at specific locations on the head and body that elicit aimed grooming of those locations (*Hampel et al., 2020a*; *Hampel et al., 2017*; *Hampel et al., 2015*; *Zhang et al., 2020*). The aimed movements are performed in a prioritized sequence when mechanosensory neurons detect dust at different locations and become simultaneously activated (i.e. head and body completely dirty). In support of this, simultaneous optogenetic activation of mechanosensory neurons across the head and body elicits a grooming sequence that resembles the dust-induced sequence (*Hampel et al., 2017*; *Zhang et al., 2020*). Among the different mechanosensory neurons, the BMNs are particularly important, as their activation alone is sufficient to elicit a grooming sequence (*Zhang et al., 2020*). Thus, activation of individual BMN types elicits aimed grooming, while their simultaneous activation elicits a sequence.

Here, we define the parallel architecture of BMN types that elicit the head grooming sequence that starts with the eyes and proceeds to other locations, such as the antennae and ventral head. The different BMN types are hypothesized to connect with parallel circuits that elicit grooming of specific locations (described above and shown in *Figure 1—figure supplement 1A and C*). Indeed, we identify distinct projections and connectivity among BMNs innervating distant bristles on the head, providing evidence supporting this parallel architecture (*Figure 8D–F*). However, we also find partially overlapping projections and connectivity among BMNs innervating neighboring bristles. Further, optogenetic activation of BMNs at specific head locations elicits grooming of both those locations and neighboring locations (*Figure 9*). These findings raise questions about the resolution of the parallel architecture underlying grooming. Are BMN types connected with distinct postsynaptic circuits that elicit aimed grooming of their corresponding bristle populations (e.g. Ant bristles)? Or are neighboring BMN types that innervate bristles in particular head areas connected with circuits that elicit grooming of those areas (e.g. dorsal or ventral head)? Future studies of the BMN postsynaptic circuits will be required to define the resolution of the parallel pathways that elicit aimed grooming.

The parallel-projecting head BMNs are also hypothesized to connect with postsynaptic circuits that perform additional functions to produce the sequence (*Seeds et al., 2014*). Simultaneous activation of the parallel architecture by dust causes competition among all movements to be performed in the sequence. This competition is resolved through a hierarchical suppression mechanism whereby earlier movements suppress later ones (*Figure 1—figure supplement 1B*). Performance order is established by an activity gradient among the parallel circuits where earlier movements have the highest activity and later ones have the lowest. This gradient was proposed to be produced by controlling sensory gain among the BMNs, or through putative lateral inhibitory connections between the parallel circuits. A winner-take-all network selects the movement with the highest activity and suppresses the others. Our work here provides the foundation for studies that will examine how the BMN postsynaptic circuitry is organized to drive these different functions and produce the grooming sequence.

The BMNs are hypothesized to have roles in both eliciting and terminating the different movements in the grooming sequence through dust detection (*Hampel et al., 2017*; *Seeds et al., 2014*; *Zhang et al., 2020*). That is, dust on particular body parts would be detected by BMNs that are activated with displacement of their corresponding bristles and elicit aimed grooming. While a completely dirty body part would cause strong BMN activation, the level of activation would decrease as a consequence of the decreased dust levels that occur with grooming. This reduced activity would terminate the selected movement in the sequence, allowing a new round of competition among the remaining movements and selection of the next movement through hierarchical suppression. However, it has not been directly demonstrated that the BMNs elicit or terminate the sequence through dust sensing. For example, blocking BM-InOm neurons does not reduce dust-induced grooming of the head (*Zhang et al., 2020*). However, this may be due to compensation from neighboring mechanosensory neurons. Thus, the presumed role of the BMNs in detecting dust remains to be directly demonstrated.

# Materials and methods

**Key resources table**

| Reagent type (species) or resource | Designation | Source or reference | Identifiers | Additional information |
|---|---|---|---|---|
| Genetic reagent (*D. melanogaster*) | R52A06-GAL4 | *Jenett et al., 2012* | RRID:BDSC_38810 | |
| Genetic reagent (*D. melanogaster*) | VT017251-LexA | *Hampel et al., 2017* | | aka InOmBMN-LexA |
| Genetic reagent (*D. melanogaster*) | VT019023-AD | *Tirian and Dickson, 2017* | RRID:BDSC_71430 | |
| Genetic reagent (*D. melanogaster*) | VT050279-DBD | *Tirian and Dickson, 2017* | RRID:BDSC_72433 | |
| Genetic reagent (*D. melanogaster*) | R28D07-AD | *Dionne et al., 2017* | RRID:BDSC_70168 | |
| Genetic reagent (*D. melanogaster*) | VT023783-AD | *Tirian and Dickson, 2017* | RRID:BDSC_73261 | |
| Genetic reagent (*D. melanogaster*) | R11D02-DBD | *Dionne et al., 2017* | RRID:BDSC_68554 | |
| Genetic reagent (*D. melanogaster*) | dBMN-spGAL4 | This paper | | Stock contains VT019023-AD and VT050279-DBD |
| Genetic reagent (*D. melanogaster*) | pBMN-spGAL4 | This paper | | Stock contains R28D07-AD and VT050279-DBD |
| Genetic reagent (*D. melanogaster*) | TasteBMN-spGAL4 | This paper | | Stock contains VT023783-AD and R11D02-DBD |
| Genetic reagent (*D. melanogaster*) | *20XUAS-IVS-mCD8::GFP* | *Pfeiffer et al., 2010* | RRID:BDSC_32194 | |
| Genetic reagent (*D. melanogaster*) | *13XLexAop2-IVS-myr::GFP* | *Pfeiffer et al., 2010* | RRID:BDSC_32209 | |
| Genetic reagent (*D. melanogaster*) | C155-GAL4, UAS-nSyb.eGFP | Kendal Broadie | RRID:BDSC_6920 | |
| Genetic reagent (*D. melanogaster*) | BPADZp; BPZpGDBD | *Hampel et al., 2015* | RRID:BDSC_79603 | spGAL4 control |
| Genetic reagent (*D. melanogaster*) | BDPLexA | *Pfeiffer et al., 2010* | RRID:BDSC_77691 | |
| Genetic reagent (*D. melanogaster*) | *20XUAS-IVS-CsChrimson-mVenus* | *Klapoetke et al., 2014* | RRID:BDSC_55134 | |
| Genetic reagent (*D. melanogaster*) | MCFO-5 | *Nern et al., 2015* | RRID:BDSC_64089 | |
| Genetic reagent (*D. melanogaster*) | MCFO-3 | *Nern et al., 2015* | RRID:BDSC_64087 | |
| Genetic reagent (*D. melanogaster*) | *13XLexAop2-IVS-CsChrimson-mVenus* | *Klapoetke et al., 2014* | RRID:BDSC_55137 | |
| Antibody | Anti-GFP (Rabbit polyclonal) | Thermo Fisher Scientific | Cat# A-11122, RRID:AB_221569 | IF(1:500) |
| Antibody | Anti-Brp (Mouse monoclonal) | DSHB | Cat# nc82, RRID:AB_2314866 | IF(1:50) |
| Antibody | Anti-FLAG (Rat monoclonal) | Novus Biologicals | Cat# NBP1-06712, RRID:AB_1625981 | IF(1:300) |
| Antibody | Anti-HA (Rabbit monoclonal) | Cell Signaling Technology | Cat# 3724, RRID:AB_1549585 | IF(1:500) |
| Antibody | Anti-V5 (Mouse monoclonal) | Bio-Rad | Cat# MCA1360, RRID:AB_322378 | IF(1:300) |

*Continued on next page*

*Continued*

| Reagent type (species) or resource | Designation | Source or reference | Identifiers | Additional information |
|---|---|---|---|---|
| Antibody | Anti-Rabbit AF488 (Goat polyclonal) | Thermo Fisher Scientific | Cat# A-11034, RRID:AB_2576217 | IF(1:500) |
| Antibody | Anti-Mouse AF568 (Goat polyclonal) | Thermo Fisher Scientific | Cat# A-11031, RRID:AB_144696 | IF(1:500) |
| Antibody | Anti-Rat AF633 (Goat polyclonal) | Thermo Fisher Scientific | Cat# A-21094, RRID:AB_2535749 | IF(1:500) |
| Chemical compound, drug | Paraformaldehyde 20% | Electron Microscopy Sciences | Cat# 15713 | |
| Chemical compound, drug | DiD solid | Thermo Fisher Scientific | Cat# 07757 | |
| Chemical compound, drug | all-*trans*-Retinal | Toronto Research Chemicals | Cat# R240000 | |
| Software, algorithm | neuTube | *Feng et al., 2015* | | https://www.neutracing.com/ |
| Software, algorithm | Vcode | *Hagedorn et al., 2008* | | http://social.cs.uiuc.edu/projects/vcode.html |
| Software, algorithm | Fiji | *Schindelin et al., 2012* | | http://fiji.sc/ |
| Software, algorithm | R | R Core Team | RRID:SCR_001905 | https://www.r-project.org/ |
| Software, algorithm | CMTK | *Jefferis et al., 2007* | | https://www.nitrc.org/projects/cmtk/ |
| Software, algorithm | FluoRender | *Wan et al., 2012* | | http://www.sci.utah.edu/software/fluorender.html |
| Software, algorithm | Blender version 2.79 | Blender Online Community | RRID:SCR_008606 | https://www.blender.org/download/releases/2-79/ |
| Software, algorithm | MATLAB | MathWorks Inc, Natick, MA, USA | RRID:SCR_001622 | |
| Software, algorithm | natverse | *Bates et al., 2020* | | http://natverse.org/ |
| Software, algorithm | Cytoscape | *Shannon et al., 2003* | | https://cytoscape.org/ |

## Rearing conditions and fly stocks

GAL4, LexA, and *Split GAL4* (spGAL4) lines were generated by the labs of Gerald Rubin and Barry Dickson, and most lines are available from the Bloomington Drosophila Stock Center (*Dionne et al., 2017*; *Jenett et al., 2012*; *Pfeiffer et al., 2008*; *Tirian and Dickson, 2017*). Canton S flies were obtained from Martin Heisenberg's lab in Wurzburg, Germany. Other stocks used in this study are listed in the Key resources table.

GAL4, spGAL4, and LexA lines were crossed to either UAS or LexAop driver lines as described below. Flies were reared on Fisherbrand Jazz-Mix *Drosophila* food (Fisher Scientific, Fair Lawn, NJ, USA) containing corn meal, brown sugar, yeast, agar, benzoic acid, methyl paraben, and propionic acid. The flies were kept in an incubator at 21°C and 55–65% relative humidity. Flies that were not used for optogenetic experiments were kept on a 16/8 hr light/dark cycle. Flies used for optogenetic experiments were reared on food containing 0.4 mM all-*trans*-retinal (Toronto Research Chemicals, Toronto, Canada) in vials that were wrapped in aluminum foil and kept in a box to keep them in the dark. Unless stated otherwise, the flies used for experiments were 5- to 8-day-old males.

## Imaging the head bristles

One to 2 mm was cut off the tip of an Eppendorf 200 µL pipette tip, then an approximately 6 mm length was cut off and the remainder discarded. A freeze-killed (>1 hr) male or female Canton S fly was then gently pushed in with a piece of wire, until the head protruded from the tip. The tip was then mounted in a small piece of soft wax. An observation chamber was constructed on a microscope slide, by cutting a 5 mm square hole in three layers of Highland electrical insulation tape (3M, St.

Paul, MN, USA) and covering the bottom with a translucent white plastic square (cut from a Farmland Traditions dog treat bag). The fly, held in the tube, was mounted over the chamber using the wax, first dorsal side up (imaged), then ventral side up (imaged). The head was carefully cut off using sharpened iridectomy scissors, falling into the chamber where it was arranged anterior side up, held in place with a piece of coverslip, imaged, flipped posterior side up, and imaged again. Imaging was done with a Zeiss Axio Examiner D1 microscope equipped with a 10x Achroplan objective (0.25 NA) (Karl Zeiss, Oberkochen, Germany). The objective was surrounded with a cylinder of the same translucent white plastic in order to diffuse the light source and to avoid air movements that could move the antennae. The cylinder was illuminated from both sides at 2–3 cm distance by a Dolan-Jenner Fiber-lite (Dolan-Jenner Industries, Boxborough, MA, USA). A small amount of additional back-lighting was provided by the microscope light source (20% power) with a blue filter. Images were captured with a Zeiss Axiocam 512 at 60 ms exposure. The focal plane was advanced in small increments manually, resulting in approximately 50 images per head.

The image Sequence function of Fiji software (http://fiji.sc/) was used to combine all the images into a stack. The pixel size was adjusted to 0.3125 µm and calibrated with an image of a slide micrometer. The stack was downsized to 2048 pixels minimum dimension (usually the height), and an Unsharp Mask filter applied (3 pixel radius, 0.6 mask weight), then the sides of the stack were cropped to remove unnecessary space.

The Fiji *extended depth of field* (EDF) plugin (Alex Prudencio, EPFL, École polytechnique fédérale de Lausanne) was used to superimpose in-focus areas from the stacks. For acceptable processing times, the stack was downsized to 1024 pixels minimum dimension. The EDF process requires square images, so to avoid excessive cropping the canvas size was increased to 2048 pixels square, resulting in a black surround. The best results were obtained by averaging (with Image Calculator) (1) the result of EDF Easy Mode Fast setting, Gaussian-blurred by 1 pixel radius, with (2) the result of EDF Easy Mode High setting. The resulting image was cropped to 1024 minimum dimension, sharpened with Unsharp Mask, radius 1 pixel, and adjusted for optimal contrast.

With anterior views of the head, the EDF algorithm has difficulties separating the aristae from the underlying eye facets. Thus, for presentation images of the front view of the head, the EDF process was carried out separately on an anterior stack with the aristae present, and a more posterior stack with the antennae absent. These results were imported as layers into GIMP (GNU Image Manipulation Program) and manually combined by masking.

To facilitate bristle identification, a color-coded depth map was constructed from the downsized 1024-height stack, sharpened with Unsharp-mask (1 pixel). The color channels were split, and G and B channels discarded. The R channel was Inverted (Edit menu), the contrast was adjusted, and a Gamma correction of 1.34 was applied to the stack. A lookup table (LUT) had been previously created ranging from light blue, through white and yellow to dark red ('Stellar'). The Image: Hyperstacks: Temporal-Color Code function (LUT Stellar) was applied to the stack, giving a depth-coded image. Finally, an Enhanced Local Contrast (CLAHE, blocksize 63) was applied, followed by a gamma adjustment (1.5–1.8).

## Bristle nomenclature

Published names for the different bristles were used when possible. However, bristle abbreviations are from the present work unless otherwise indicated. Some bristle names were from *Bodenstein et al., 1994*, including the *frontal* (Fr), *frontoorbital* (FrOr), *orbital* (Or), *ocellar* (Oc), *interocellar* (InOc), *vertical* (Vt), *postorbital* (PoOr), and *vibrissae* (Vib). We deviated from this nomenclature in the following cases. First, although the PoOr bristles form a continuous row along the back margin of each eye (*Figure 1B*), we subdivided them into dPoOr and vPoOr populations based on whether their associated BMNs project through the OcciNv or EyeNv, respectively (*Figure 2F and J*). Second, we did not use the bristle name *postvertical*, but instead used *postocellar* (PoOc) that was previously proposed to better describe the location of these bristles as posterior to the other ocellar bristles (*Steyskal, 1976*). Third, instead of occipital, we used the name *supracervical* (Su) for the bristles located immediately above the cervical connective on the back of the head (*Steyskal, 1976*). This is because we named two populations of small bristles on the back of the head the dOcci and vOcci bristles. This name was previously proposed for these bristles in other species of flies (*Steyskal, 1976*). The abbreviation (Occi) was taken from the blowfly literature (*Theib, 1979*). Given that the Occi bristles are found as

two distinct populations in *D. melanogaster*, we refer to them in this work as the dOcci and vOcci bristles (*Figure 1B*).

Our abbreviation for the *vertical* (Vt) bristles describes all vertical bristles. The largest two have been referred to as *Vt inner* (Vti) and *exterior* (Vte) bristles in different fly species (*Steyskal, 1976*), and were named in the present work Vt 1 and Vt 2, respectively (*Figure 4—figure supplement 5A*). The other previously described Vt bristle (named Vt 3) is posterior to Vt 2. Medial to the Vt 2 bristle is a newly categorized fourth Vt bristle that we could not find a previous description of in *Drosophila* (labeled Vt 4 in *Figure 4—figure supplement 5A*). This bristle could be the paravertical bristle that was previously described (*Steyskal, 1976*).

The bristles on the eyes were referred to as InOm bristles (*Honegger et al., 1979*; *Zack and Bacon, 1981*). For the bristles on the outer labellum, we used the common name, *taste* (Taste) bristles (*Stocker, 1994*). We refer to the other bristles on the proboscis as MaPa and Hau bristles. Note: while most of the bristles on the head are termed trichoid sensilla that are innervated by a single BMN, the Taste bristles are mostly basiconic sensilla that are each innervated by a BMN and multiple gustatory neurons. The bristles on the first and second segments of the antennae are referred to as Ant bristles.

## Bristle quantification

We counted the bristle populations that are defined above and shown in *Figure 1A–D*. Most bristles were counted using color-coded depth maps (described above) that aided the identification of the bristles within each population (*Figure 1—figure supplement 2A–H*). The Ant, Fr, FrOr, Or, Oc, InOc, Vt, dOcci, PoOr, Vib, MaPa, Hau, PoOc, and Su bristles were counted using this method on eight male and four female heads (*Figure 1E*, *Figure 1—figure supplement 3A–E*, *Supplementary file 1* [Table 1]). Bristles on each half of the head were counted using the Fiji Cell Counter plugin (Kurt De Vos, University of Sheffield) and then averaged. These averages were used to calculate average and standard deviation for each bristle population from all counted male and female heads. Two-tailed t-tests were performed to compare male and female bristle numbers for each population. Although in this manuscript we make a distinction between the dPoOr and vPoOr bristles (*Figure 1B*), we counted all PoOr bristles together when comparing their numbers between males and females (*Figure 1—figure supplement 3E*). We used other approaches to obtain or estimate the numbers of dPoOr, vPoOr, InOm, vOcci, and Taste bristles (described below).

BMNs that innervate the PoOr bristles project through two different nerves, the Occi- and EyeNvs (*Figure 2F*). Specifically, BMNs innervating *dorsal PoOr* (dPoOr) bristles project through the OcciNv, while those innervating vPoOr bristles project through the EyeNv. We determined the average number of dPoOr or vPoOr bristles based on whether they were innervated by an Occi- or EyeNv-projecting BMN. The BMNs were labeled using R52A06-GAL4 to express membrane-targeted *green fluorescent protein* (mCD8::GFP) and imaged using a confocal microscope (imaging method described below). We then counted the dPoOr and vPoOr bristles from confocal images (example shown in *Figure 2—figure supplement 1D*). Multiple different heads were counted to determine the average number of dPoOr and vPoOr bristles (N=10, *Supplementary file 1* [Table 2]).

The InOm bristles were too small and numerous to be counted, and it was only necessary to estimate their numbers in this work. The eyes contain the majority of bristles on the head and the numbers of these bristles can vary. Data from a previous study indicated that each eye contains between 745 and 828 regularly spaced ommatidia (776 average), and most ommatidia have an associated bristle (*Ready et al., 1976*). However, some ommatidia around the eye edges are not associated with bristles. Based on one example of an eye from a Canton S fly (*Ready et al., 1976*), we calculated that 78% of the ommatidia had an associated bristle for that eye. This percentage was used to estimate that there are between 607 and 645 bristles on each eye from the above counted bristle ranges. This estimate was sufficient for identifying the EM-reconstructed BMNs that innervate the InOm bristles (described below) based on their overwhelming number relative to other head BMNs.

In this work we identified the vOcci bristles that were not previously described. These bristles were too small to be reliably observed at the level of resolution of the images shown in *Figure 1A–D*. To help visualize and count these bristles, we used a transgenic driver line pBMN-spGAL4 (R28D07-AD ∩ VT050279-DBD) that labels BMNs innervating these bristles (shown in *Figure 3F and F'*). pBMN-spGAL4 was used to express mCD8::GFP, and the ventral posterior head was imaged with a confocal microscope (see below for imaging method). The bristles could be counted by using the labeled BMN

dendrites to highlight their locations (shown in *Figure 3—figure supplement 1C*). pBMN-spGAL4 labeled almost all of the visible vOcci bristles, but in some heads, we could see bristles that did not have an labeled BMN (not shown). Therefore, it is possible that the counts of these bristles using the GFP-labeled BMNs are lower than the actual number. We determined the average number of Occi bristles by counting different heads (N=13, *Supplementary file 1* [Table 3]).

Most of the heads that we imaged had their proboscises oriented such that we could not observe and count all of the Taste bristles. However, Taste bristles have been counted in previous studies (*Falk et al., 1976*; *Jeong et al., 2016*; *Nayak and Singh, 1983*; *Shanbhag et al., 2001*). See *Supplementary file 1* (Table 4) for Taste bristle counts from different publications (published counts for Ant and MaPa bristles are also shown). One of these studies also determined that there were no differences in the numbers of Taste bristles between males and females (*Shanbhag et al., 2001*). We took the highest and lowest numbers from these different references for the range that is shown in *Figure 1E*.

## Head immunostaining and nerve reconstructions

R52A06-GAL4 (RRID:BDSC_38810), dBMN-spGAL4, pBMN-spGAL4, and TasteBMN-spGAL4 were crossed to *20XUAS-IVS-mCD8::GFP* (RRID:BDSC_32194) while VT017251-LexA (InOmBMN-LexA) was crossed to *13XLexAop2-IVS-myr::GFP* (RRID:BDSC_32209). Anesthetized male progeny were decapitated using a standard razor blade and heads were placed in phosphate-buffered saline (PBS). To facilitate antibody penetration for staining, we used #5 Dumoxel forceps (Fine Science Tools, Foster City, CA, USA) to tear small holes in the cuticle and pull off the antennae or proboscis. Heads were fixed in PBS with 2% paraformaldehyde for 1 hr at room temperature, and then washed with PAT (PBS, 1% bovine serum albumin, 0.5% Triton X) six times within 2 hr. Heads were blocked overnight at 4°C in PAT with 3% normal goat serum (PAT-NGS), then incubated for 3 days (room temperature during the day and 4°C at night) in PAT-NGS containing rabbit anti-GFP (Thermo Fisher Scientific, Waltham, MA, USA, RRID:AB_221569). Heads were washed with PAT for 5 hr and then incubated for 3 days in PAT-NGS with goat anti-rabbit Alexa Fluor-488 (Thermo Fisher Scientific, Waltham, MA, USA, RRID:AB_2576217). Heads were washed for 2 days with several exchanges of PAT, and then in PBS for 2 hr at room temperature. A standard slide was used for mounting with a small 'well' created by stacking five Avery reinforcement labels (Avery Products Corporation, Brea, CA, USA). A drop of Vectashield (Vector Laboratories, Inc, Burlingame, CA, USA) was added to the well and the heads were positioned either anteriorly or dorsally. The well was then covered with a circular coverslip (Electron Microscopy Sciences, Hatfield, PA, USA, 1.5 Micro Coverglass 12 mm diameter, Cat# 72230-01).

Heads were imaged using a Zeiss LSM800 confocal microscope (Carl Zeiss, Oberkochen, Germany) equipped with a 20× objective (Plan-Apochromat 20×/0.8). Fiji software (http://fiji.sc/) was used for examining the morphology of the imaged BMNs and for image processing steps, including adjustment of brightness and contrast, stitching, and image inversion. Reconstructions of the labeled head BMNs and their respective nerves from the confocal Z-stacks (*Figure 2* and *Figure 3*) were performed using the software neuTube (*Feng et al., 2015*). Image stacks of the heads are displayed as maximum intensity projections (*Figure 2* and *Figure 3*).

## CNS immunostaining and analysis

The different GAL4, spGAL4, and LexA driver lines described above were crossed to either *20XUAS-IVS-mCD8::GFP* or *13XLexAop2-IVS-myr::GFP*. Brains and VNCs were dissected and stained as previously described (*Hampel et al., 2015*; *Hampel et al., 2011*). The following primary and secondary antibodies were used for staining GFP and the neuropil: rabbit anti-GFP, mouse anti-nc82 (Developmental Studies Hybridoma Bank, University of Iowa, RRID:AB_2314866) to stain Bruchpilot, goat anti-rabbit Alexa Fluor-488, and goat anti-mouse Alexa Fluor-568 (Thermo Fisher Scientific, RRID:AB_144696). The stained CNSs were imaged, and confocal stacks processed as described above for heads.

To display the GFP expression patterns of different spGAL4 and LexA lines together as shown in *Figure 3J*, we used the Computational Morphometry Toolkit (CMTK) (https://www.nitrc.org/projects/cmtk/) (*Jefferis et al., 2007*) to computationally register individual confocal stacks of each line to the JFRC-2010 standard brain (https://www.virtualflybrain.org). The PIC file of each registered stack was loaded into Fiji and merged to display each in a different color channel.

For MCFO experiments, dBMN-spGAL4, pBMN-spGAL4, and TasteBMN-spGAL4 were crossed to the MCFO-5 stock (RRID:BDSC_64089) (*Nern et al., 2015*). 9- to 12-day-old fly brains were dissected

for pBMN-spGAL4 and dBMN-spGAL4, while 4- to 6-day-old brains were dissected for Taste-spGAL4. Brains were stained using rat anti-FLAG (Novus Biologicals, LLC, Littleton, CO, USA, RRID:AB_1625981), rabbit anti-HA (Cell Signaling Technology, Danvers, MA, USA, RRID:AB_1549585), mouse anti-V5 (Bio-Rad, Hercules, CA, USA, RRID:AB_322378), goat anti-rabbit Alexa Fluor-488, goat anti-mouse Alexa Fluor-568, goat anti-rat Alexa Fluor-633 (Thermo Fisher Scientific, RRID:AB_2535749). Stained brains were imaged, and confocal stacks processed as described above. Individually labeled neurons from each line are shown as maximum projections in *Figure 4* and *Figure 4—figure supplements 6–8*. Given that very few flipout events occurred using the MCFO-5 stock in combination with the dBMN-spGAL4 and pBMN-spGAL4, we used a different MCFO stock (MCFO-3, RRID:BDSC_64087) to obtain a higher number of individually labeled neurons. Using dBMN-spGAL4 crossed with MCFO-3, we aged males for 9–12 days and obtained approximately one to two individually labeled neurons in seven dissected brains. Crossing pBMN-spGAL4 with MCFO-3, we obtained about one to four labeled neurons in each dissected brain when we aged males 4–6 days. The flipout events in the latter cross occurred in much higher frequency given that most brains labeled more than a single neuron with the MCFO-3 stock.

## Identification of driver lines that express in different subsets of head BMNs

We used the spGAL4 system to produce driver lines that expressed in different subsets of head BMNs (*Luan et al., 2006*; *Pfeiffer et al., 2010*). spGAL4 allows for independent expression of the GAL4 DNA binding domain (DBD) and activation domain (AD). When DBD and AD are expressed in the overlapping neurons, these domains reconstitute into a transcriptionally active protein. To label specific subpopulations of head BMNs, we visually screened through an image collection of the CNS expression patterns of enhancer-driven lines and identified candidate lines that were predicted to express in different subsets of head BMNs (*Dionne et al., 2017*; *Jenett et al., 2012*; *Tirian and Dickson, 2017*). We selected two candidate lines to express the DBD in subsets of head BMNs: VT050279 (VT050279-DBD, RRID:BDSC_72433) and R11D02 (R11D02-DBD, RRID:BDSC_68554). VT050279-DBD or R11D02-DBD flies carrying the *20XUAS-IVS-CsChrimson-mVenus* transgene (RRID:BDSC_55134) (*Klapoetke et al., 2014*) were crossed to 55 different candidate-ADs. The progeny were placed in behavioral chambers and exposed to red light for optogenetic activation (described below). We tested three flies for each DBD/AD combination to identify those that expressed in neurons whose activation could elicit grooming. Grooming 'hits' were stained using a GFP antibody to detect CsChrimson-mVenus expression in the CNS and anti-NC82 to mark the neuropil as described above. We identified three different combinations that expressed in restricted subsets of BMNs, that included the ADs R28D07-AD (RRID:BDSC_70168), VT019023-AD (RRID:BDSC_71430), and VT023783-AD (RRID:BDSC_73261). We generated stable lines containing both the AD and DBD, including dBMN-spGAL4 (VT019023-AD ∩ VT050279-DBD), pBMN-spGAL4 (R28D07-AD ∩ VT050279-DBD), and TasteBMN-spGAL4 (VT023783-AD ∩ R11D02-DBD).

## Behavioral analysis procedures

For behavioral experiments, dBMN-spGAL4, pBMN-spGAL4, TasteBMN-spGAL4, and BPADZp; BPZpGDBD (spGAL4 control, RRID:BDSC_79603) were crossed to *20XUAS-CsChrimson-mVenus*. InOmBMN-LexA and BPADZp and BDPLexA (LexA control, RRID:BDSC_77691) were crossed to *13XLexAop2-IVS-CsChrimson-mVenus* (RRID:BDSC_55137). The controls used with the spGAL4 and LexA lines contain the vector backbone that was used to produce each line (including the coding regions for each spGAL4 half or LexA), but lack any enhancer to drive spGAL4 or LexA expression (*Hampel et al., 2015*; *Pfeiffer et al., 2010*; *Pfeiffer et al., 2008*).

We used a previously reported behavioral optogenetic rig, camera setup, and methods for the recording of freely moving flies (*Hampel et al., 2017*; *Hampel et al., 2015*; *Seeds et al., 2014*). The stimulus parameters used were 656 nm red light at 27 mW/cm$^2$ intensity delivered at 5 Hz for 5 s (0.1 s on/off) with 10 or 30 s interstimulus intervals (total of three stimulations). While most of the driver lines were recorded using 30 s interstimulus intervals, pBMN-spGAL4 was recorded using 10 s intervals. This was because the elicited head nodding behavior occurred more robustly when stimulated every 10 s rather than 30 s.

Manual scoring of behavior from prerecorded video was performed using VCode software (*Hagedorn et al., 2008*) and analyzed in MATLAB (MathWorks Incorporated, Natick, MA, USA). Some grooming and avoidance-like movements were annotated as previously described, including antennal, eye, ventral head, proboscis, and backward motion (*Hampel et al., 2020a*; *Hampel et al., 2017*; *Hampel et al., 2015*; *Seeds et al., 2014*). In this work, ventral head and proboscis grooming were combined and referred to as ventral head grooming. Head nodding was annotated when the fly tilted its head downward by any amount until it returned its head back in its original position. This movement often occurred in repeated cycles. Therefore, the 'start' was scored at the onset of the first forward movement and the 'stop' when the head returned to its original position on the last nod. Dorsal head grooming was scored when the fly used one or both legs to touch its dorsal head, which included the dorsal part of the eye. During dorsal head grooming, flies sometimes nodded, rotated, or kept their heads in their original position to groom the outer dorsal areas of the head (dorsal eye and posterior dorsal eye). The start of dorsal head grooming was scored when the legs reached their farthest posterior position on the head, before sweeping in the opposite direction. The movement was scored as stopped three frames after the legs last touched the head.

Behavioral data was analyzed using nonparametric statistical tests as previously reported (*Hampel et al., 2020a*; *Hampel et al., 2017*; *Hampel et al., 2015*). The percent time flies spent performing each behavior was calculated. To compare the behavior performed by each experimental genotype with its corresponding genetic control, we performed pairwise comparisons for each behavior using a Mann-Whitney U test and applied Bonferroni correction. Note that we tested both male and female flies for optogenetic activation of behavior. Although only males are presented in this manuscript, optogenetic activation was found to elicit similar behaviors in both males and females (not shown).

## Dye filling of BMNs that innervate large bristles

The dye filling protocol used in this study was adapted from one that was previously published (*Kays et al., 2014*). C155-GAL4, *UAS-nSyb.eGFP* flies (RRID:BDSC_6920) were used for the dye fill experiments to label the neuropil. Flies were decapitated with a standard razor blade and their heads glued to a microscope cover glass (Fisher Scientific, Pittsburgh, PA, USA) using TOA 400 UV cured glue (Kemxert, York, PA, USA). Heads were submerged in 3.7% wt/vol paraformaldehyde in 0.2 M carbonate-bicarbonate buffer at pH 9.5 overnight at 4°C. Heads were washed 24 hr later by dipping in 0.2 M carbonate-bicarbonate buffer at pH 9.5 for 30 s, and subsequently in ddH$_2$O for 30 s. Heads were gently blotted dry to prevent dye from spilling over the cuticle, and the selected bristles on the head were plucked with #5 Dumoxel forceps. Bristles were selected from either the left or right side of the head, depending on which bristle was in the most optimal orientation for plucking and filling.

Micropipettes for dye filling were prepared from Borosilicate Thin Wall capillaries (Warner Instruments, Holliston, MA, USA, G100T-4). Capillaries were filled with 5–10 µL of dye solution of 10 µg/µL DiD (Thermo Fisher Scientific) in 100% ethanol and the tip was approached to the bristle socket with a micromanipulator. The tip of the capillary was made to contact the edge of the bristle socket such that the dye diffused into the socket until a stable bubble of solution formed. Heads were then dried for 5 min and then submerged in a 0.2 M carbonate-bicarbonate buffer at pH 9.5, in the dark, and at room temperature for 48 hr. The brains were then dissected and imaged immediately without fixation. Dissected brains were placed on a microscope slide with two Avery circular reinforcement labels and a circular coverslip. The brains were imaged immediately using a Zeiss LSM 800 confocal microscope. The native fluorescence of nSyb.eGFP was preserved enough at the conclusion of the experiment to image the brain neuropil (*Figure 4—figure supplement 1B–E*).

We attempted to fill BMNs innervating the following head bristles: PoOc, Oc, Or, Ant, Vib, and Vt. Many attempts to fill particular bristles resulted in unfilled or partially filled BMNs. Anecdotally, there also seemed to be a difference in how well the filling method worked for the different bristle populations. For example, multiple attempts to fill the Vt bristles only resulted in one successful fill of a BMN from Vt 1 and Vt 3 bristles. Additionally, we were unable to fill a BMN with multiple attempts of the PoOc bristle. Successful fill trials and the locations of specific bristles that were filled are shown in *Figure 4C–Q* and *Figure 4—figure supplements 2–5*.

## Light microscopy image stack storage and availability

The Z-stacks used to produce panels for *Figure 1*, *Figure 2*, *Figure 3*, and *Figure 4* and their figure supplements are all available for download at the Brain Image Library (RRID:SCR_017272). Links for each image Z-stack can be found in *Supplementary file 2*. The group DOI for all these Z-stacks is https://doi.org/10.35077/g.1144.

## Reconstruction and analysis of BMNs from an EM volume

BMNs were reconstructed in a complete EM volume of the adult female brain (FAFB) dataset (*Zheng et al., 2018*) using the FlyWire.ai platform (*Dorkenwald et al., 2023*; *Dorkenwald et al., 2022*). We first identified the locations of the Ant-, Occi-, and merged Eye/LabNvs in the EM volume based on their identified locations from light microscopy data. We then chose a cross section of each nerve, close to where they enter brain neuropil, and where segmentation was available for all neurons in the nerve (*Figure 5—figure supplement 1A–D*). We seeded every profile in the Occi- and merged Eye/LabNvs (*Figure 5—figure supplement 1C and D*). The Eye/LabNvs had a bundle of soma tracts from an SEZ interneuron hemilineage crossing the seed plane that was excluded from the seeding process based on the morphology of their initial segmentation (*Figure 5—figure supplement 1C*). Previous studies reconstructed major portions of JONs in FAFB AntNv using the CATMAID platform (*Hampel et al., 2020a*; *Kim et al., 2020*). Because the FlyWire FAFB brain was locally realigned, we transformed those JONs into the FlyWire space using natverse version 0.2.4 (*Bates et al., 2020*). After overlaying these JONs onto the seed plane, they were excluded during the seeding effort to identify BMNs in the AntNv. This left a small ventral-medial area of the nerve with previously undocumented neurons that were seeded and reconstructed (*Figure 5—figure supplement 1B*). We focused our reconstructions in the right hemisphere nerves. However, we also examined the OcciNv in the left hemisphere, given that it only contained a small number of neurons (*Figure 6—figure supplement 4A–D*).

The segmentations of all seeded neurons were then fully proofread by a human annotator. This process involves splitting falsely merged parts and merging falsely missing parts of a neuron using the tools available in the FlyWire neuroglancer instance (flywire.ai). FlyWire neuroglancer was also used to examine the morphologies of neurons for classification purposes.

The neurons classified as sensory origin (no soma in the brain, only axonal projections and entering through a nerve) were skeletonized using natverse (skeletor in fafbseg package version 0.10.0). Skeletons were pruned to synapse-rich areas to exclude the smooth axon in the nerve bundle. We created a three-dimensional mesh of the synapses (for synapses see below) and pruned the skeletons to arbors within the mesh volume. We then compared their morphology using the NBLAST algorithm (*Costa et al., 2016*) and clustered the similarity scores with Ward (*Figure 6—figure supplement 1A and B*). The skeletons of CATMAID reconstructions of JONs from *Hampel et al., 2020a*; *Kim et al., 2020*, and skeletonizations of sensory neurons from this publication were transformed into JRC2018F standard brain space for plotting using natverse. The transformed skeletons, meshes from the FlyWire proofread segmentation, and brain neuropil meshes were plotted with natverse in RStudio 2022.02.3. We used the standard brain transformations to analyze if BMNs crossed the midline in the brain. Defining the midline accurately is possible due to the symmetric nature of this brain space. If any skeleton node coordinates of a given BMN were located in the contralateral hemisphere (x of node >x of midline) we classed the BMN as midline-crossing.

FlyWire provides access to a synapse table imported from *Buhmann et al., 2021*. We queried the table for pre- and postsynaptic sites belonging to the reconstructed sensory neurons for connectome analysis (*Figure 8—figure supplements 1–3*). We used the cleft scores of the cleft prediction from *Heinrich et al., 2018*, to filter synapses with scores below 50, which reliably excludes falsely predicted synapses.

We then analyzed the connectivity of BMNs with each other, and with other postsynaptic partners. First, we analyzed BMN-type connectivity by adding up synaptic weights of BMN-to-BMN edges by type and normalizing by the number of possible edges between the groups (*Figure 8—figure supplement 1B*). Graphs were plotted using Cytoscape version 3.9.1 and the RCy3 (version 2.17.1) and igraph (version 1.3.0) packages for Cytoscape control from R (*Shannon et al., 2003*). Further, we compared BMN types regarding their pre- and postsynaptic connection counts (*Figure 8—figure supplement 1A*). We counted all entries in the FlyWire synapse table at which a given BMN was

either the pre- and postsynaptic partner (not number of presynaptic sites, cleft scores ≥50). This revealed that BM-InOm neurons had fewer synaptic connections than other BMN types, which was not surprising given their small, non-complex axonic arbors in the brain. This posed a challenge when comparing BM-InOm neuron synaptic connectivity to other BMN types, as described below.

We calculated cosine similarity of BMNs to cluster them based on their connectivity similarity. Cosine similarity emphasizes the similarity of BMNs which have similar sets of postsynaptic targets. We took advantage of the good proofreading state of the FlyWire datasat to perform postsynaptic connectivity analysis (*Dorkenwald et al., 2023*; *Dorkenwald et al., 2022*). Note that this strategy is agnostic about partner types. A comprehensive typing of the postsynaptic partners will be included in a follow-up study as it exceeds the focus of this study. For this analysis, we excluded synapses between BMNs and only considered postsynaptic connectivity to neuron partners of other types (non-BMNs). As mentioned, BM-InOm neurons have few connections to postsynaptic partners (pre-counts in *Figure 8—figure supplement 1A*), so we choose a low threshold of three synapses and excluded edges below that. When clustering the cosine similarity (ComplexHeatmap version 2.11.2 package in R) for all BMNs, we found that subsets of the BM-InOm neurons clustered together with other BMN types (data not shown), but the clustering failed to capture meaningful groups. Choosing a higher threshold resulted in most of the BM-InOm neuron connectivity being excluded, giving low cosine similarity scores even when compared to each other. We thus choose a higher threshold of seven synapses to analyze the cosine similarity of the BMN types excluding BM-InOm neurons (*Figure 8—figure supplement 2*) and the low synapse threshold of three for analysis of the BM-InOm neurons separately (*Figure 8—figure supplement 3*). This analysis better captured the somatotopic mapping of the BMN types in the brain. From our preliminary analysis we expect BM-InOm neurons to share postsynaptic partners with other BMN types, but further analysis is required to investigate patterns of connectivity of all BMNs to postsynaptic targets.

Connections to postsynaptic partners were calculated as follows: We analyzed all postsynaptic partners of the BMNs (after applying the above-mentioned cleft score) and summed the BMN input by BMN type. We assigned postsynaptic partners to the BMN type giving the highest input to them (e.g. if BM-Ant neurons contribute the highest synaptic input to a given postsynaptic neuron this neuron is termed Ant_post). We then plotted the connections from BMN types to the postsynaptic partner neurons grouped by their major input partner. To show only strong connections we applied a threshold (>5%) by normalizing the edges by the total output of a given BMN type (*Figure 8F*).

## Matching EM-reconstructed BMN projections with light microscopy imaged BMNs that innervate specific bristles

Manual categorization of the EM-reconstructed sensory neurons revealed that they consist of BMNs, JONs, TPMNs, and *gustatory neurons* (GRNs). JONs and GRNs were previously identified and described in the EM volume using the CATMAID platform (*Engert et al., 2022*; *Hampel et al., 2020a*; *Kim et al., 2020*). The TPMNs were identified based on their morphological similarity to previous light microscopy descriptions (*Jeong et al., 2016*; *Miyazaki and Ito, 2010*; *Zhou et al., 2019*). 25 sensory neurons were reconstructed that could not be identified based on dye fill, MCFO, or published neurons. These neurons are referred to in this work as unknown and shown in *Figure 5—figure supplement 2A–Y*. We also identified interneurons that were not classified because sensory neurons were the focus of this study. The BMNs, JONs, and TPMNs are shown in *Figure 5—figure supplement 1A and E*. The GRNs and interneurons were excluded from further analysis in this study.

The 705 BMNs and 25 unknown sensory neurons were clustered based on NBLAST similarity scores as described above (*Figure 6—figure supplement 1A and B*). We used these clusters in conjunction with manual anatomical inspection of light microscopy images to assign the BMNs to specific bristle populations. In particular, we compared the morphologies of the EM-reconstructed BMNs with dye-filled, stochastically labeled (MCFO), and driver line-labeled BMNs (*Figure 6—figure supplement 2A–N*). In some cases we did not have light microscopy images to enable direct matching of particular BMNs to their corresponding bristles. These BMNs were matched based on the morphology of BMNs innervating neighboring bristle populations (described for specific BMNs below). We also verified that the numbers of BMNs for each type were consistent with the numbers of their bristles (*Figure 6—figure supplement 3B–F*). In the specific cases described below, we used additional evidence to match the different BMNs to their bristles, including comparing the reconstructed BMNs on both

sides of the brain (OcciNv only, *Figure 6—figure supplement 4A–D*) and based on common connectivity of the BMNs with their postsynaptic partners (*Figure 8A–C* and *Figure 8—figure supplement 1*). A full list of the BMNs can be found in *Supplementary file 3*.

The BMNs have been referred to in some previous work as external sensilla (es neurons). We adopted BMN here because it was similar to the widely adopted nomenclature for different sensory modalities, such as GRNs, ORNs, JONs. Further, it more explicitly links the neurons to the bristles.

Three BMNs that innervate the InOc bristles (BM-InOc neurons, *Figure 6F*) were identified based on comparison with MCFO data (*Figure 6—figure supplement 2C*). One BMN that innervates the Oc bristle (BM-Oc neuron, *Figure 6G*) was identified based on comparison with dye fill data (*Figure 6—figure supplement 2D*). Six BMNs are proposed to innervate the Fr bristles (BM-Fr neurons, *Figure 6H*). The Fr bristles are located near the midline, immediately above the Ant bristles and below the InOc and Oc bristles. BMNs that innervate the Ant, InOc, and Oc bristles show very similar morphology with the BM-Fr neurons, including the ipsilateral and midline projections. 20 BMNs that innervate the Ant bristles (BM-Ant neurons, *Figure 6I*) were identified based on comparison with dye fill data (*Figure 6—figure supplement 2E*). Three BMNs that innervate the Or bristles (BM-Or neurons, *Figure 6J*) were identified based on comparison with dye fill data (*Figure 6—figure supplement 2F*). Six BMNs are proposed to innervate the FrOr bristles (BM-FrOr neurons, *Figure 6K*). The FrOr bristles are located immediately ventral to the Or bristles. The BM-Or neurons show very similar morphology with the proposed BM-FrOr neurons, including an ipsilateral projection.

555 BMNs that innervate the InOm bristles (BM-InOm neurons, *Figure 6L*) were identified using previous descriptions of the projections of these neurons (*Hampel et al., 2017*; *Zhang et al., 2020*). The similarity of the collective projections of the BM-InOm neurons with those labeled by the InOmBMN-LexA driver line are shown in *Figure 6—figure supplement 2N*. Clustering based on NBLAST similarity scores revealed that there were morphologically distinct groups of BM-InOm neurons. However, in contrast to other BMN types, the BN-InOm neurons were small with few relatively simple branches, and the clusters were likely due to relatively minor differences in these branches. For example, at the selected cut height (H=5) resulted in 11 different clusters (*Figure 6—figure supplement 1A and B*). Additionally, while the BM-InOm neurons showed some differential postsynaptic connectivity based on cosine clustering results, these differences were relatively low (*Figure 8—figure supplement 3*). Therefore, the BM-InOm neurons are treated as a single group that innervates eye bristles in this work, while a future study will further examine the heterogeneity of the BM-InOm neurons.

18 EyeNv-projecting BMNs that innervate the Vib bristles (BM-Vib neurons, *Figure 6M*) were identified based on comparison with dye fill data (*Figure 6—figure supplement 2G*). This includes three BMNs that project through the AntNv (BM-Vib [AntNv] neurons) that are morphologically similar to the BM-Vib (EyeNv) neurons. Based on 52A06-GAL4 labeling of BMNs on the head, BM-Vib (AntNv) neurons innervate smaller anterior Vib bristles that are lateral to Vib 1 and 2 bristles. In total, 15 BMNs projecting through the EyeNv (BM-Vib [EyeNv]) and 3 projecting through the AntNv (BM-Vib [AntNv]) were identified in the EM dataset.

15 BMNs are proposed to innervate the MaPa bristles (BM-MaPa neurons, *Figure 6N*), as their morphology matches a previous description (*Naresh Singh and Nayak, 1985*). It was difficult to distinguish between BM-MaPa and BM-Vib neurons based on morphology or nerve projection. However, all BM-MaPa neurons showed high cosine similarity in their postsynaptic connectivity to non-BMN neurons, and clustered together (*Figure 8—figure supplement 2*).

BMNs that innervate the Taste bristles (BM-Taste neurons, *Figure 6O*) were identified based on their similarity to the MCFO images (*Figure 6—figure supplement 2H*). We did not have dye fill or MCFO images of the BMNs that innervate the Hau bristles (BM-Hau neurons). These BMNs were presumed to show similar morphology to the BM-Taste neurons, given their close proximity. Indeed, visual inspection revealed five BMNs that had similar, yet slightly different morphology (*Figure 6P*). The axons appeared larger in diameter and showed a dorsal midline-crossing branch that was not found in BM-Taste neurons. Further, the five BM-Hau neurons showed high cosine similarity in their postsynaptic connectivity with non-BMN neurons (*Figure 8—figure supplement 2*).

Five BMNs that project through the OcciNv that could innervate the Vt bristles (BM-Vt neurons, *Figure 6Q*) were identified based on comparison with dye fill data (*Figure 6—figure supplement 2I–K*). Because there are only four Vt bristles, the additional BMN in this group is proposed to innervate the PoOc bristle (BM-PoOc neuron). However, because we had no dye fills for the BM-PoOc

neuron and limited dye fill examples of BM-Vt neurons, we included all five BMNs in the category BM-Vt/PoOc neurons. PoOc is hypothesized to project through the OcciNv and show similar morphology to the neighboring BM-Oc and -InOc neurons, and at least one BM-Vt/PoOc neuron has the expected morphology. Although the BM-InOc neuron is presumed to exist, we could not find evidence of a neuron innervating the PoOc bristle using any of the BMN driver lines reported in this study (*Figure 2—figure supplement 1G*).

Seven BMNs projecting through the OcciNv showed very similar morphology and were identified as innervating the dPoOr and dOcci bristles and named BM-dPoOr and BM-dOcci neurons, respectively. The morphology of the BM-dPoOr neurons was determined from MCFO experiments using dBMN-spGAL4 (*Figure 6—figure supplement 2L*). MCFO experiments using pBMN-spGAL4 did not enable us to distinguish between BM-dPoOr and BM-dOcci neurons, as we could not tell which BMN came from which bristle population (*Figure 4S, T*, *Figure 4—figure supplement 7B–D*). One difficulty in matching the reconstructed BM-PoOr and BM-dOcci neurons was that their numbers in the left brain hemisphere did not match what we expected from the numbers of their corresponding bristles. For example, there were only four tentatively assigned BM-dPoOr neurons, while we expected between five and nine (*Figure 6—figure supplement 3E*). Because the OcciNv contains only a small number of BMNs, we also reconstructed BMNs in the right brain hemisphere OcciNv (*Figure 6—figure supplement 4A–D*). This revealed four additional BMNs on the right than on the left for the BM-dPoOr neurons, which matched what we expected (*Figure 6—figure supplement 4D*). The numbers of the other BMN types that project through the OcciNv (i.e. BM-Vt/PoOc and BM-dOcci neurons) were the same in both hemispheres and matched with the expected number of bristles.

26 BMNs projecting through the EyeNv show very similar morphology and were identified as innervating the vPoOr and vOcci bristles. The vOcci-innervating BMNs (BM-vOcci neurons) were matched with MCFO images (*Figure 6—figure supplement 2M*). However, we did not obtain dye fill or MCFO images of BMNs innervating the vPoOr bristles (BM-vPoOr neurons). These neurons were expected to show similar morphology with the BM-vOcci neurons based on their close proximity. Because we could not distinguish between these BMNs, they were grouped together and named (BM-vOcci/vPoOr neurons, *Figure 6T*).

## Acknowledgements

We thank Brian Chen's lab at McGill University Health Centre for essential feedback on the development of the dye labeling protocol from this study. We thank the Princeton FlyWire team and members of the Murthy and Seung labs for development and maintenance of FlyWire (supported by BRAIN Initiative grant MH117815 to Murthy and Seung), and the Princeton EM proofreading team for contributing neuronal edits. We thank FlyWire users who contributed >10% proofreading of individual neurons or identification of neurons: Mala Murthy group ([>10% of 31 neurons] Claire McKellar, Austin T Burke, Sarah Morejohn, Kyle Patrick Willie, Merlin Moore, Doug Bland, Ben Silverman, Jay Gager, Ryan Willie), Carlos Ribeiro group ([>10% of 35 neurons] Ibrahim Tastekin, Raquel Barajas), the *Drosophila* Connectomics Group, Cambridge University (leads: M Costa and G Jefferis) for contributing to the tracing of FlyWire neurons ([>10% of 20 neurons] Yijie Yin, Paul Brooks, Laia Serratosa Capdevila, Siqi Fang, A Javier, Paul Brooks), Rachel Wilson group ([>10% of 8 neurons] Peter Gibb), Julie Simpson group ([>10% of 2 neurons] Li Guo), Wolf Hütteroth group (>10% of 4 neurons), and Jinseop Kim group ([>10% of 753 neurons] Hyunsoo Yim, Senongbong Yu, Chan Hyuk Kang, Taehyun Choe, Yeonju Nam, Tae Kim, Jinmook Kim).

Research reported in this publication was supported by the National Institute of Neurological Disorders and Stroke of the National Institutes of Health under Award Number RF1NS121911. The content is solely the responsibility of the authors and does not necessarily represent the official views of the National Institutes of Health. This work was also funded by the Whitehall Foundation (2017-12-69), Puerto Rico Science, Technology & Research Trust (2020-00195), NIMHD MD007600 (RCMI), NIGMS-NIH RISE (R25GM061838), NIGMS-NIH RISE (5R25GM061151), NIH NIGMS GM103642, and NSF (HRD-1736019).

## Additional information

### Funding

| Funder | Grant reference number | Author |
|---|---|---|
| BRAIN Initiative | 1RF1NS121911-01 | Andrew M Seeds<br>Stefanie Hampel |
| Puerto Rico Science, Technology and Research Trust | 2020-00195 | Andrew M Seeds |
| Whitehall Foundation | 2017-12-69 | Andrew M Seeds |
| National Institute on Minority Health and Health Disparities | MD007600 | Andrew M Seeds |
| National Institute of General Medical Sciences | R25GM061838 | Alexis Santana-Cruz |
| National Institute of General Medical Sciences | 5R25GM061151 | Adrián Alejandro-García |
| National Institute of General Medical Sciences | GM103642 | Andrew M Seeds<br>Stefanie Hampel |
| National Science Foundation | HRD-1736019 | Andrew M Seeds<br>Alexis Santana-Cruz |

The funders had no role in study design, data collection and interpretation, or the decision to submit the work for publication.

### Author contributions

Katharina Eichler, Data curation, Formal analysis, Investigation, Methodology, Writing - original draft, Writing - review and editing; Stefanie Hampel, Conceptualization, Formal analysis, Supervision, Validation, Investigation, Writing - original draft, Writing - review and editing; Adrián Alejandro-García, Formal analysis, Investigation, Methodology; Steven A Calle-Schuler, Alexis Santana-Cruz, Lucia Kmecova, Investigation; Jonathan M Blagburn, Investigation, Methodology, Writing - review and editing; Eric D Hoopfer, Software, Formal analysis; Andrew M Seeds, Conceptualization, Data curation, Formal analysis, Supervision, Funding acquisition, Investigation, Writing - original draft, Writing - review and editing

### Author ORCIDs

Katharina Eichler http://orcid.org/0000-0002-7833-8621
Stefanie Hampel http://orcid.org/0000-0001-8287-549X
Steven A Calle-Schuler http://orcid.org/0000-0003-1304-585X
Alexis Santana-Cruz http://orcid.org/0000-0002-8162-8072
Jonathan M Blagburn http://orcid.org/0000-0002-7681-2764
Andrew M Seeds http://orcid.org/0000-0002-4932-6496

Reviewer #2 (Public Review): https://doi.org/10.7554/eLife.87602.3.sa1
Reviewer #3 (Public Review): https://doi.org/10.7554/eLife.87602.3.sa2
Author response https://doi.org/10.7554/eLife.87602.3.sa3

## Additional files

### Supplementary files

• Supplementary file 1. Quantification of bristle populations on male and female heads. Table 1. Bristles counted using white light-illuminated male and female heads. Results of a two-tailed t-test comparing male and female bristle numbers are also shown. Table 2 shows counts of PoOr bristles innervated by BMNs that project through the OcciNv (dPoOr bristles) or the EyeNv (vPoOr bristles). BMNs were labeled by R52A06-GAL4. Table 3 shows vOcci bristles counted based on having an

associated BMN labeled in the pBMN-spGAL4 (R28D07-AD ∩ VT050279-DBD) pattern (males). Table 4 shows a summary of published bristle numbers.

• Supplementary file 2. Table of download links for light microscopy image stacks used in this manuscript. Image stacks are stored in the Brain Image Library (BIL), and were used to generate panels for *Figure 1*, *Figure 2*, *Figure 3*, *Figure 4*, and figure supplements for these figures. Rows in the spreadsheet correspond to each image stack. Columns provide information about each stack including: figure panels that each image stack contributed to, image stack title, DOI for each stack (link provides metadata for each stack and file download link), image stack file name, genotype of imaged fly, and information about image stack.

• Supplementary file 3. Table of electron microscopy (EM)-reconstructed bristle mechanosensory neurons (BMNs), Johnston's organ neurons (JONs), and taste peg mechanosensory neurons (TPMNs). Table includes FlyWire.ai neuron XYZ coordinates and IDs, nerve groups, names, types, NBLAST clusters, and cosine similarity clusters.

• MDAR checklist

### Data availability

The Z-stacks used to produce panels for *Figures 1–4* and their figure supplements are all available for download at the Brain Image Library (RRID:SCR_017272). The group DOI for all these Z-stacks is https://doi.org/10.35077/g.1144. Neuron reconstructions are available at FlyWire.ai.

The following dataset was generated:

| Author(s) | Year | Dataset title | Dataset URL | Database and Identifier |
|---|---|---|---|---|
| Seeds A, Hampel S, Blagburn J, Alejandro A | 2023 | Image stacks of bristles and bristle mechanosensory neurons on the head of *Drosophila melanogaster* described in the manuscript entitled: Somatotopic organization among parallel sensory pathways that promote a grooming sequence in *Drosophila* | https://doi.org/10.35077/g.1144 | Brain Image Library, 10.35077/g.1144 |

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
