## [Editor Report · eLife assessment]

This **valuable** work provides a near-complete description of the mechanosensory bristles on the *Drosophila melanogaster* head and the anatomy and projection patterns of the bristle mechanosensory neurons that innervate them. The data presented are **solid**. The study has generated numerous resources for the community that will be of interest to neuroscientists in the field of circuits and behaviour, particularly those interested in mechanosensation and behavioural sequence generation.

---

## [Referee Report · Reviewer #2 (Public Review)]

The authors combine genetic tools, dye fills and connectome analysis techniques to generate a "first-of-its-kind", near complete, synaptic resolution map of the head bristle neurons of *Drosophila*. While some of the BMN anatomy was already known based on previous work by the authors and other researchers, this is the first time a near complete map has been created for the head BMNs at electron microscopy resolution.

Strengths:

(1) The authors cleverly use techniques that allow moving back and forth between periphery (head bristle location) and brain, as well as moving between light microscopy and electron microscopy data. This allows them to first characterize the pathways taken by different head BMNs to project to the brain and also characterize anatomical differences among individual neurons at the level of morphology and connectivity.

(2) The work is very comprehensive and results in a near complete map of all head BMNs.

(3) Authors also complement this anatomical characterization with a first-level functional analysis using optogenetic activation of BMNs that results in expected directed grooming behavior.

Weaknesses:

(1) While not strictly needed here, it could help provide context if authors revealed some of the important downstream pathways that could explain optogenetics behavioral phenotypes: This point was addressed by authors in the revisions and I agree a detailed description of downstream circuits is not needed at this point.

(2) In contrast to the rigorous quantitative analysis of the anatomical data, the behavioral data is analyzed using much more subjective methods. While I do not think it is necessary to perform a rigorous analysis of behaviors in this anatomy focused manuscript, the conclusions based on behavioral analysis should be treated as speculative in the current form e.g. calling "nodding + backward motions" as an avoidance response is not justified as it currently stands. Strong optogenetic activation could lead to sudden postural changes that due to purely biomechanical constraints could lead to a couple of backward steps as seen in the example videos. Moreover since the quantification is manual, it is not clear what the analyst interprets as backward walking or nodding. Interpretation is also concerning because controls show backward walking (although in fewer instances based on subjective quantification): This point was addressed by the authors during revisions and I'm mostly satisfied with their response, where authors agree that the behavioral results are currently used to speculate about the role of BMNs in aversive behaviors. Still, the fact that controls show some "backward motions" is a bit concerning when talking about "significant differences" between control and test groups based on manual annotations and I would recommend future studies focusing on these behaviors to use more unbiased quantitative analysis wherever possible.

Summary:

The authors end up generating a near-complete map of head BMNs that will serve as a long-standing resource to the *Drosophila* research community. This will directly shape future experiments aimed at modeling or functionally analyzing the head grooming circuit to understand how somatotopy guides behaviors. I appreciate the authors taking the time to revise the manuscript and address reviewer concerns.

---

## [Referee Report · Reviewer #3 (Public Review)]

Eichler et al. set out to catalog the mechanosensory bristles of the fly head in an effort to understand the extent to which their organization is consistent with the parallel model of hierarchical suppression in the context of grooming behavior. They map the locations of the mechanosensory bristles on the fly head, examine the axonal morphology of the bristle mechanosensory neurons (BMNs) that innervate them, and match these to electron microscopy reconstructions of the same BMNs in a previously published EM volume of the female adult fly brain. They use BMN synaptic connectivity information to create clusters of BMNs that they show occupy different regions of the subesophageal zone brain region and use optogenetic activation of subsets of BMNs to evaluate the behaviors evoked by specific activation of BMN subpopulations innervating the head.

The authors have beautifully cataloged the mechanosensory bristles and the projection paths and patterns of the corresponding BMN axons in the brain using detailed and painstaking methods. The result is a neuroanatomy resource that will be an important community resource. To match BMNs reconstructed in an electron microscopy volume of the adult fly brain, the authors matched clustered reconstructed BMNs with light-level BMN classes observed using precise dye-fills and stochastic labeling techniques. The authors then employ a variety of clustering methods to demonstrate that BMN populations that innervate different regions of the head project into the subesophageal zone and terminate in distinctive yet, in some cases, partially overlapping zones. By clustering BMNs on the basis of their synaptic partners, the authors find that BMNs from distant areas of the head have non-overlapping synaptic partners while those from neighbor areas have overlapping synaptic partners. This result calls into question the scale at which the parallel model of hierarchical suppression may be operating. Finally, the authors use tools that were generated during the light-level characterization of BMN projections to show that activating BMNs that innervate specific areas of the head leads to grooming of the innervated regions and neighboring regions, consistent with the observed overlap in downstream circuits between BMNs innervating neighboring regions of the head. This result suggests that while the parallel model could be operating on a broad scale, additional circuit mechanisms may be operating on a finer scale to produce grooming of the area surrounding the source of mechanosensory input.

This work will have a positive impact on the field by contributing a complete accounting of the mechanosensory bristles of the fruit fly head, describing the brain projection patterns of the BMNs that innervate them, and linking them to BMN sensory projections in an electron microscopy volume of the adult fly brain. It will also have a positive impact on the field by providing genetic tools to help functionally subdivide the contributions of different BMN populations to circuit computations and behavior. This contribution will pave the way for further mechanistic study of central circuits that subserve grooming circuits.

---

## [Author Response]

The following is the authors’ response to the original reviews.

**eLife assessment**
This valuable work provides a near-complete description of the mechanosensory bristles on the *Drosophila melanogaster* head and the anatomy and projection patterns of the bristle mechanosensory neurons that innervate them. The data presented are solid. The study has generated numerous invaluable resources for the community that will be of interest to neuroscientists in the field of circuits and behaviour, particularly those interested in mechanosensation and behavioural sequence generation.

We express our gratitude to the Reviewers for their valuable suggestions, which significantly enhanced the manuscript. The revisions were undertaken, not with the expectation of acceptance, but rather driven by our sincere belief that these revisions would enhance the manuscript's impact for future readers.

**Public Reviews:**

**Reviewer #1 (Public Review):**
Sensory neurons of the mechanosensory bristles on the head of the fly project to the sub esophageal ganglion (SEZ). In this manuscript, the authors have built on a large body of previous work to comprehensively classify and quantify the head bristles. They broadly identify the nerves that various bristles use to project to the SEZ and describe their region-specific innervation in the SEZ. They use dye-fills, clonal labelling, and electron microscopic reconstructions to describe in detail the phenomenon of somatotopy - conserved peripheral representations within the central brain - within the innervation of these neurons. In the process they develop novel tools to access subsets of these neurons. They use these to demostrate that groups of bristles in different parts of the head control different aspects of the grooming sequence.
**Reviewer #2 (Public Review):**
The authors combine genetic tools, dye fills and connectome analysis techniques to generate a "first-of-its-kind", near complete, synaptic resolution map of the head bristle neurons of *Drosophila*. While some of the BMN anatomy was already known based on previous work by the authors and other researchers, this is the first time a near complete map has been created for the head BMNs at electron microscopy resolution.Strengths:(1) The authors cleverly use techniques that allow moving back and forth between periphery (head bristle location) and brain, as well as moving between light microscopy and electron microscopy data. This allows them to first characterize the pathways taken by different head BMNs to project to the brain and also characterize anatomical differences among individual neurons at the level of morphology and connectivity.(2) The work is very comprehensive and results in a near complete map of all I’m head BMNs.(3) Authors also complement this anatomical characterization with a first-level functional analysis using optogenetic activation of BMNs that results in expected directed grooming behavior.Weaknesses:(1) The clustering analysis is compelling but cluster numbers seem to be arbitrarily chosen instead of by using some informed metrics.

We made revisions to the manuscript that address this concern. Please see our response to “recommendations for authors” for a description of these revisions.

(2) It could help provide context if authors revealed some of the important downstream pathways that could explain optogenetics behavioral phenotypes and previously shown hierarchical organization of grooming sequences.

We made revisions to the manuscript that address this recommendation. Please see our response to “recommendations for authors” for a description of these revisions.

(3) In contrast to the rigorous quantitative analysis of the anatomical data, the behavioral data is analyzed using much more subjective methods. While I do not think it is necessary to perform a rigorous analysis of behaviors in this anatomy focused manuscript, the conclusions based on behavioral analysis should be treated as speculative in the current form e.g. calling "nodding + backward walking" as an avoidance response is not justified as it currently stands. Strong optogenetic activation could lead to sudden postural changes that due to purely biomechanical constraints could lead to a couple of backward steps as seen in the example videos. Moreover since the quantification is manual, it is not clear what the analyst interprets as backward walking or nodding. Interpretation is also concerning because controls show backward walking (although in fewer instances based on subjective quantification).

While unbiased machine vision-based methods would nicely complement the present work, this type of analysis is not yet working to distinguish between different head grooming movements. Therefore, we are currently limited to manual annotation for our behavioral analysis. That said, we do not believe that our manual annotation is subjective. The grooming movements that we examine in this work are distinguishable from each other through frame-by-frame manual annotation of video at 30 fps. Our annotation of the grooming and backward motions performed by flies are based on previous publications that established a controlled vocabulary defining each movement (Hampel et al., 2020a, 2017, 2015; Seeds et al., 2014). In this work, we added head nodding to this controlled vocabulary that is described in the Materials and methods. We have added additional text to the third paragraph of the Material and methods section entitled “Behavioral analysis procedures” that we hope better describes our behavioral analysis. This description now reads:

Head *nodding* was annotated when the fly tilted its head downward by any amount until it returned its head back in its original position. This movement often occurred in repeated cycles. Therefore, the “start” was scored at the onset of the first forward movement and the “stop” when the head returned to its original position on the last nod.

We do not make any firm conclusions about the head movements (nodding) and backwards motions. We refer to nodding as a descriptive term that would allow the reader to better understand what the behavior looks like. We make no firm conclusions about any behavioral functional role that either the nodding or the backward motions might have, with the exception of nodding in the context of grooming. We only suggest that the behaviors appear to be avoidance responses. Furthermore, backward walking was not mentioned. Instead we refer to backward motions. We are only reporting our annotations of these movements that do occur, and are significantly different from controls. We speculate that these could be avoidance responses based on support from the literature. Future studies will be required to understand whether these movements serve real behavioral roles.

Summary:The authors end up generating a near-complete map of head BMNs that will serve as a long-standing resource to the *Drosophila* research community. This will directly shape future experiments aimed at modeling or functionally analyzing the head grooming circuit to understand how somatotopy guides behaviors.
**Reviewer #3 (Public Review):**
Eichler et al. set out to map the locations of the mechanosensory bristles on the fly head, examine the axonal morphology of the bristle mechanosensory neurons (BMNs) that innervate them, and match these to electron microscopy reconstructions of the same BMNs in a previously published EM volume of the female adult fly brain. They used BMN synaptic connectivity information to create clusters of BMNs that they show occupy different regions of the subesophageal zone brain region and use optogenetic activation of subsets of BMNs to support the claim that the morphological projections and connectivity of defined groups of BMNs are consistent with the parallel model for behavioral sequence generation.The authors have beautifully cataloged the mechanosensory bristles and the projection paths and patterns of the corresponding BMN axons in the brain using detailed and painstaking methods. The result is a neuroanatomy resource that will be an important community resource. To match BMNs reconstructed in an electron microscopy volume of the adult fly brain, the authors matched clustered reconstructed BMNs with light-level BMN classes using a variety of methods, but evidence for matching is only summarized and not demonstrated in a way that allows the reader to evaluate the strength of the evidence. The authors then switch from morphology-based categorization to non-BMN connectivity as a clustering method, which they claim demonstrates that BMNs form a somatotopic map in the brain. This map is not easily appreciated, and although contralateral projections in some populations are clear, the distinct projection zones that are mentioned by the authors are not readily apparent. Because of the extensive morphological overlap between connectivity-based clusters, it is not clear that small projection differences at the projection level are what determines the post-synaptic connectivity of a given BMN cluster or their functional role during behavior. The claim the somatotopic organization of BMN projections is preserved among their postsynaptic partners to form parallel sensory pathways is not supported by the result that different connectivity clusters still have high cosine similarity in a number of cases (i.e. Clusters 1 and 3, or Clusters 1 and 2). Finally, the authors use tools that were generated during the light-level characterization of BMN projections to show that specifically activating BMNs that innervate different areas of the head triggers different grooming behaviors. In one case, activation of a single population of sensory bristles (lnOm) triggers two different behaviors, both eye and dorsal head grooming. This result does not seem consistent with the parallel model, which suggests that these behaviors should be mutually exclusive and rely on parallel downstream circuitry.

We made revisions to the manuscript that address this recommendation. Please see our response to “recommendations for authors” for a description of these revisions.

This work will have a positive impact on the field by contributing a complete accounting of the mechanosensory bristles of the fruit fly head, describing the brain projection patterns of the BMNs that innervate them, and linking them to BMN sensory projections in an electron microscopy volume of the adult fly brain. It will also have a positive impact on the field by providing genetic tools to help functionally subdivide the contributions of different BMN populations to circuit computations and behavior. This contribution will pave the way for further mechanistic study of central circuits that subserve grooming circuits.
**Recommendations for the authors:**
All three reviewers appreciated the work presented in this manuscript. There were also a few overlapping concerns that were raised that are summarised below, should the authors wish to address them:Somatotopy: We recommend that the authors describe the extent of prior knowledge in more detail to highlight their contribution better.

We made revisions that better highlight the extent of prior knowledge about somatotopy. We describe how previous studies showed bristle mechanosensory neurons in insects are somatotopically organized, but these studies were not comprehensive descriptions of complete somatotopic maps for the head or body. To our knowledge, our study provides the first comprehensive and synaptic resolution somatotopic map of a head for any animal. This sets the stage for the complete definition of the interface between somatotopically-organized mechanosensory neurons and postsynaptic circuits, which has broad implications for future studies on aimed grooming, and mechanosensation in general. Below we itemize revisions to the Introduction, Discussion, and Figures to provide a clearer statement of the significance of our study as it relates to somatotopy.

(1) Newly added Figure 1 – figure supplement 1 more explicitly grounds the study in somatotopy, providing a working model of the organization of the circuit pathways that produce the grooming sequence. This model features somatotopy as shown in Figure 1 – figure supplement 1C.

(2) Figure 1 – figure supplement 1 is incorporated into the Introduction in the second, third, and fourth paragraphs, the first paragraph of the Results section titled “Somatotopically-organized parallel BMN pathways”, and the second and third paragraphs of the last Discussion section titled “Parallel circuit architecture underlying the grooming sequence”.

(3) We added text to the end of the fourth paragraph of the Introduction that now reads: “In this model, parallel-projecting mechanosensory neurons that respond to stimuli at specific locations on the head or body could connect with somatotopically-organized parallel circuits that elicit grooming of those locations (Figure 1 – figure supplement 1A-C). The previous discovery of a mechanosensory-connected circuit that elicits aimed grooming of the antennae provides evidence of this organization (Hampel 2015). However, the extent to which distinct circuits elicit grooming of other locations is unknown, in part, because the somatotopic projections of the mechanosensory neurons have not been comprehensively defined for the head or body.”

(4) There is a Discussion section that further explains the extent of prior knowledge and our contributions on somatotopy that is titled “A synaptic resolution somatotopic map of the head BMNs”. Additionally, the previous version of this section had a paragraph on the broader implications of our work as it relates to somatotopy across species. In light of the reviewer comments, we decided to make this paragraph into its own Discussion section to better highlight the broader significance of our work. This section is titled “First synaptic resolution somatotopic map of the head”.

The somatotopy isn't overtly obvious - perhaps they could try mapping presynaptic sites and provide landmarks to improve visualisation.

We made the following revisions to better highlight the head BMN somatotopy. One point of confusion from the previous manuscript version stemmed from us not explicitly defining the somatotopic organization that we observed. There seemed to be confusion that we were defining the head somatotopy based only on the small projection differences among BMNs from neighboring head locations. While we believe that these small differences indeed correspond to somatotopy, we failed to highlight that there are overt differences in the brain projections of BMNs from distant locations on the head. For example, Figure 5B (right panel) shows the distinct projections between the LabNv (brown) and AntNv (blue) BMNs that innervate bristles on the ventral and dorsal head, respectively. Thus, BMN types innervating neighboring bristles show overlapping projections with small projection differences, whereas those innervating distant bristles show non overlapping projections into distinct zones.

Our analysis of postsynaptic connectivity similarity also shows somatotopic organization among the BMN postsynaptic partners, as BMN types innervating the same or neighboring bristle populations show high connectivity similarity (Figure 8, old Figure 7). Below we highlight major revisions to the text and Figures that hopefully better reveal the head somatotopy.

(1) In the last paragraph of the Introduction we added text that explicitly frames the experiments in terms of somatotopic organization: “This reveals somatotopic organization, where BMNs innervating neighboring bristles project to the same zones in the CNS while those innervating distant bristles project to distinct zones. Analysis of the BMN postsynaptic connectome reveals that neighboring BMNs show higher connectivity similarity than distant BMNs, providing evidence of somatotopically organized postsynaptic circuit pathways.”

(2) We mention an example of overt somatotopy from Figure 5 in the Results section titled “EM-based reconstruction of the head BMN projections in a full adult brain”. The text reads “For example, BMNs from the Eye- and LabNv have distinct ventral and anterior projections, respectively. This shows how the BMNs are somatotopically organized, as their distinct projections correspond to different bristle locations on the head (Figure 5B,C).”

(3) In new Figure 8 (part of old Figure 7), we modified panels that correspond to the cosine similarity analysis of postsynaptic connectivity. The major revision was to plot the cosine similarity clusters onto the head bristles so that the bristles are now colored based on their clusters (C). This shows how neighboring BMNs cluster together, and therefore show similar postsynaptic connectivity. We believe that this provides a nice visualization of somatotopic organization in BMN postsynaptic connectivity. We also added the clustering dendrogram as recommended by Reviewer #2 (Figure 8A).

(4) In new Figure 8, we added new panels (D-F) that summarize our anatomical and connectomic analysis showing different somatotopic features of the head BMNs. Different BMN types innervate bristles at neighboring and distant proximities (D). BMNs that innervate neighboring bristles project into overlapping zones (E, example of reconstructed BM-Fr and -Ant neurons with non-overlapping BM-MaPa neurons) and show postsynaptic connectivity similarity (F, example connectivity map of three BM types on cosine similarity data).

(5) To accompany the new Figure 8D-F panels, we added a paragraph to summarize the different somatotopic features of the head BMNs that were identified based on our anatomical and connectomic analysis. This is the last paragraph in the Results section titled “Somatotopically-organized parallel BMN pathways”:

Our results reveal head bristle proximity-based organization among the BMN projections and their postsynaptic partners to form parallel mechanosensory pathways. BMNs innervating neighboring bristles project into overlapping zones in the SEZ, whereas those innervating distant bristles project to distinct zones (example of BM-Fr, -Ant, and -MaPa neurons shown in Figure 8D,E). Cosine similarity analysis of BMN postsynaptic connectivity revealed that BMNs innervating the same bristle populations (same types) have the highest connectivity similarity. Figure 8F shows example parallel connections for BM-Fr, -Ant, and -MaPa neurons (vertical arrows), where the edge width indicates the number of synapses from each BMN type to their major postsynaptic partners. Additionally, BMNs innervating neighboring bristle populations showed postsynaptic connectivity similarity, while BMNs innervating distant bristles show little or none. For example, BM-Fr and -Ant neurons have connections to common postsynaptic partners, whereas BM-MaPa neurons show only weak connections with the main postsynaptic partners of BM-Fr or -Ant neurons (Figure 8F, connections under 5% of total BMN output omitted). These results suggest that BMN somatotopy could have different possible levels of head spatial resolution, from specific bristle populations (e.g. Ant bristles), to general head areas (e.g. dorsal head bristles).

We also refer to Figure 8D-F to illustrate the different somatotopic features in the Discussion. These references can be found in the following Discussion sections titled “A synaptic resolution somatotopic map of the head BMNs (fourth paragraph)”, and “Parallel circuit architecture underlying the grooming sequence (second paragraph)”.

(6) In addition to improving the Figures, we provide additional tools that enable readers to explore the BMN somatotopy in a more interactive way. That is, we provide 5 different FlyWire.ai links in the manuscript Results section that enable 3D visualization of the different reconstructed BMNs (e.g. FlyWire.ai link 1).

Note: In working on old Figure 7 to address this Reviewer suggestion, we also reordered panels A-E. We believe that this was a more logical ordering than in the previous draft. These panels are now the only data shown in Figure 7, as the cosine similarity analysis is now in Figure 8. We hope that splitting these panels into two Figures will improve manuscript readability.

Light EM Mapping: A better description of methods by which this mapping was done would be helpful. Perhaps the authors could provide a few example parallel representations of the EM and light images in the main figure would help the reader better appreciate the strength of their approach.

We have done as the Reviewers suggested and added panels to Figure 6 that show examples of the LM and EM image matching (Figure 6A,B). We added two examples that used different methods for labeling the LM imaged BMNs, including MCFO labeling of an individual BM-InOc neuron and driver line labeling of a major portion of BM-InOm neurons using InOmBMN-LexA. These panels are referred to in the first paragraph of the Results section titled “Matching the reconstructed head BMNs with their bristles”. Note that examples for all LM/EM matched BMN types are shown in Figure 6 – figure supplement 2.

We had provided Figure 6 – figure supplement 2 in the reviewed manuscript that shows all the above requested “parallel representations of the EM and light images”. However, the Reviewer critiques made us realize that the purpose of this figure supplement was not clearly indicated. Therefore, we have revised Figure 6 – figure supplement 2 and its legend to make its purpose clearer. First, we changed the legend title to better highlight its purpose. The legend is now titled: “Matching EM reconstructed BMN projections with light microscopy (LM) imaged BMNs that innervate specific bristles”. Second, we added label designations to the figure panel rows that highlight the LM and EM comparisons. That is, the rows for light microscopy images of BMNs are indicated with LM and the rows for EM reconstructed BMN images are labeled with EM. Reviewer #3 had indicated that it was not clear what labeling methods were used to visualize the LM imaged BM-InOm neurons in Figure 6 – figure supplement 2N. Therefore, we added text to the figure and the legend to better highlight the different methods used. Panels A and B were also cropped to accommodate the above mentioned revisions.

The manuscript also provides an extensive Materials and methods section that describes the different lines of evidence that were used to assign the reconstructed BMNs as specific types. We changed the title to better highlight the purpose of this methods section to “Matching EM reconstructed BMN projections with light microscopy imaged BMNs that innervate specific bristles”. The evidence used to support the assignment of the different BMN types is also summarized in Figure 6 – figure supplement 3.

Parallel circuit model: The authors motivate their study with this. We're recommending that they define expectations of such circuitry, its alternatives (including implications for downstream pathways), and behavior before they present their results. We're also recommending that they interpret their behavioural results in the context of these circuits.

Our primary motivation for doing the experiments described in this manuscript was to help define the neural circuit architecture underlying the parallel model that drives the *Drosophila* grooming sequence. This manuscript provides a comprehensive assessment of the first layer of this circuit architecture. A byproduct of this work is a contribution that offers immediate utility and significance to the *Drosophila* connectomics community. Namely, the description of the majority of mechanosensory neurons on the head, with their annotation in the recently released whole brain connectome dataset (FlyWire.ai). In writing this manuscript, we tried to balance both of these things, which was difficult to write. We very much appreciate the Reviewers' comments that have highlighted points of confusion in our original draft. We hope that the revised draft is now clearer and more logically presented. We have made revisions to the text and provided a new figure supplement (Figure 1 - figure supplement 1) and new panels in Figure 8. Below we highlight the major revisions.

(1) The Introduction was revised to more explicitly ground the study in the parallel model, while also removing details that were not pertinent to the experiments presented in the manuscript.

The first paragraph introduces different features of the parallel model. To better focus the reader on the parts of the model that were being assessed in the manuscript, we removed the following sentences: “Performance order is established by an activity gradient among parallel circuits where earlier actions have the highest activity and later actions have the lowest. A winner-take-all network selects the action with the highest activity and suppresses the others. The selected action is performed and then terminated to allow a new round of competition and selection of the next action.” Note that these sentences are included in the third and fourth paragraphs of the last Discussion section titled “Parallel circuit architecture underlying the grooming sequence”.

The first paragraph of the Introduction now introduces a bigger picture view of the model that emphasizes the two main features: (1) a parallel circuit architecture that ensures all mutually exclusive actions to be performed in sequence are simultaneously readied and competing for output, and (2) hierarchical suppression among the parallel circuits, where earlier actions suppress later actions.

(2) Newly added Figure 1 – figure supplement 1 provides a working model of grooming (Reviewer # 1 suggestion). We now more strongly emphasize that the study aimed to define the parallel neural circuit architecture underlying the grooming sequence, focusing on the mechanosensory layer of this architecture. In particular, we refer to the new Figure 1 – figure supplement 1 that has been added to better convey the hypothesized grooming neural circuit architecture. Figure 1 – figure supplement 1 is incorporated into the Introduction (paragraphs two, three, and four), Results section titled “Somatotopically-organized parallel BMN pathways (first paragraph)”, and last Discussion section titled “Parallel circuit architecture underlying the grooming sequence (second and third paragraphs)”.

(3) New panels in Figure 8 update the model of parallel circuit organization as it relates to somatotopy (D-F). These panels show the parallel circuits hypothesized by the model, but also indicate convergence, with different possible levels of head resolution for these circuits. We describe above where these panels are referenced in the text.

(4) We added a new paragraph in the last Discussion section titled “Parallel circuit architecture underlying the grooming sequence” that better incorporates the results from this manuscript into the working model of grooming. This paragraph is shown below.

Here we define the parallel architecture of BMN types that elicit the head grooming sequence that starts with the eyes and proceeds to other locations, such as the antennae and ventral head. The different BMN types are hypothesized to connect with parallel circuits that elicit grooming of specific locations (described above and shown in Figure 1 – figure supplement 1A,C). Indeed, we identify distinct projections and connectivity among BMNs innervating distant bristles on the head, providing evidence supporting this parallel architecture (Figure 8D-F). However, we also find partially overlapping projections and connectivity among BMNs innervating neighboring bristles. Further, optogenetic activation of BMNs at specific head locations elicits grooming of both those locations and neighboring locations (Figure 9). These findings raise questions about the resolution of the parallel architecture underlying grooming. Are BMN types connected with distinct postsynaptic circuits that elicit aimed grooming of their corresponding bristle populations (e.g. Ant bristles)? Or are neighboring BMN types that innervate bristles in particular head areas connected with circuits that elicit grooming of those areas (e.g. dorsal or ventral head)? Future studies of the BMN postsynaptic circuits will be required to define the resolution of the parallel pathways that elicit aimed grooming.

Aside from this summary of major concerns, the detailed recommendations are attached below.
**Reviewer #1 (Recommendations For The Authors):**
I appreciate the quality and exhaustive body of work presented in this manuscript. I have a few comments that the authors may want to consider:(1) The authors motivate this study by posing that it would allow them to uncover whether the complex grooming behaviour of flies followed a parallel model of circuit function. It would have been nice to have been introduced to what the alternative model might be and what each would mean for organisation of the circuit architecture. Some guiding schematics would go a long way in illustrating this point. Modifying the discussion along these lines would also be helpful.

We made several revisions to the manuscript that address this recommendation. Among these revisions, we added Figure 1 – figure supplement 1 that includes a working model for grooming. Please see above for a description of these revisions.

(2) The authors mention the body of work that has mapped head bristles and described somatotopy. It would be useful to discuss in more detail what these studies have shown and highlight where the gaps are that their study fills.

We made several revisions to the manuscript that address this recommendation. Please see above for a description of these revisions.

(3) The dye-fills and reconstructions that are single colour could use a boundary to demarcate the SEZ. This would help in orienting the reader.

We agree with Reviewer #1 that Figure 4 and its supplements could use some indicator that would orient the reader with respect to the dye filled or stochastically labeled neurons. The images are of the entire SEZ in the ventral brain, and in the case of some panels, the background staining enables visualization of the brain e.g. Figure 4H,M,N. To help orient the reader in this region, we added a dotted line to indicate the approximate SEZ midline. This also enables the reader to more clearly see which of the BMN types cross the midline.

Midline visual guides were added for Figure 4, Figure 4 – figure supplement 2, Figure 4 – figure supplement 3, Figure 4 – figure supplement 4, Figure 4 – figure supplement 5, Figure 4 – figure supplement 6, Figure 4 – figure supplement 7, Figure 4 – figure supplement 8, Figure 6 – figure supplement 2.

(4) The comparison between the EM and the fills/clones are not obvious. And particularly because they are not directly determined, it would be nice to have the EM reconstruction alongside the dye-fills. This would work very nicely in the supplementary figure with the multiple fills of the same bristles. I think this would really drive home the point.

We made several revisions to the manuscript that address this recommendation. Please see above for a description of these revisions.

(5) Are there unnoticed black error-bars floating around in many of the gray-scale images?

The black bars were masking white scale bars in the images. We have removed the black bars and remade the images without scale bars. This was done for the following Figures: Figure 4, Figure 4 – figure supplement 2, Figure 4 – figure supplement 3, Figure 4 – figure supplement 4, Figure 4 – figure supplement 5, Figure 4 – figure supplement 6, Figure 4 – figure supplement 7, Figure 4 – figure supplement 8, Figure 6 – figure supplement 2.

**Reviewer #2 (Recommendations For The Authors):**
(1) The only point in the paper I found myself going back and forth between methods/supp and text was when authors discuss about the clustering. I think it would help the reader if a few sentences about cosine clustering used for connectivity based clustering were included in the main text. Also, for NBLAST hierarchical clustering, it would help if some informed metrics could be used for defining cluster numbers (e.g. Braun et al, 2010 PLOS ONE shows how Ward linkage cost could be used for hierarchical clustering).

Depending on where the cut height is placed on the dendrogram for cosine similarity of BMNs, different features of the BMN type postsynaptic connectivity are captured. As the number of clusters is increased (lower cut height), clustering is mainly among BMNs of the same type, showing that these BMNs have the highest connectivity similarity. As the number of clusters is reduced (higher cut height), BMNs innervating neighboring bristles on the head are clustered, revealing three general clusters corresponding to the dorsal, ventral, and posterior head. This reveals somatotopy based clustering among same and neighboring BMN types. The cut height shown in Figure 8 and Figure 8 – figure supplement 2 was chosen because it highlighted both of these features.

The NBLAST clustering shows similar results to the connectivity based clustering with respect to neighboring and distant BMN types. As the number of clusters increases BMNs of the same type are clustered, and these types can be further subdivided into morphologically distinct subtypes. As the number of clusters is reduced, the clustering captures neighboring BMNs. Thus, neighboring BMN types showed high morphology similarity (and proximity) with each other, and low similarity with distant BMN types.

Please see our responses to a Reviewer #3 critique below for further description of the clustering results.

On the same lines it would help if the clustering dendrograms were included in the main figure.

We thank Reviewer #2 for this comment. We have added the dendrogram to Figure 8A, a change that we feel makes this Figure much easier to understand.

(2) It could help provide intuition if the authors revealed some of the downstream targets and their implication in explaining the behavioral phenotypes.

While this will be the subject of at least two forthcoming manuscripts, we have added text to the present manuscript that provides insight into BMN postsynaptic targets. Our previous work (Hampel et al. 2015) described a mechanosensory connected neural circuit that elicits grooming of the antennae. While this previous study demonstrated that the Johnston’s organ mechanosensory neurons are synaptically and functionally connected with this circuit, our preliminary analysis indicates that it is also connected with BM-Ant neurons. We hypothesize that there are additional such circuits that are responsible for eliciting grooming of other head locations.

To better highlight potential downstream targets in the manuscript, we now mention the antennal circuit in the Introduction. This text reads: In this model, parallel-projecting mechanosensory neurons that respond to stimuli at specific locations on the head or body could connect with somatotopically-organized parallel circuits that elicit grooming of those locations (Figure 1 – figure supplement 1A-C). The previous discovery of a mechanosensory-connected circuit that elicits aimed grooming of the antennae provides evidence of this organization (Hampel 2015). However, the extent to which distinct circuits elicit grooming of other locations is unknown, in part, because the somatotopic projections of the mechanosensory neurons have not been comprehensively defined for the head or body.

There is also text in the Discussion that addresses this Reviewer comment. It describes the antennal circuit and mentions the possibility that other similar circuits may exist. This can be found in the third paragraph of the section titled “Circuits that elicit aimed grooming of specific head locations”.

(3) Authors find that opto activation of BMNs leads to grooming of targeted as well as neighboring areas. Is there any sequence observed here? i.e. first clean targeted area and then clean neighboring area? I wonder if the answer to this is something as simple as common post-synaptic targets which is essentially reducing the resolution of the BMN sensory map. Some more speculation on this interesting result could be helpful.

We appreciate and agree with this point from Reviewer #2, and have tried to better emphasize the possible implications for grooming that the overlapping projections and connectivity among BMNs innervating neighboring bristles may have. This is now better addressed in the Results and Discussion sections. Below we highlight where this is addressed:

(1) In the second paragraph of the Results section titled “Activation of subsets of head BMNs elicits aimed grooming of specific locations” we added text that suggests the possibility that grooming of the stimulated and neighboring locations could be due to the overlapping projections and connectivity. This text reads: This suggested that head BMNs elicit aimed grooming of their corresponding bristle locations, but also neighboring locations. This result is consistent with our anatomical and connectomic data indicating that BMNs innervating neighboring bristles show overlapping projections and postsynaptic connectivity similarity (see Discussion).

(2) In the fourth paragraph of the Discussion section titled “A synaptic resolution somatotopic map of the head BMNs”, we added a sentence to the end of the fourth paragraph that alludes to further discussion of this topic. This sentence reads: This overlap may have implications for aimed grooming behavior. For example, neighboring BMNs could connect with common neural circuits to elicit grooming of overlapping locations (discussed more below).

(3) In the fourth paragraph of the Discussion section titled “Circuits that elicit aimed grooming of specific head locations” there is a paragraph that mentions the possibility of mechanosensory convergence onto common postsynaptic circuits to promote grooming of the stimulated area, along with neighboring areas. This paragraph is below.

We find that activation of specific BMN types elicits both aimed grooming of their corresponding bristle locations and neighboring locations. This suggests overlap in the locations that are groomed with the activation of different BMN types. Such overlap provides a means of cleaning the area surrounding the stimulus location. Interestingly, our NBLAST and cosine similarity analysis indicates that neighboring BMNs project into overlapping zones in the SEZ and show common postsynaptic connectivity. Thus, we hypothesize that neighboring BMNs connect with common neural circuits (e.g. antennal grooming circuit) to elicit overlapping aimed grooming of common head locations.

(4) In the new second paragraph of the Discussion section titled “Parallel circuit architecture underlying the grooming sequence” we further discuss the issue of the BMN “sensory map. This paragraph is below.

Here we define the parallel architecture of BMN types that elicit the head grooming sequence that starts with the eyes and proceeds to other locations, such as the antennae and ventral head. The different BMN types are hypothesized to connect with parallel circuits that elicit grooming of specific locations (described above and shown in Figure 1 – figure supplement 1A,C). Indeed, we identify distinct projections and connectivity among BMNs innervating distant bristles on the head, providing evidence supporting this parallel architecture (Figure 8D-F). However, we also find partially overlapping projections and connectivity among BMNs innervating neighboring bristles. Further, optogenetic activation of BMNs at specific head locations elicits grooming of both those locations and neighboring locations (Figure 9). These findings raise questions about the resolution of the parallel architecture underlying grooming. Are BMN types connected with distinct postsynaptic circuits that elicit aimed grooming of their corresponding bristle populations (e.g. Ant bristles)? Or are neighboring BMN types that innervate bristles in particular head areas connected with circuits that elicit grooming of those areas (e.g. dorsal or ventral head)? Future studies of the BMN postsynaptic circuits will be required to define the resolution of the parallel pathways that elicit aimed grooming.

(4) If authors were to include a summary table that shows all known attributes about BMN type as columns that could be very useful as a resource to the community. Table columns could include attributes like "bristle name", "nerve tract", "FlyWire IDs of all segments corresponding to the bristle class". "split-Gal4 line or known enhancer" , etc.

We provided a table that includes much of this information after the manuscript had already gone out for review. We regret that this was not available. This is now provided as Supplementary file 3. This table provides the following information for each reconstructed BMN: BMN name, bristle type, nerve, flywire ID, flywire coordinates, NBLAST cluster (cut height 1), NBLAST cluster (cut height 5), and cosine cluster (cut height 4.5). Note that the driver line enhancers for targeting specific BMN types are shown in Figure 3I.

Specific Points:Figure 4C-V:I find it a bit difficult to distinguish ipsi- from contra-lateral projections. Maybe indicate the midline as a thin, stippled line?

We thank the Reviewer #2 for this suggestion. We have now added lines in the panels in Figure 4C-V to indicate the approximate location of the midline. We also added lines to the Figure 4 – figure supplements as described above.

I think this Fig reference is wrong "the red-light stimulus also elicited backward motions with control flies (Figure 6B,C, control, black trace, Video 5)." should be Fig 8B,C

We have fixed this error.

**Reviewer #3 (Recommendations For The Authors):**
Introduction:Motivating this study in terms of understanding the neural mechanisms that execute the parallel model seems to overstate what you will achieve with the current study. If you want to motivate it this way, I suggest focusing on the grooming sequence of the head along (eyes, antennae, proboscis).

We made several revisions to the manuscript that address this recommendation. Please see above for a description of these revisions. Please note that many of the revisions focus on the head grooming sequence. We also made minor revisions to the Introduction that further emphasize the focus on head grooming.

Results:Figure 1. Please indicate that this is a male fly in either the figure title or in the figure itself.

We added a male symbol to Figure 1A.

Figure 3. Panel J is referenced in the main body text and in the figure caption, but there is no Fig 3J.

Panel J is shown in the upper right corner of Figure 3. We realize that the placement of this panel is not ideal, but this was the only place that we could fit it. Additionally, the panel works nicely at that location to better enable comparison with panel C. We have revised the text in the Figure 3 legend to better highlight the location of this Figure panel: “Shown in the upper right corner of the figure are the aligned expression patterns of InOmBMN-LexA (red), dBMN-spGAL4 (green), and TasteBMN-spGAL4 (brown).”

We also added text to a sentence in the results section entitled “Head BMNs project into discrete zones in the ventral brain” that indicates the panel location. This text reads: To further visualize the spatial relationships between these projections, we computationally aligned the expression patterns of the different driver lines into the same brain space (Figure 3J, upper right corner).

Matching the BMNs to EM reconstructions: why cut the dendrogram at H=5? Would be better to determine cluster number using an unbiased method.

To match the morphologically distinct EM reconstructed BMNs to their specific bristles, we relied on different lines of evidence, including NBLAST results (discussed more below), dye fill/stochastic labeling/driver line labeling matches, published morphology, nerve projection, bristle number, proximity to other BMNs, and postsynaptic connectivity (summarized in Figure 6 – figure supplement 3). The following Materials and methods section provides a detailed description of the evidence used to assign each BMN type in “Matching EM reconstructed BMN projections with light microscopy imaged BMNs that innervate specific bristles”. In many cases, BMN type could be assigned with confidence solely based on morphological comparisons with our light level data (e.g. dye fills), in conjunction with bristle counts to indicate an expected number of BMNs showing similar morphology. Thus, the LM/EM matches and NBLAST clustering were largely complementary.

The EM reconstructed BMNs were matched as particular BMN types, in part based on examination of the NBLAST data at different cut heights. NBLAST clustering of the BMNs revealed general trends at higher and lower cut heights (Figure 6 – figure supplement 1A, Supplementary file 3). The lowest cut heights included mostly BMNs of the same type innervating the same bristle populations, and smaller clusters that subdivided into morphologically distinct subtypes (see Supplementary file 3 for clusters produced at cut height 1). This revealed that BMNs of the same type tended to show the highest morphological similarity with each other, but they also showed intratype morphological diversity. Higher cut heights produced clusters of BMNs innervating neighboring bristles populations (e.g. ventral head BMNs), showing high morphological similarity among neighboring BMN types.

We selected the cut height 5 shown in Figure 6 – figure supplement 1A,B because it captures examples of both same and neighboring type clustering. For example, it captures a cluster of mostly BM-Taste neurons (Cluster 16), and neighboring BMN types, including those from the dorsal head (Cluster 14) or ventral head (Cluster 15).

Based on reviewer comments, we realized that the way we wrote the BMN matching section in the Results indicated more reliance on the NBLAST clustering than what was actually necessary, distorting the way we actually matched the BMNs. Therefore, we softend the first couple of sentences to place less emphasis on the importance of the NBLAST. We also indicated that the readers can find the resulting clusters at different cut heights, referring to Figure 6 – figure supplement 1A and Supplementary file 3. The first two sentences of the first paragraph in the Results section titled “Matching the reconstructed head BMNs with their bristles” now read:

The reconstructed BMN projections were next matched with their specific bristle populations. The projections were clustered based on morphological similarity using the NBLAST algorithm (example clustering at cut height 5 shown in Figure 6 – figure supplement 1A,B, Supplementary file 3, FlyWire.ai link 2) (Costa et al., 2016). Clusters could be assigned as BMN types based on their similarity to light microscopy images of BMNs known to innervate specific bristles.

The number of reconstructed BMNs is remarkably similar to what is expected based on bristle counts for each group except for lnOm. Why do you think there is such a large discrepancy there?

We believe that there is a discrepancy between the number of reconstructed BM-InOm neurons and the number expected based on InOm bristle counts because these bristle counts were based on few flies and these numbers appear to be variable. We did not further investigate the numbers of InOm bristles in this manuscript because we only needed an estimate of their numbers, given that there is over an order of magnitude difference in the eye bristles versus any other head bristle population. Therefore, we could relatively easily conclude that the head BMNs were related to the InOm bristles, based on their sheer numbers and their morphology.

Figure 6 - figure supplement 2N, please describe these panels better. Main text says the upper image is from lnOmBMN-LexA, but the figure legend doesn't agree.

We have added text to the figure legend that now makes the contents of panel 2N clear to the reader. Further, we now indicate in the figure legend for each panel, the method used to obtain the labeled neurons (i.e. fill, MCFO, driver), to avoid similar confusion for the other panels.

Figure 6 - figure supplement 4D. How frequently is there a mismatch between the number of BMNs for a given type across hemispheres?

Although the full reconstruction of the BMNs on both sides of the brain was beyond the scope of this work, the BMNs on both sides have since been reconstructed and annotated (Schlegal et al. 2023). We plan to provide more analysis of BMNs on both sides of the brain in a forthcoming manuscript. However, the BMN numbers tend to show agreement on both sides of the brain. The table below shows a comparison between the two sides:

**Author response table 1. sa3table1:** 

Type	LM counts	EM left side	EM right side
Ant	18-22	20	21
Fr	4-7	6	7
FrOr	4-6	6	7
Or	3	3	3
Oc	1	1	1
InOc	3-4	3	3
Vib	13-18	18	19
InOm	607-645	555	557
vOcci/vPoOr	21-33	26	26
MaPa	14-18	15	15
Hau	5	5	5
Taste	31-42	35	29
Vt/PoOc	4-5	5	5
dPoOr	5-9	4	8
dOcci	2-5	3	3

Figures 6 and 7. It would be helpful to include a reference brain in all panels that show cluster morphology. Without landmarks there is nothing to anchor the eye to allow the reader to see the described differences in BMN projection zones and patterns.

While we apologize for not making this specific change, we have made revisions to other parts of the manuscript to better highlight the somatotopic organization among the BMNs (revisions described above). Please note that we now provide FlyWire.ai publicly available links that enable readers to view the BMN projections in 3D. They can also toggle a brain mesh on and off to provide spatial reference.

"BMN somatotopic map": It would be helpful to show or describe in more detail what the unique branch morphology for each zone is. It is quite difficult to appreciate, as the groups also have a lot of overlap. Would the unique regions that the BMN groups innervate be easier to see if you plotted presynaptic sites by group? I am left unsure about whether there is a somatotopic map here.

We made several revisions to the manuscript that address this recommendation. Please see above for a description of these revisions. Please note that we did not examine the fine branch morphological differences between BMN types having overlapping projections. Showing these differences would require more extensive anatomical analysis that is beyond the scope of this work. For showing definitive somatotopy, we focused on the overt differences between BMNs innervating bristles at distant locations on the head.

Overall the strict adherence to the parallel model impacts the interpretation of the data. It would be helpful for the authors to discuss which aspects of the current study are consistent with the parallel model and which results are not consistent.

We made several revisions to the manuscript that address this recommendation. Please see above for a description of these revisions.

Discussion:"Circuits that elicit aimed grooming of specific head locations": In the previous paragraph you mention "BMN types innervating neighboring bristle populations have overlapping projections into zones that correspond roughly to the dorsal, ventral, and posterior head. The overlap is likely functionally significant, as cosine similarity analysis revealed that neighboring head BMN types have common postsynaptic partners. However, overlap between neighboring BMN types is only partial, as they show differing projections and postsynaptic connectivity." Then in this paragraph, you say, "How do the parallel-projecting head BMNs interface with postsynaptic neural circuits to elicit aimed grooming of specific head locations? Different evidence supports the hypothesis that the BMNs connect with parallel circuits that each elicit a different aimed grooming movement (Seeds et al., 2014)." The overlapping postsynaptic BMN connectivity seems in conflict with the claim that the circuits are parallel.

We apologize for this confusion. We now better describe this apparent discrepancy between our results and the parallel model of grooming behavior. We made several revisions to the manuscript that address this recommendation. Please see above for a description of these revisions.

We have made additional changes to the manuscript:

(1) We added Supplementary file 2 that includes links for downloading the image stacks used to generate panels in Figure 1, Figure 2, Figure 3, Figure 4, and figure supplements for these figures. These image stacks are stored in the Brain Image Library (BIL). Rows in the spreadsheet correspond to each image stack. Columns provide information about each stack including: figure panels that each image stack contributed to, image stack title, DOI for each stack (link provides metadata for each stack and file download link), image stack file name, genotype of imaged fly, and information about image stack. References to this file have been made at different locations throughout the text and Figure legends. We also added a section on the BIL data in the Materials and methods entitled “Light microscopy image stack storage and availability”. Old Supplementary file 2 has been renamed Supplementary file 3.

(2) We added a new reference for FlyWire.ai (Dorkenwald et al. 2023) that was posted as a preprint during the revision of this manuscript.